# Understanding and Minimising Outlier Features in Transformer Training

**Bobby He**[1][*]    **Lorenzo Noci**[1]    **Daniele Paliotta**[2]    **Imanol Schlag**[1][†]    **Thomas Hofmann**[1]
[1]Department of Computer Science, ETH Zürich
[2]Machine Learning Group, University of Geneva

## Abstract

Outlier Features (OFs) are neurons whose activation magnitudes significantly exceed the average over a neural network's (NN) width. They are well known to emerge during standard transformer training and have the undesirable effect of hindering quantisation in afflicted models. Despite their practical importance, little is known behind *why OFs emerge during training*, nor *how one can minimise them*.

Our work focuses on the above questions, first identifying several quantitative metrics, such as the kurtosis over neuron activation norms, to measure OFs. With these metrics, we study how architectural and optimisation choices influence OFs, and provide practical insights to minimise OFs during training. As highlights, we introduce a novel unnormalised transformer block, the *Outlier Protected* block, and present a previously unknown benefit of non-diagonal preconditioning optimisers, finding both approaches to significantly reduce OFs and improve quantisation without compromising convergence speed, at scales of up to 7B parameters. Notably, our combination of OP block and non-diagonal preconditioner (SOAP) achieves 14.87 weight-and-activation int8 perplexity (from 14.71 in standard precision), compared to 63.4 int8 perplexity (from 16.00) with a default OF-prone combination of Pre-Norm model and Adam, when quantising OPT-125m models post-training.

## 1 Introduction

Despite their widespread use, our understanding of deep neural networks (NNs) and their training dynamics is very much incomplete. This, in part, reflects the complexity of traversing high-dimensional non-convex loss landscapes but is also symptomatic of the myriad design choices, such as NN architecture and optimiser hyperparameters, that a practitioner must take before training. While standard choices of architecture and optimiser exist, it is often unclear how these choices affect model performance or the emergence of various empirically observed phenomena during NN training.

Outlier Features (OF) are one such training phenomenon. OFs are neurons whose activation magnitudes are significantly larger than average in the same layer, i.e. across NN width [1–3]. They have been widely observed in the popular transformer NN architecture [4–6], as we verify in Fig 1, and are of practical interest because their existence hinders quantisation [3, 7–12]. In particular, OFs cause large dynamic ranges in activations across NN width, leading to high quantisation errors in low precision matrix multiplications. As such, Outlier Feature Emergence (OFE) during training hinders low-precision training and inference, and minimising OFE could yield significant efficiency gains.

In this paper, we tackle OFE from two related angles: by (1) proposing interventions to minimise OFE without affecting model convergence or training stability, using insights motivated through (2) enhancing our understanding of why OFs appear during training. We argue that it is important to first understand why OFs appear during standard NN training dynamics in order to identify which design

---

[*]Correspondence to `bobby.he@inf.ethz.ch`
[†]ETH AI Center

38th Conference on Neural Information Processing Systems (NeurIPS 2024).

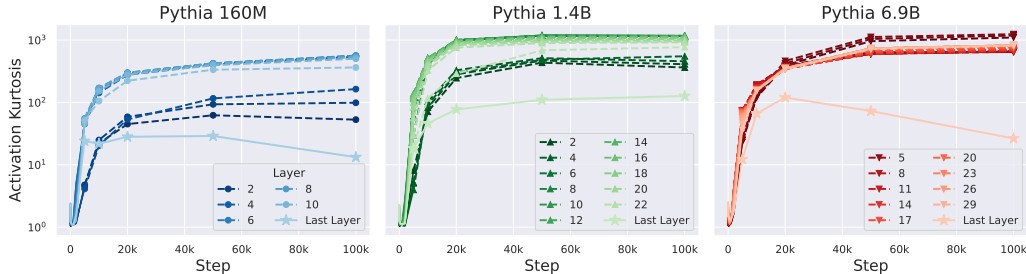

Figure 1: Outlier Features appear in open-source transformers [16] during training, as measured by our Kurtosis metric Eq (1). Our work investigates the design choices that influence their emergence.

choices influence OFE, and how. Though progress has been made [1, 13, 14, 10, 15], the mechanisms behind OFE remain largely unknown.

Alongside the practical motivation of model quantisation, we believe understanding OFs, and their causes during training, to be an interesting research question for several reasons. The emergence of Outlier Features in standard transformer training regimes raises the question if OFs are simply an artifact of certain design choices or a fundamental property of transformer training, essential for best performance. Understanding (the causes of) OFE better helps us to better understand NN training dynamics in general, and the roles played by different design choices. Moreover, we shed light on the differences between models at initialisation compared to during, or after, training. While NN initialisation is more commonly studied owing to analytic tractability [17–30], understanding trained NNs is arguably more important as they exhibit rich feature learning behaviour [31–34], like OFs, that arise during training. In the case of OFs, this has potentially wide-reaching implications, including for NN interpretability, which often focuses on the roles of individual neurons [35].

**Our contributions** Overall, we show that OFE can be mitigated relative to standard practices, and highlight key design choices to do so. We start by introducing OFs, and in particular quantitative metrics to measure OFs in Sec 2. In Sec 3, we study the role of normalisation layers for OFE, and find that existing hypotheses do not fully capture the OF phenomenon. We proceed to show that removing normalisation through our *Outlier Protected* transformer block minimises OFs, without loss of convergence speed or training stability compared to standard transformer blocks. In Sec 4, we consolidate our findings by identifying signal propagation as an important object that can predict OFs during training, and that choices that improve signal propagation during training also minimise OFE. In Sec 5, we consider optimisation hyperparameters, and highlight the importance of large diagonal adaptive learning rates for OFE. Finally, in Sec 6 we demonstrate the performance of our proposals to minimise OFs both at larger scales, up to 7B parameters, and in terms of improved quantisation performance. In the interests of space, in App A.3 we discuss additional related work.

## 2   Problem Setting

Consider an activation matrix $\mathbf{X} \in \mathbb{R}^{n \times d}$ obtained from some neural network layer, where $n$ is the number of batch inputs/sequence positions, and $d$ is the number of neurons across NN width. In a typical NN layer, we matrix multiply $\mathbf{X}$ by a weight matrix $\mathbf{W} \in \mathbb{R}^{d \times d}$ to give $\mathbf{X}\mathbf{W} \in \mathbb{R}^{n \times d}$, with $(\alpha, j)^{\text{th}}$ element: $\sum_{k=1}^{d} \mathbf{X}_{\alpha,k}\mathbf{W}_{k,j}$. This fundamental operation is central to NN computation and can be seen as a sum over $d$ terms, one for each neuron.

Several works have established that if the magnitudes of the summands $\{\mathbf{X}_{\alpha,k}\mathbf{W}_{k,j}\}_{k=1}^{d}$ have large variations, then it becomes difficult to compute their sum in low precision, thereby precluding potential efficiency gains from "vector-wise" quantised training or inference (though significant progress has been made on the latter, [8, 36, 11]). These works have shown that (pre-)trained transformer [37] models possess such a deficiency, which is attributed to the existence of *Outlier Features* (OFs) whose activations are much larger in magnitude compared to the other $d-1$ neurons.

**Measuring OFs**   Existing works have measured OFs in architecture-specific ways [1] or using activation scales $\|\mathbf{X}\|_F^2 \overset{\text{def}}{=} \sum_{\alpha \leq n, j \leq d} \mathbf{X}_{\alpha,j}^2$ [8]. We argue that measuring OFs should be independent of architecture/activation scale: barring exploding/vanishing activation scales, the relative difference in summands is what causes issues for vector-wise quantisation. We use two metrics to measure OFs:

1. **Kurtosis of neuron activation RMS**: Let $\mathbf{s} \in \mathbb{R}^d$, such that $\mathbf{s}_j = \sqrt{\frac{1}{n}\sum_{\alpha=1}^n \mathbf{X}_{\alpha,j}^2}$, be the vector of root mean-squared activations across inputs.[3] Then, let $\text{Kurt}(\mathbf{X})$ be the ratio of the fourth moment $m_4$ to the squared second moment $m_2$ over the empirical distribution of $\mathbf{s}$:

$$\text{Kurt}(\mathbf{X}) = \frac{m_4(\mathbf{X})}{m_2(\mathbf{X})^2} \overset{\text{def}}{=} \frac{\frac{1}{d}\sum_{j=1}^d \mathbf{s}_j^4}{\left(\frac{1}{d}\sum_{j=1}^d \mathbf{s}_j^2\right)^2} \tag{1}$$

   We see that $\min(\text{Kurt}(\mathbf{s})) = 1$ when all $\mathbf{s}_j$ are equal and no outlier features exist, and $\max(\text{Kurt}(\mathbf{X})) = d$, which is the limit when $d-1$ neurons have activation magnitudes dominated by a single outlier feature.

2. **Max-Median Ratio** (across neurons): A metric for OFs more aligned with the original motivation of studying variation in summand magnitudes. Specifically, we compute:

$$\text{MMR}(\mathbf{X}) \overset{\text{def}}{=} \text{Aggregate}_\alpha \left( \frac{\max_j |\mathbf{X}_{\alpha,j}|}{\text{median}_j |\mathbf{X}_{\alpha,j}|} \right), \tag{2}$$

   or in words, the max neuron divided by the median absolute neuron, aggregated in some permutation invariant way across inputs. We typically use the mean to aggregate over inputs, but could also take e.g. median or max. MMR takes a minimum value 1 when all activations are identical in magnitude, and is unbounded when a dominant outlier feature exists.

Variants of $\text{Kurt}(\mathbf{X})$ have previously been proposed [14, 3], but our formulation in Eq (1) aggregates activations over inputs first, which allows us to link OFs and signal propagation in Sec 4. Though we focus our analysis on $\text{Kurt}(\mathbf{X})$, Figs 11, 13 and 17 show that both our OF metrics are highly correlated. In this work, we measure OFs on the residual stream across different layers/blocks. For example, for Pre-Norm or Post-Norm models this is $\mathbf{X}_{\text{out}}$ in the notation of App A.1.[4]

**Experimental Setup**   Throughout this work, we train transformers on the next-token language modelling task, and study OFs, on a range of datasets, including: 1) CodeParrot,[5] 2) Languini Books [39], 3) BookCorpus [40] and English Wikipedia,[6] and 4) FineWeb-Edu [41]. Unless stated otherwise our experimental results are conducted on CodeParrot, but importantly our conclusions regarding OFs are consistent throughout across language modelling datasets. In App E.1, we also explore OFs in image classification settings with other architectures like Vision Transformers [42] and MLPs.

In terms of architecture and optimiser, our default choices are the Pre-Norm transformer (App A.1) and AdamW [43] respectively, which are known to be prone to OFs, e.g. Fig 1. Our default architecture scale has width $d = 768$ and 6 layers, giving around 130M parameters, but we demonstrate our findings continue to hold at larger scales (up to 7B parameters) in Secs 3 and 6. In Secs 3 and 4 we examine alternatives to the Pre-Norm architecture and their effects on OFs, keeping AdamW as optimiser, while from Sec 5 onwards we additionally consider modifications to AdamW. Further experimental details and results beyond the main paper can be found in Apps D and E respectively.

## 3   Normalisation Layers and Outlier Features

Several works have highlighted the architectural choice of *Layer Normalisation* (LN) [44] as a cause of OFE [1, 7, 15]. LN belongs to a family of normalisation (Norm) layers commonly used in sequence models, which normalise a representation vector $\boldsymbol{x} \in \mathbb{R}^d$ across the width dimension independently

---

[3]We do not centre $\mathbf{X}$ in $s_j$ for simplicity. Fig 15 shows centring makes no qualitative difference for $\text{Kurt}(\mathbf{X})$.

[4]As $\text{MMR}(\mathbf{X})$ is invariant to normalisation layers like RMSNorm without trainable parameters [38], there is some redundancy here in where exactly we take OF measurements.

[5]`https://huggingface.co/datasets/transformersbook/codeparrot-train`.

[6]`https://huggingface.co/datasets/google/wiki40b`, using the same setup as [15].

for different sequence positions. In general, for a centring scalar $c \in \{0, 1\}$, a Norm layer maps $\boldsymbol{x}$ to:

$$\text{Norm}(\boldsymbol{x}) = \frac{\boldsymbol{x} - c\mu(\boldsymbol{x})}{\sigma(\boldsymbol{x})} \odot \boldsymbol{\gamma} + \boldsymbol{\beta}, \quad \mu(\boldsymbol{x}) = \frac{1}{d}\sum_{i=1}^{d} \boldsymbol{x}_i, \quad \sigma(\boldsymbol{x})^2 = \frac{1}{d}\sum_{i=1}^{d}(\boldsymbol{x}_i - c\mu(\boldsymbol{x}))^2 \quad (3)$$

LN is when $c = 1$, with a trainable scale $\boldsymbol{\gamma}$ and bias $\boldsymbol{\beta}$ vectors initialised to all 1s and 0s respectively.

Previous works have attributed OFE to the $\boldsymbol{\gamma}, \boldsymbol{\beta}$ parameters of LN incurring outliers during training [1, 7]. It is therefore natural to ask if simpler Norms with different formulations of Eq (3) remove OFE. In particular, *Root Mean Square Normalisation* (RMSNorm) [45] is a commonly used Norm known to be as performant as LN in Transformer training [46, 47]. Compared to LN, RMSNorm fixes the bias $\boldsymbol{\beta} = 0$ and removes the centring by setting $c = 0$, which highlights that centring is not a crucial operation in modern sequence modelling practices. One step further would be to remove trainable parameters entirely by fixing $\boldsymbol{\gamma} = 1$, simply projecting $\boldsymbol{x}$ to the hypersphere of norm $\sqrt{d}$. This is dubbed *Simple RMSNorm* (SRMSNorm) by Qin et al. [38], who find that SRMSNorm has minimal performance degradation but is more computationally efficient than LN and RMSNorm.

We compare these different Norms in Fig 2, where we see that independent of Norm choice, all Pre-Norm transformers incur OFE: the peak kurtosis during training across Norms is over 4 orders of magnitude larger than initialisation. We also show OFE not only in Pre-Norm [48, 49] but also Post-Norm [37] blocks (more details on transformer blocks in App A.1), highlighting OFE occurs independent of where Norms are placed. In this experiment, the Pre-SRMSNorm model has highest Kurtosis, despite its lack of trainable Norm weights.

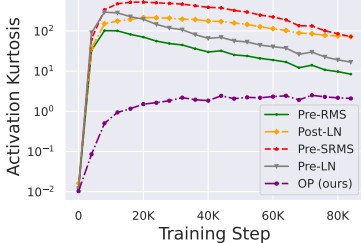

Figure 2: Kurtosis becomes large (i.e. OFE) when training with different Norms at 130M scale. We plot the residual stream entering the 2nd of 6 blocks. Other layers in Fig 14.

Having established that removing trainable weights in Norms still results in OFE, the next question we ask is: *how does removing standard Norms entirely influence Outlier Feature emergence*?

**Recovering training benefits in unnormalised Transformers**  This is a challenging question to answer, not least because comparing OFE in architectures that converge at different speeds may not be a fair comparison: Norms are well known to be an important component in most NN architectures, providing various benefits for initialisation, convergence speed, and training stability. Thus, to answer the above question, we must first review different hypotheses for the benefits of Norms in transformer training dynamics in order to motivate a novel transformer block that matches the Pre-Norm block in convergence speed, while eschewing standard Norm layers.

Several works [50–55, 23, 27, 28, 30] have observed that the initialisation benefits of Pre-Norm architectures can be recovered in unnormalised residual models using downweighted residual branches, through a theory known as Signal Propagation (Signal Prop) [17, 18, 56]. Notably, Brock et al. [53] achieve state of the art performance on the ImageNet benchmark using unnormalised convolutional architectures. However, it has been observed that fixing Signal Prop at initialisation is not sufficient to fully capture the benefits of Norms for training dynamics in unnormalised transformers [28, 30], which implies that Norms have training benefits specific to the self-attention based transformer model.

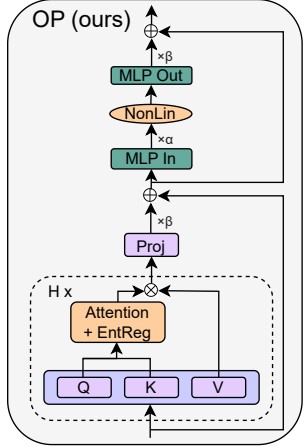

Figure 3: The Outlier Protected Transformer Block. We remove Pre-Norms and replace them with an Entropy Regulation mechanism to prevent entropy collapse, as well as downscaling residuals with $\beta < 1$.

At the same time, Zhai et al. [57] show *Entropy Collapse*, where the Stochastic attention matrix has rows with low entropy and each sequence position attends to only one position instead of many, to be a key transformer training instability (see Eq (10)). Entropy collapse occurs because large attention logits saturate the softmax, and several *Entropy Regulation* (EntReg) mechanisms have been proposed to control the attention logits and thus prevent entropy collapse. Existing entropy regulating methods include QK-Norm [58, 59], *tanh* thresholding (Grok-1), $\sigma$Reparam [57] and clamping the QK logits (DBRX). In standard Pre/Post-Norm attention blocks, a Norm layer appears before Query and Key weights and implicitly regulates attention entropy, to an extent.

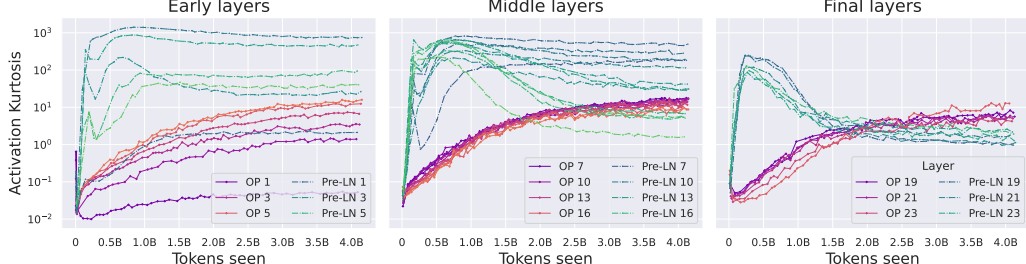

Figure 4: Our OP block mitigates OFE. We plot activation kurtosis of the residual stream across layers. Experiments are at 1.2B scale on Languini Books using a max AdamW learning rate of 0.001 with linear warmup for the first 1.5% steps and linear decay thereafter. Notice the shared log-scaled y-axis: activation kurtosis is consistently (up to 4 orders of magnitude) lower in OP block, particularly in earlier layers. Also, peak kurtosis during training is always higher in Pre-LN. The OP model also removes the final LN before unembedding; the effect of the final LN on OFE is shown in Fig 10.

Our key insight is to combine ideas from Signal Propagation and Entropy Collapse prevention to remove Normalisation layers while keeping their training benefits. This brings us to our *Outlier Protected* (OP) Block, Fig 3, which replaces the Pre-Norm block by removing its normalisation layers in both Attention and MLP sub-blocks, and making three additional changes: 1) downweighting residual branches with some $\beta = O(1/\sqrt{\text{depth}}) < 1$ to recover Signal Prop benefits of Pre-Norms [51, 23, 27, 30], 2) adding an Entropy Regulation mechanism to prevent Entropy Collapse; we mainly use QK-Norm as it is relatively simple and performed well in all of our settings, but present experiments with tanh in App E.2, and 3) (optionally) scaling the inputs before the MLP nonlinearity by a scalar $\alpha$ to ensure the nonlinearity inputs are of order 1, as derived by Brock et al. [53] using straightforward Signal Prop arguments. App B presents a mathematical description of the OP block.

In Tab 1, we show that our Outlier Protected block matches the standard Pre-LN block in terms of convergence speed at scales up to 1.2B parameters when trained with next token prediction on the Languini books dataset [39] for nearly 4.5B tokens.[7] In App E.2, we ablate our OP block and show that the lack of an entropy regulation mechanism without normalisation layers causes training instabilities. This demonstrates that preventing entropy collapse is necessary to match training stability and convergence speed in unnormalised Transformers.

Table 1: OP matches Pre-LN performance at scales up to 1.2B params, on Languini Books [39].[7]

| Params | Block | Eval PPL |
|--------|-------|----------|
| 100M | Pre-LN | 19.1 |
| | OP | 18.9 |
| 320M | Pre-LN | 16.2 |
| | OP | 16.2 |
| 1.2B | Pre-LN | 13.9 |
| | OP | 13.9 |

We note that independent of OFs, the OP block is interesting in its own right because it shows that the initialisation-time Signal Prop and Entropy Collapse benefits of Norms in Transformers can be disentangled, and also reveals what was missing in previous methods that used Signal Prop arguments to correct initialisation defects in simplified unnormalised Transformers [28, 30]. However, we now focus on the benefits of the Outlier Protected block in reducing outlier features.

**Removing Norms mitigates Outlier Features**   In Fig 2 we see that the Outlier Protected (OP) block greatly reduces OFE compared to standard blocks. Fig 4 presents the corresponding plots in our 1.2B parameter experiments using our kurtosis metric, for different layers. We draw several consistent conclusions: 1) peak kurtosis across the course of training is consistently higher in Pre-LN, sometimes by over 2 orders of magnitude, across different layers; 2) kurtosis across training is usually higher in Pre-LN (up to 4 orders of magnitude here), especially at early training times and in earlier layers; 3) OFE (measured via our metrics) does not need to be monotonic in training time. Together, these findings suggest that the OP block will lead to more quantisable models compared to standard Pre-Norm, as we will show in Sec 6. Tab 4 ablates the effect of Norm positioning on OFE.

Nevertheless, we observe in Fig 4 that kurtosis still slightly increases in our OP blocks (to relatively modest values, maximum around 20), usually monotonically throughout training. Moreover, the question of why normalisation layers cause outlier features is still unanswered despite the clear evidence that removing them mitigates OF prevalence. We investigate these questions next.

---

[7]We train for 4.2B tokens at 1.2B scale as this took 24 hours on 4 A100 80GB GPUs; we were unable to train for longer due to compute constraints. Scales under 1B were trained on a single A5000 or RTX-2080Ti GPU, taking around 2 days for 3.3B tokens (or equivalently, 50K steps at batch size 128 and sequence length 512).

Sec 3 key takeaways: normalisation layers and OFE.

- OFE still occurs for weight-less or uncentred Norms, & both Pre/Post-Norm (Figs 2, 14 and 17).
- The OP Block (Fig 3) matches Pre-LN training speed/stability (Tabs 1 and 3), without standard Norms. It does so through an Entropy Regulation method to prevent attention entropy collapse.
- The OP Block greatly reduces OFE compared to standard blocks (Figs 2, 4 and 13).

## 4 Signal Propagation and Outlier Features

To better understand why OFs still appear (albeit greatly reduced) in the OP block, and why normalisation layers cause OFs, we examine *Signal Propagation* behaviour during training and its effect on OFE. This will also clarify why modifications that improve Signal Propagation reduce OFE [10]. Signal Propagation [17, 18, 56, 23, 25, 27, 28] studies the *input-wise* Gram matrix $\Sigma_I = \mathbf{X}\mathbf{X}^\top \in \mathbb{R}^{n \times n}$, and how $\Sigma_I$ evolves in deep NNs for different layer features $\mathbf{X} \in \mathbb{R}^{n \times d}$.

On the other hand, as we will see below, our kurtosis metric is related to the *feature-wise* Gram matrix $\Sigma_F \stackrel{\text{def}}{=} \mathbf{X}^\top \mathbf{X} \in \mathbb{R}^{d \times d}$. Recall our kurtosis is a normalised $4^{\text{th}}$ moment of $\mathbf{X} \in \mathbb{R}$, normalised by the square of the second moment $m_2(\mathbf{X}) = \frac{1}{nd} \sum_{\alpha \leq n, j \leq d} \mathbf{X}_{\alpha,j}^2 = \frac{1}{nd}\|\mathbf{X}\|_F^2$. Because kurtosis is scale-invariant we can consider the setting where $m_2(\mathbf{X}) = 1$ and the average squared activation is 1 without loss of generality.[8] In this case, $\text{Tr}(\Sigma_I) = \text{Tr}(\Sigma_F) = nd$ by the cyclic trace property.

Then, our kurtosis, Eq (1), is $\text{Kurt}(\mathbf{X}) = \frac{1}{d}\sum_{j=1}^d \left(\frac{1}{n}\sum_{\alpha=1}^n \mathbf{X}_{\alpha,j}^2\right)^2 = \frac{1}{d}\sum_{j=1}^d (\frac{1}{n}\Sigma_F)_{j,j}^2$, which is simply a second moment (or average squared value) of diagonal entries of the feature-wise Gram matrix $\Sigma_F$. At the same time, again by the cyclic property of the trace, we have:

$$\text{Tr}(\Sigma_F \Sigma_F) = \text{Tr}(\mathbf{X}^\top \mathbf{X}\mathbf{X}^\top \mathbf{X}) = \text{Tr}(\mathbf{X}\mathbf{X}^\top \mathbf{X}\mathbf{X}^\top) = \text{Tr}(\Sigma_I \Sigma_I) \tag{4}$$

$$\implies n^2 d \cdot \text{Kurt}(\mathbf{X}) + \sum_{i,j \leq d; i \neq j} \left(\Sigma_F\right)_{i,j}^2 = \sum_{\alpha,\beta \leq n} \left(\Sigma_I\right)_{\alpha,\beta}^2 \tag{5}$$

In words, Eq (4) tells us that the sum of squared elements of $\Sigma_F$ is equal to the sum of squared elements of $\Sigma_I$. On the left of Eq (5) we decompose Eq (4) into our feature-wise kurtosis (Eq (1), of interest for OFE), plus the sum of squared off-diagonal elements of $\Sigma_F$, equal to the sum of squared elements of $\Sigma_I$ on the right. Hence, it is clear that Signal Propagation is relevant for OFE. Contrary to most existing works in Signal Propagaton, Eq (5) is true throughout training, not only at initialisation.

In particular, we see that the right-hand side of Eq (5) captures both the (normalised) activation norms across inputs $\sum_{\alpha \leq n} \left(\Sigma_I\right)_{\alpha,\alpha}^2$ from the diagonal terms, and inner products between inputs $\sum_{\alpha,\beta \leq n; \alpha \neq \beta} \left(\Sigma_I\right)_{\alpha,\beta}^2$ in the off-diagonals. If $\mathbf{X}$ is the output of a Norm layer, then $\frac{1}{d}\Sigma_I$ becomes a cosine similarity matrix with diagonals equal to 1. Deep NNs, and Transformers in particular, are well known to be susceptible to a particular Signal Prop defect called *rank collapse* [60, 25] where this cosine similarity matrix $\frac{1}{d}\Sigma_I$ degenerates to the all ones matrix and all inputs look identical to a deep layer. Noci et al. [27] and He et al. [28] demonstrate that, at least at initialisation, the off-diagonals of $\Sigma_I$ are positive and increase monotonically with depth in deep Transformers towards rank collapse, even with Signal Prop inspired modifications that ensure a non-degenerate deep limit exists.

**Bad Signal Prop encourages OFE** For OFE, the upshot of these observations is that poor Signal Propagation (in terms of large off-diagonal values of $\Sigma_I$, close to rank collapse) will make the right-hand side of Eq (5) large (the rank collapsed limit has RHS $n^2 d^2$, compared to $nd^2$ when the inputs are orthogonal and $\Sigma_I$ is diagonal). In turn, this puts pressure on the LHS, which contains the feature kurtosis, to be large, hence OFE. This argument is not fully rigorous because the off-diagonals $\sum_{i,j \leq d, i \neq j} \left(\Sigma_F\right)_{i,j}^2$, which captures correlations between different neuron features, could increase on the LHS to allow the kurtosis to remain low.[9] Theoretically predicting this behaviour deep into modern NN training is outside the scope of this work; we note that while it is possible to write down training trajectories in feature learning regimes [33, 34], most works interpreting feature learning in

---

[8]In all experiments concerning signal propagation (i.e. input-wise correlations or equivalently, the elements of $\Sigma_I$), we first scale $\mathbf{X}$ down by $\sqrt{m_2(\mathbf{X})}$ to give $m_2(\mathbf{X}) = 1$ and make $\mathbf{X}$ scale invariant.

[9]Indeed, Fig 33 shows the link between OFs and signal propagation depends on the diagonality of optimiser.

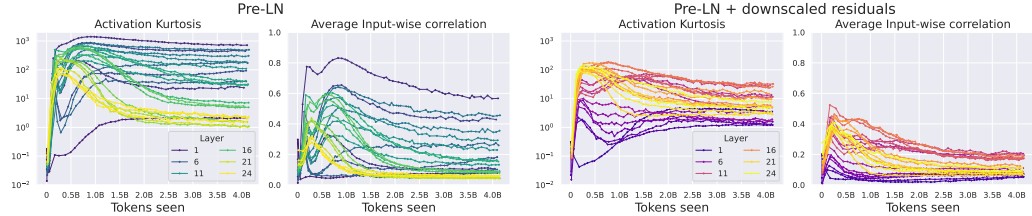

Figure 5: Adam-trained Pre-LN layers at 1.2B scale with extreme OFE (left) are those with bad Signal Prop close to rank collapse during training (centre left). (**Right vs. left two plots**) Downweighting residual branches improves signal propagation during training and results in smaller OFs, particularly in early layers. Respective plots for OP (with & without final LN before unembedding) in Fig 10.

NNs focus only on a single gradient step [61–64]. Having said that, we formalise the intuition of bad Signal Prop leading to larger feature kurtosis in the context of Gaussian features in Prop G.1.

In any case, we can empirically study the link between bad signal propagation and OFEs, which we do in Figs 5 and 10 for Pre-LN & OP blocks trained with AdamW at 1.2B scale on Languini Books. We plot both the layerwise evolution of the kurtosis on the left and the average off-diagonal entry of $\frac{1}{d}\Sigma_I = \frac{1}{d}\mathbf{X}\mathbf{X}^\top$ (i.e. the average input-wise correlation) on the right, normalised so that $m_2(\mathbf{X}) = 1$.

As suggested by Eq (5), we see a strong association between kurtosis and Signal Propagation: the layers with larger kurtosis tend to be the ones with larger input correlations, and vice versa. In particular, in Fig 5, we see that the Pre-LN layer (2 in this case) with the most extreme OFE (kurtosis peaking over 1000) is precisely the one with the worst Signal Propagation closest to rank collapse (average input correlation peaking over 0.8) during training. Moreover, the trajectory of kurtosis closely tracks the trajectory of input correlations throughout training, with their peaks appearing at similar training steps, across layers. Fig 12 shows that the Adam-trained Pythia models [16] are very close to rank collapse, which partially explains their large OFs in Fig 1.

Given that Signal Propagation characteristics during training depict how a model creates structure (through increasing or decreasing the inner product for different inputs) in its layer representations to best learn the task at hand, our results suggest that OFs occur partly due to the inherent nature of the task that the model is trained on, particularly in architectures that are less prone to OFs, such as our OP block. In Transformers, this appears most apparent in the inputs to the final unembedding layer, which are linearly projected to the predictions: they tend to have similar kurtosis levels in both OP and Pre-Norm blocks, and the most extreme OFE rarely occurs in the final layers, (Figs 1, 4, 14 and 18). We hypothesise this is because extreme OFE in late layers would imply high kurtosis which could imply representations close to rank collapse by Eq (5), from which it may be hard to learn useful linear predictions with optimisers like Adam.

The correlation we identify between OFE and Signal Propagation also allows us to observe that interventions that worsen Signal Propagation *during training* cause increased OFE. Likewise, methods improving Signal Propagation throughout training help to mitigate OFE. This can be seen in Fig 5 for downscaled residuals, $h(x) = x + \beta f(x)$ with some $\beta \ll 1$, which Wortsman et al. [10] show improve quantisation on vision-language models. We explore this link further in terms of normalisation layers and other architectural choices inspired by Signal Prop in App C.

## 5 Optimisation Choices and Outlier Features

So far, we have focused on the impact of architecture for OFE. As a result, up until now all models have been trained with AdamW [43] optimiser, which is an adaptive diagonal preconditioner, with default hyperparameters e.g. $\beta_1 = 0.9, \beta_2 = 0.999, \epsilon = 10^{-8}$. As OFE is a training phenomenon, it is important to also consider the role of optimsation choices, which we now explore.

**Learning Rate**   Perhaps unsurprisingly, we find that using smaller LRs leads to reduced OFE during training (Figs 6, 25 and 26), across different architectures. In these cases, slightly reducing the maximum LR in our scheduler (e.g. $0.001 \rightarrow 0.0003$ in Fig 6) did not lead to a loss in convergence speed (Fig 27), highlighting that one should use a smaller LR to avoid OFs if convergence is not affected.

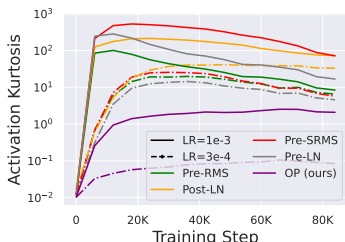

Figure 6: Smaller LRs lead to smaller OFs across different blocks.

**Adaptivity** Having seen the importance of LR in OFE, we now assess the impact of adaptive LRs through the $\epsilon$ hyperparameter in Adam, where the Adam update is $-\eta m_t/(\sqrt{v_t}+\epsilon)$, $\eta$ is global LR, and $m_t$ and $v_t$ denote first and second-moments of each parameter's gradient, respectively. $\epsilon$ dampens adaptive preconditioning, with larger $\epsilon$ reducing adaptivity for parameters with smaller $v_t$. In Figs 7, 29 and 9 and Tab 2 we show that increasing $\epsilon$ also reduces OFE. Thus, one should increase $\epsilon$ to reduce OFE, if convergence is not impacted (like Fig 28).

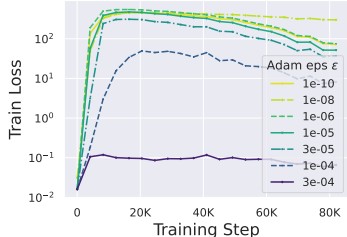

Figure 7: Larger Adam $\epsilon$ reduces OFs in 130M Pre-LN transformers.

**Non-Diagonal Preconditioners** To push the question of adaptivity further we consider the effect of diagonal adaptivity and OFE. First-order optimisers like AdamW [43] or AdaFactor [65] are the de-facto optimisers in deep learning, acting as *diagonal* preconditioners where each parameter has its own adaptive learning rate to scale its gradient. On the other hand, popular second-order optimisers like K-FAC [66] or Shampoo [67, 68] are *non-diagonal* preconditioners acting on the full gradient. Second-order optimisers are known to converge faster *per-update* than first-order methods, but first-order optimisers are much more widespread due to the additional overheads of non-diagonal preconditioning. We provide background on different NN optimisers in App A.2.

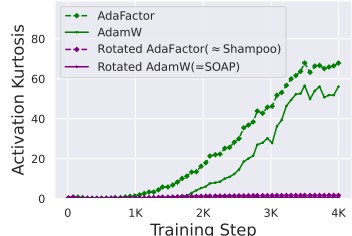

Figure 8: Diagonal optimisation on rotated parameters reduces OFs.

Recently, Vyas et al. [69] established a precise connection between first and second-order optimisers, namely: Shampoo [67] can be seen as running the diagonal AdaFactor [65] method after first rotating into the eigenbasis of Shampoo's preconditioner, before rotating back. This insight disentangles two effects of non-diagonal preconditioners: 1) transforming the parameter space in which one optimises via a rotation, and 2) using a diagonal optimisation method in the rotated space. Vyas et al. [69] use this insight to propose the SOAP optimiser, which applies AdamW in the rotated parameter space obtained from Shampoo's eigenbasis. SOAP is shown to converge slightly faster per step than Shampoo, which itself converges faster per step than AdamW/AdaFactor (verified in Fig 32).

In Fig 8, we compare OFE in popular first-order optimisers, 1) AdamW and 2) AdaFactor, to rotated non-diagonally preconditioned alternatives, 3) SOAP and 4) AdaFactor in Shampoo's eigenbasis (akin to Shampoo as shown by Vyas et al. [69]), trained using 130M Pre-Norm transformers on CodeParrot. We clearly see that rotating the parameter space in which one optimises, as done in SOAP/Shampoo, mitigates OFEs. This effect is independent of the diagonal preconditioner used and occurs even in spite of the Pre-Norm layers, which we know are susceptible to OFs, as in Sec 3.[10] To further highlight the importance of diagonal adaptivity, in App E.1 we compare SGD to Adam on a setting where SGD can match Adam's convergence speed: image classification with an MLP. There, we again see that the non-adaptive SGD suffers less from OFs compared to the diagonal Adam.

Elhage et al. [14] show a similar result as Fig 8 using random rotations, while QuaRot [11] applies random Hadamard rotations to remove OFs for post-training quantisation (PTQ) using computational invariance. SpinQuant [70] extends QuaRot using learnt rotations but again only considers inference time, after OFs have emerged during training. Both QuaRot and SpinQuant incur additional overheads to apply rotations in the forward pass at inference time. In contrast, non-diagonal preconditioners optimise using "learnt" rotations that adapt during training, which leaves the forward pass intact post training and enables the improvement in convergence speed per step that Shampoo/SOAP enjoy relative to AdamW/AdaFactor (seen in Fig 32 and Tab 2). In sum, our results reveal an additional appeal of second-order optimisers: not only are they faster to converge per step, but they also lead to models that are not as prone to OFs and are thus easier to quantise, as we will see in Sec 6.

**Breaking down kurtosis updates** The findings in this section point to the importance of large diagonal adaptive LRs for OFE. This motivates us to break down the updates to kurtosis into terms of different powers in the learning rate $\eta$, in App F. We find that sub-leading order updates (in terms of LR) are the key driver in increasing kurtosis, providing a consistent mechanism for OFE that encapsulates our different observations concerning the roles of optimiser and architecture.

---

[10]This, coupled with our findings with the OP block in Sec 3, suggests that extreme OFs found in LLMs are due to an *interaction* of the architectural choice of Norm layers and the optimiser choice of diagonal adaptive preconditioners. Note we use Pre-SRMSNorm in Fig 8, which doesn't have trainable Norm parameters.

> **Sec 5 key takeaways: Optimisation Choices and OFE.**
>
> - Large diagonal adaptive LRs can lead to Outlier Features during training (Figs 6, 25, 26 and 8).
> - Reducing AdamW adaptivity via increased $\epsilon$ hyperparameter reduces OFs (Figs 7, 29 and 37).
> - The non-diagonal preconditioning of second-order optimisers greatly minimises OFE (Fig 8 and Tab 2), even with OF-prone architectures like Pre-Norm.

## 6  Additional Experiments

We conclude our study with additional experiments regarding scaling and quantisation properties of our suggestions to minimise OFs. Further experimental details can be found in App D.

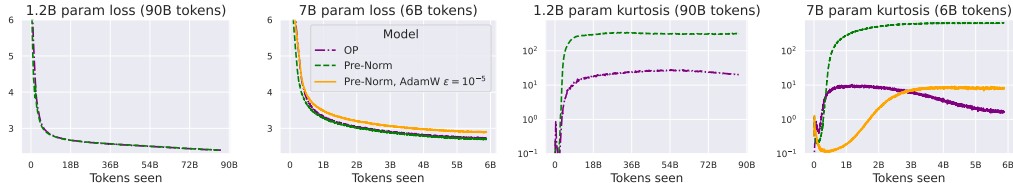

Figure 9: Loss (left two plots) and kurtosis (right two plots) curves for different models trained with AdamW. Our conclusions on OFs continue to hold when scaling both token count and model scale.

**Scale**  Up until now, we have studied OFE and how we can minimise it at scales up to 1.2B parameters and 5B tokens. We now consider scaling both model size and training length in terms of loss performance and OFE. We compare scaling the OP and Pre-Norm blocks keeping AdamW as optimiser; it would be interesting to considering scaling experiments using second-order optimisers (in distributed settings e.g. [68, 71]) in future work. The dataset is FineWeb-Edu [41]. We warmup the LR for 5% of training steps to a maximum value (0.001 and 0.0003 for 1.2B and 7B respectively), before cosine decay. Due to computational cost, very little hyperparameter tuning was performed with these experiments and all default hyperparameters were optimised for the Pre-Norm baseline.

So far we have studied OFE in training token counts that are relatively small compared to the "compute-optimal" Chinchilla recipe [72]. In Fig 9 (first & third subplots), we scale the number of training tokens at 1.2B parameter scale to around 90B, which is beyond the Chinchilla prescription. We see that the OP block is indeed able to closely match Pre-Norm loss at longer token counts, but still benefits from significantly reduced OFs (at least an order of magnitude lower kurtosis throughout training) despite the aggressive AdamW LR and long training horizon.

In Fig 9 (second and fourth subplots), we additionally scale the model size to 7B parameters, a scale that has previously been hypothesised to be a sharp cutoff above which "systematic" OFs emerge [8]. Due to cost, we only train for 6B tokens, which is more than enough for extreme OFs to emerge (over 600 kurtosis averaged across residual layers) with Pre-Norm layers. On the other hand, the OP block has peak kurtosis under 10 and yet still matches Pre-Norm loss performance at 7B scale. Although increasing AdamW $\epsilon$ from $10^{-8}$ to $10^{-5}$ also reduces peak kurtosis to under 10 with the OF-prone Pre-Norm model, it leads to a significant decrease in convergence speed in this setting.

**Quantisation**  Returning to our original motivation, we investigate the effect of our suggested architectural and optimisation choices to minimise OFs in terms of quantisation. In Tab 2, we take the OPT-125m [6] setting of Bondarenko et al. [15], training models using AdamW in standard mixed FP16/FP32 precision on BookCorpus+Wikipedia for around 12B tokens. Post training, we quantise (PTQ) to int8 weight-and-activations, using the same quantisation recipe as [15]. We report standard deviation over 3 seeds for PTQ, as random subsets of data are used to estimate the quantiser range.

Tab 2 compares both the standard precision perplexity (FP16/32) and also 8-bit quantised perplexity (W8A8) across architecture and optimiser choices. We additionally present our kurtosis metric, Eq (1), calculated after training and averaged across layers. We compare 3 different transformer blocks: a) standard Pre-LN, b) the Gated Attention block proposed by Bondarenko et al. [15] to reduce OFs, and c) our OP block, as well 5 different optimisation setups that are added one after another: 1) the default hyperparameters of [15], 2) removing dropout regularisation, 3) increasing the maximum LR from $4 \times 10^{-4} \to 10^{-3}$, 4) increasing AdamW $\epsilon$ from $10^{-8} \to 10^{-5}$, and 5) changing AdamW to SOAP optimiser (keeping $\epsilon = 10^{-8}$). Optimiser choices 2) and 3) were designed to improve standard precision performance, albeit potentially at the detriment of quantisation performance due to increased OFs. Optimiser choices 4) and 5) were chosen to reduce OFs, from our findings in Sec 5.

Table 2: Average kurtosis across layers, plus standard precision (FP16/32) and quantised int8 (W8A8) perplexity of various 125m OPT models [6] trained on BookCorpus+Wikipedia [15]. Kurtosis strongly correlates with int8 error across settings, and the best int8 setup combines our architectural (OP) and optimiser (SOAP) suggestions, with only 1.2 kurtosis and 0.16 perplexity increase.

| Architecture | Optimiser Hyperparameters | Kurtosis | FP16/32 ($\downarrow$) | W8A8 ($\downarrow$) |
|---|---|---|---|---|
| Pre-LN | Default from [15] | 25.6 | 16.00 | $63.4_{\pm 50.1}$ |
| | $-$Dropout Regularisation | 46.7 | 15.53 | $105.9_{\pm 25.0}$ |
| | $+$Big LR ($4 \times 10^{-4} \rightarrow 10^{-3}$) | 61.4 | 15.40 | $59.0_{\pm 10.7}$ |
| | $+$Big Adam $\epsilon$ ($10^{-8} \rightarrow 10^{-5}$) | 43.6 | 15.19 | $216.2_{\pm 87.5}$ |
| | $+$Non-diag Precond (SOAP) | 5.1 | 14.80 | $16.43_{\pm 0.12}$ |
| Gated Attention [15] | Default from [15] | 4.5 | 15.63 | $16.2_{\pm 0.09}$ |
| | $-$Dropout Regularisation | 28.8 | 15.08 | $18.7_{\pm 0.09}$ |
| | $+$Big LR ($4 \times 10^{-4} \rightarrow 10^{-3}$) | 28.8 | 14.99 | $17.04_{\pm 0.09}$ |
| | $+$Big Adam $\epsilon$ ($10^{-8} \rightarrow 10^{-5}$) | 16.0 | 14.78 | $15.54_{\pm 0.01}$ |
| | $+$Non-diag Precond (SOAP) | 16.7 | 14.65 | $15.64_{\pm 0.01}$ |
| OP (ours) | Default from [15] | 3.6 | 15.64 | $16.01_{\pm 0.01}$ |
| | $-$Dropout Regularisation | 7.1 | 15.15 | $15.78_{\pm 0.03}$ |
| | $+$Big LR ($4 \times 10^{-4} \rightarrow 10^{-3}$) | 12.0 | 14.96 | $15.60_{\pm 0.01}$ |
| | $+$Big Adam $\epsilon$ ($10^{-8} \rightarrow 10^{-5}$) | 6.0 | 14.89 | $15.62_{\pm 0.02}$ |
| | $+$Non-diag Precond (SOAP) | **1.2** | 14.71 | $\mathbf{14.87}_{\pm 0.01}$ |

As seen in Tab 2, our findings throughout the rest of our paper are validated. Firstly, our kurtosis metric to measure OFs is indeed highly correlated with *quantisation error*, which we define as the increase in perplexity from FP16/FP32 to W8A8. For example, the Pre-LN model without SOAP has consistently high kurtosis (over 25), and also consistently poor performance at W8A8 (over 45 quantisation error across all AdamW optimiser settings). Secondly, our OP block has consistently low kurtosis (below 12) compared to the other models, and this directly translates to low quantisation error (below 0.73 across all optimiser settings). This low kurtosis/quantisation error with OP block holds true even for aggressive optimiser choices, like large diagonal adaptive LRs, that improve standard precision perplexity but also increase kurtosis. Moreover, the baseline Gated Attention model of [15] struggles with OFs when dropout is removed and a large learning rate is used, leading to increased quantisation error (2-4 perplexity increase), but increasing AdamW $\epsilon$ as suggested in Sec 5 reduces kurtosis (29 to 16) and quantisation error to 0.76, whilst also improving FP16/32 perplexity.

Finally, changing AdamW to SOAP optimiser either dramatically reduces kurtosis (in the case of Pre-LN and OP) or matches the kurtosis reduction of increasing Adam $\epsilon$ (for Gated Attention), while also improving mixed precision performance for all models.[11] This leads to the only non-catastrophic W8A8 perplexity (16.43) with Pre-LN, and the best overall W8A8 model when combining SOAP optimiser with our OP architecture (14.87 perplexity, with only 0.16 degradation from standard precision). This result highlights the combination of our architectural and optimiser suggestions for minimising OFs as a promising approach to training fast-converging and easily quantisable models.

## 7 Discussion

The goal of this work was to better understand the emergence of Outlier Features during standard NN training, and propose architectural and optimisation interventions that minimise their prevalence. On the architectural side, we have shown that Normalisation layers can have unwanted effects on OFs during training. Removing standard Norms through our Outlier Protected transformer block minimises OFs during training without loss of convergence speed or training stability. On the optimisation side, we highlight that large diagonal adaptive learning rates are crucial for OFs, and non-diagonal preconditioners offer an appealing combination of reduced OFs and improved convergence speed. We demonstrate our methods to minimise OFs are effective at scales of up to 7B parameters, and also directly translate to improved post-training quantisation performance. Overall, our results reveal the complex interactions between architecture and optimiser that lead to the OFs widely observed in LLMs. In future work, it would be interesting to consider applying our methods to minimise OFs for post-training quantisation performance at larger scales, and also low-precision training.

---

[11]We keep the batch size 196 & context length 512 same as [15] for fair comparison. It is likely the FP16/FP32 gains of non-diagonal preconditioners e.g. SOAP would increase with higher token counts per step [73].

## Acknowledgements

We would like to thank Sam Smith for helpful suggestions at the beginning of this project, Tiago Pimentel for constructive feedback on an early version of this manuscript, and Saleh Ashkboos for various insightful discussions around rotations and quantisation. We are also grateful for the mostly constructive feedback we received from anonymous reviewers. Finally, we would also like to thank the Swiss National Supercomputing Centre for access to GPU resources to perform some of the experiments in this work, via a grant under project ID a06 on Alps as part of the Swiss AI Initiative. We thank Alex Hägele for help setting up our experiments on FineWeb-Edu.

## Reproducibility Statement

Our code for experiments on the CodeParrot dataset can be found at `https://github.com/bobby-he/simplified_transformers`.

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

# Appendix

## Table of Contents

## A   Additional Background Knowledge

### A.1   Mathematical Description of Transformer Blocks

A Transformer architecture [37] is formed by sequentially composing Transformer blocks. The two most popular Transformer blocks are Pre-Norm and Post-Norm. The Pre-Norm Transformer block [48, 49] is nowadays more widespread than the original Post-Norm block [37] due to advantageous depth-scaling properties [74, 51].

For an input sequence representation $\mathbf{X}_{\text{in}} \in \mathbb{R}^{T \times d}$, with $T$ tokens and dimension $d$, the output $\mathbf{X}_{\text{out}}$ of a **Pre-Norm** Transformer block is:

$$\mathbf{X}_{\text{out}} = \hat{\mathbf{X}} + \text{MLP}(\text{Norm}_2(\hat{\mathbf{X}})), \quad \text{where } \hat{\mathbf{X}} = \mathbf{X}_{\text{in}} + \text{MHA}(\text{Norm}_1(\mathbf{X}_{\text{in}})). \tag{6}$$

On the other hand, the **Post-Norm** Transformer block can be expressed as:

$$\mathbf{X}_{\text{out}} = \text{Norm}_2(\hat{\mathbf{X}} + \text{MLP}(\hat{\mathbf{X}})), \quad \text{where } \hat{\mathbf{X}} = \text{Norm}_1(\mathbf{X}_{\text{in}} + \text{MHA}(\mathbf{X}_{\text{in}})). \tag{7}$$

Here, "MHA" stands for Multi-Head Attention (detailed below), and "Norm" denotes a normalisation layer like LayerNorm [44] or RMSNorm [45]. In words, we see that the Pre-Norm transformer block consists of two sequential sub-blocks (one attention and one MLP), with normalisation layers and residual connections for both sub-blocks. Crucially the Norm layers in Pre-Norm blocks are placed within the residual branch, whereas in Post-Norm blocks the Norm layers are placed after the residual connection.

When we say "Pre-LN" we mean that the block is Pre-Norm and the Norm is LayerNorm [44], and likewise "Post-RMSNorm" would mean that the block is Post-Norm and the Norm is RMSNorm, and so on in Fig 2. There is no notion of "Pre" or "Post" with our OP block because there is no difference between Pre-Norm and Post-Norm if one removes the Norms.

In our work, the MLP architecture is single hidden-layer with hidden dimension that is $4d$ (as in [37]), and acts on each token in the sequence independently.

The MHA sub-block uses softmax self-attention to share information between tokens. For a given input sequence $\mathbf{X}$, the softmax self-attention mechanism computes:

$$\text{Attn}(\mathbf{X}) = \mathbf{A}(\mathbf{X})\mathbf{X}\mathbf{W}^V, \quad \text{where } \mathbf{A}(\mathbf{X}) = \text{Softmax}\left(\frac{1}{\sqrt{d_k}}s(\mathbf{X}) + \mathbf{M}\right), \tag{8}$$

where $s(\mathbf{X}) = \mathbf{X}\mathbf{W}^Q\mathbf{W}^{K^\top}\mathbf{X}^\top$ are the pre-softmax attention scores/logits, and $\mathbf{W}^Q, \mathbf{W}^K \in \mathbb{R}^{d \times d_k}$ and $\mathbf{W}^V \in \mathbb{R}^{d \times d_v}$ are trainable query, key and value parameters respectively.

The attention matrix $\mathbf{A}(\mathbf{X}) \in \mathbb{R}^{T \times T}$ allows different tokens to "attend" to each other, with mask $\mathbf{M} \in \mathbb{R}^{T \times T}$ determining which tokens any given token is allowed to "attend" to. For causal auto-regressive transformers like GPT, $\mathbf{M}_{i,j} = 0$ if $i \geq j$ and $-\infty$ else, which prevents a token from obtaining information from future tokens.

The Multi-Head Attention name arises due to the fact that in practice it is typical to apply self-attention on $H$ different "heads" with $d_v = d_k = \frac{d}{H}$ before concatenating the heads, as follows:

$$\text{MHA}(\mathbf{X}) = \text{Concat}\big(\text{Attn}_1(\mathbf{X}), \ldots, \text{Attn}_H(\mathbf{X})\big)\mathbf{W}^P, \tag{9}$$

where $\mathbf{W}^P \in \mathbb{R}^{d \times d}$ denotes a trainable matrix that combines different attention heads via a projection.

## A.2 Background on NN optimisers

In this subsection we provide additional background on different deep learning optimisers to accompany Sec 5 for completeness.

Consider a weight matrix $W \in \mathbb{R}^{l \times r}$ with scalar loss function $L(W)$, and corresponding gradient $G = \nabla_W L \in \mathbb{R}^{l \times r}$. We denote $\eta$ as a scalar learning rate and $\epsilon$ as a scalar "damping" hyperparameter that prevents numerical instabilities. We outline the three families of optimisers that we explore at various points in our work, in terms of their effect on outlier features: 1) SGD (with momentum), 2) diagonal preconditioners, and 3) non-diagonal preconditioners.

**SGD with Momentum** keeps an (exponential) moving average of gradients $G$, $M \in \mathbb{R}^{l \times r}$, and updates $W$ as:
$$W \leftarrow W - \eta M.$$

**Diagonal preconditioners** like Adam [75] are arguably the most popular deep learning optimiser family due to their combination of improved convergence on modern architectures like transformers, compared to SGD, and computational efficiency, compared to non-diagonal preconditioners.

Adam works by maintaining not only an exponential moving average of gradients $M \in \mathbb{R}^{l \times r}$, like SGD, but also an element-wise second moment of gradients $G \odot G$, $V \in \mathbb{R}^{l \times r}$.

Then, the Adam update rule is:
$$W \leftarrow W - \eta\frac{M}{\sqrt{V} + \epsilon},$$

where all operations (e.g. division or square root) are element-wise. This element-wise nature leads us to describe Adam as *diagonal* (as it can easily be written out with matrix multiplication using diagonal matrices). Note that one can view the elements of $\frac{\eta}{\sqrt{V}+\epsilon}$ as per parameter adaptive learning rates, compared to SGD which just has a global shared LR $\eta$.

AdamW [43] is a popular variant of Adam that decouples weight decay. AdaFactor [65] is another variant of Adam, which replaces $V$ by a rank-1 approximation, $V' \in \mathbb{R}^{l \times r}$, in the interests of memory. Despite the reduced rank of $V'$, AdaFactor is similarly diagonal like Adam in the sense that there is a unique element of $V'$ for each element of $W$, and all operations are element-wise.

**Non-diagonal preconditioners** obtain their update rule by instead applying a full non-diagonal (inverse) matrix preconditioner $P \in \mathbb{R}^{lr \times lr}$ to the (flattened) first order moment $\text{flat}(M) \in \mathbb{R}^{lr}$ to give update $-\eta P^{-1}\text{flat}(M)$.

Due to the exorbitant cost of storing, updating, inverting, and applying $lr \times lr$ matrices, a popular approach introduced by Martens and Grosse [66] is to use a kronecker factored preconditioner $P = L \otimes R$, where $L \in \mathbb{R}^{l \times l}$ and $R \in \mathbb{R}^{r \times r}$ to give update rule (with dampening):

$$W \leftarrow W - \eta(L + \epsilon I_l)^{-1} M (R + \epsilon I_r)^{-1}$$

In Shampoo [67], $L$ is a moving average of $GG^\top$ and $R$ is a moving average of $G^\top G$, although different choices of exponents (e.g. $-1/4$ or $-1/2$) are more common besides $-1$. For example, the Shampoo update rule is $W \leftarrow W - \eta(L + \epsilon I_l)^{-1/2} M (R + \epsilon I_r)^{-1/2}$ for exponent $-1/2$.

As we discuss in Sec 5, Vyas et al. [69] identify a connection between Shampoo with exponent $-1/2$ and AdaFactor, and this insight gives rise to the SOAP optimiser. We refer the reader to Vyas et al. [69] for additional background on different NN optimisers and their connections.

### A.3 Additional Related Work

**Understanding Outlier Features**   Kovaleva et al. [1], Timkey and van Schijndel [2] first identified Outlier Features in trained Transformers and demonstrated that OFs are critical for representational quality and performance. Puccetti et al. [13] highlight the importance of token frequency [76] for OFs in transformers trained on language data, which is related to the representation degeneration phenomenon of Gao et al. [77], and certain "vertical" structures appearing in attention matrices during training. Bondarenko et al. [15] term this vertical structure "no-op" behaviour, where uninformative tokens are given high attention weights, and show that modifying attention to encourage no-op behaviour can mitigate OFs. Dettmers et al. [8] show that the effect of OFs is more pronounced at larger parameter scales, and Wortsman et al. [10] suggest that OFs are related to increasing activation scales during training, motivating their use of downweighted residuals. Kovaleva et al. [1], Wei et al. [7] attribute OFs to the trainable parameters in Layer Normalisation. Nrusimha et al. [12] show that OFs occur early in training, and are stronger in residual stream layers. Sun et al. [78] demonstrate the existence of "massive activations" and show they act as bias terms in transformers. Darcet et al. [79] show that outlier tokens with large activation norms lead to non-smooth attention maps in vision transformers, and propose additional "register" tokens in order to concentrate the outliers and yield smoother attention maps. Allen-Zhu and Li [80], He and Ozay [81] study a theoretical framework where sparse activations naturally appear and grow with gradient descent, owing to certain "lucky" neurons being correlated with useful features at initialisation, in order to study ensembling and knowledge distillation in two-layer convolutional NNs.

**Outlier Features and Quantisation**   Wei et al. [7], Bondarenko et al. [3] identified Outlier Features as an issue for quantised NNs. Most work in this area has focused on (weight) quantisation of already trained transformers [8, 36, 11], for efficiency gains at inference time. Dettmers et al. [8] keep outlier features in full precision to avoid their quantisation errors, while Xiao et al. [36] propose to migrate the quantisation difficulty of outlier features to their corresponding weights using some scaling factors. Chee et al. [82] introduce the idea of "incoherence processing", where pre-trained weights are rotated with random orthogonal matrices to remove outliers, which is provably and empirically shown to make quantisation easier with their method QuIP. Tseng et al. [83] extend QuIP to use random Hadamard matrices (among other changes), which are more efficient and have better theoretical properties than random orthogonal matrices. Ashkboos et al. [11] combine incoherence processing with "computational invariance" to rotate the feature vectors in addition to pre-trained weights whilst preserving the forward pass, thereby removing OFs in the rotated features and achieving state of the art performance in weight-and-activation quantisation at inference time.

In terms of quantised training, Bondarenko et al. [15] show that encouraging "no-op" behaviour can mitigate OFs and enable low-precision training, while Wortsman et al. [10] employ downweighted residuals (among other techniques) for quantised CLIP training. We discuss how our findings relate and extend these insights in the main text. Hu et al. [84] propose outlier-efficient Hopfield Layers as an alternative to traditional attention mechanisms to improve post-training quantisation. Nrusimha et al. [12] propose to regularise the kurtosis of the outputs of a linear layer for low-precision training, which the authors argue prevents migrating quantisation difficulty to the weights. We employ kurtosis to measure OFs, but focus on the kurtosis of the inputs to a linear layer.

**Normalisation Layers**   Normalisation Layers have been near ever-present in NNs since their introduction [85, 44], owing to their training benefits. Many works since have considered removing Normalisation layers, by finding alternative mechanisms that keep their benefits. De and Smith [51] identify a beneficial implicit effect of Normalisation layers in Pre-Norm Eq (6) architectures is to

downweight residual branches, and that explicit recreating this effect enables training deep NNs without Normalisation. Hayou et al. [23] show this theoretically using Signal Propagation theory, and propose downweighting residuals with a scale factor $O(1/\sqrt{\text{depth}})$ to do so, which Noci et al. [27] corroborate in the transformer setting. Martens et al. [25], Zhang et al. [86] demonstrate how to remove residual connections alongside normalisation layers in convolutional NNs using "transformed" activations, which He et al. [28] extend to the Transformer architecture by making attention more identity-like (see also "shaped" attention, Noci et al. [29]). Brock et al. [53], Smith et al. [87] propose NFNets, and achieve state of the art performance on the ImageNet benchmark in an unnormalised residual convolution architecture, highlighting that Normalisation layers are not necessary for best performance in convolutional models. NFNets employ downweighted residual branches to fix Signal Propagation at initialisation, among other techniques including adaptive gradient clipping. However, He et al. [28], He and Hofmann [30] find that removing Normalisation Layers, even with Signal Propagation modifications like downweighting residuals, leads to a loss of performance in simplified Transformer blocks, implying that transformer training has different instabilities to convolutional models, and Normalisation layers have other training benefits in transformers.

**Entropy Collapse and QK-Norm**   Zhai et al. [57] identify entropy collapse as a key training instability in transformers, where attention logits grow large during training. This causes the rows of the post-softmax attention matrix to become one-hot vectors and the attention weights are non-zero on only a single sequence position.

Mathematically, given a stochastic attention matrix $\mathbf{A}(\mathbf{X}) \in \mathbb{R}^{T \times T}$ (in the notation of Eq (8)), we can compute the entropy $\mathbf{H}(\mathbf{A})$, averaged over sequence locations, as:

$$\mathbf{H}(\mathbf{A}) = -\frac{1}{T} \sum_{s,t=1}^{T} \mathbf{A}_{s,t} \cdot \log(\mathbf{A}_{s,t}), \tag{10}$$

and define *Entropy Collapse* to be the situation where $\mathbf{H}(\mathbf{A})$ tends to 0 during training. This occurs when the rows $\mathbf{A}_{s,:}$ become one-hot vectors and token $s$ only attends to one other token, for all $s$. We treat $0 \cdot \log(0)$ to be 0.

To remedy entropy collapse, it is important to control the logits entering softmax from growing too large, and Zhai et al. [57] propose $\sigma$Reparam which regularises the spectrum of Query-Key weights in order to do so.

As an alternative, *Query-Key Normalisation* [88], where the Queries and Keys are normalised using e.g. LayerNorm or RMSNorm after the Query/Key weight matrix has seen growing popularity, particularly in ViT-22B [59] where it was crucial for stable training.

Mathematically, instead of standard attention logits $s(\mathbf{X}) = \mathbf{X}\mathbf{W}^Q(\mathbf{X}\mathbf{W}^K)^\top$ (following the notation of Eq (8)), QK-Norm first normalises the queries and keys across the $d_k$ dimension:

$$s(\mathbf{X}) = \text{Norm}(\mathbf{X}\mathbf{W}^Q)\text{Norm}(\mathbf{X}\mathbf{W}^K)^\top, \tag{11}$$

and is most commonly used as an addition on top of the Pre-Norm block Eq (6) (without removing other normalisation layers like in our OP block).

Other "entropy regulating" mechanisms include tanh thresholding (Grok-1) and clamping attention logits (DBRX). The training stability benefits of controlling attention entropy through QK-Norm were shown at smaller scales in Wortsman et al. [10], who argue that the quadratic dependence in the attention logits (on the queries and keys), causes large attention logits to appear during training, hence entropy collapse. This is as opposed to convolutional/MLP models which depend linearly on their inputs. Tian et al. [89] propose joint MLP/Attention dynamics to predict attention entropy during training. We note that the "vertical" or "no-op" attention structures discussed in previous OF works [13, 15] have collapsed attention entropy, and can be thus be seen as undesirable from the perspective of other existing works.

**Signal Propagation**   Signal Propagation studies how different inputs evolve through a deep NN, and how their feature representation magnitudes and cosine similarities evolve with depth. Our work focuses on forward signal propagation (which studies the forward-pass activations), as opposed to backward signal propagation (which studies the backward-pass activation derivatives), in line with the study of Outlier Features.

For an input activation matrix $\mathbf{X} \in \mathbb{R}^{n \times d}$ of $n$ inputs and width $d$, mapped to an activation matrix $\mathbf{X}_l \in \mathbb{R}^{n \times d}$ at layer $l$, signal propagation theory studies the evolution of the input-wise Gram matrix $\Sigma_I^l = \mathbf{X}_l \mathbf{X}_l^\top \in \mathbb{R}^{n \times n}$ for increasing depths $l$. This is a key object in an NN, as it tracks the "geometric information" that is conveyed in a deep layer, through inner products between different inputs. The diagonal elements of $\Sigma_I^l$ indicate the activation norms, and the off-diagonal elements indicates how similar a deep layer views two inputs to be.

At initialisation, $\Sigma_I^l$ can be tracked through its large $d$ limits [20, 19, 21]. By studying $\Sigma_I^l$, one can see several issues that will afflict badly designed NNs [18, 56, 90, 60, 25], that affect either the diagonal elements, the off-diagonal elements or both at large depths. For example, the diagonal elements of $\Sigma_I$ could blow up, which indicates exploding activation norms. For transformers, a particular degeneracy, known as rank collapse [60], can appear where the off-diagonals of $\Sigma_I^l$ become positive and large, and $\Sigma_I^l$ becomes proportional to the all ones matrix if activation norms are constant. Rank collapse is also possible in MLPs/CNNs Schoenholz et al. [18], Hayou et al. [56], Xiao et al. [91], Martens et al. [25], and is equivalent to the over-smoothing phenomenon in graph NNs [92]. Martens et al. [25] argue that rank collapse will lead to vanishing gradients, which Noci et al. [27] show specifically for query and key parameters in transformers. As a result, when we refer to bad signal propagation, we mean that the off-diagonals of $\Sigma_I$ are large and positive, close to rank collapse. This can be either through the RMS of input correlations, $\sqrt{\frac{1}{n(n-1)} \sum_{\alpha \neq \beta}^n \left(\frac{1}{d}\Sigma_I\right)_{\alpha,\beta}^2}$, as we show in the appendix, or the mean, $\frac{1}{n(n-1)} \sum_{\alpha \neq \beta}^n \left(\frac{1}{d}\Sigma_I\right)_{\alpha,\beta}$ as we show in Figs 5 and 10.

By applying Signal Propagation theory at initialisation, it is possible to design modifications to NN architectures and initialisations that correct potential degeneracies and/or yield simpler and/or more scalable architectures [93, 23, 25, 86, 27, 28, 30]. But the vast majority of existing works in the literature do not theoretically study training beyond initialisation, and those that do are usually restricted to the NTK [94] regime [23, 25], which precludes feature learning, and OFs. Lou et al. [95] suggest that the feature alignment [96] phenomenon during training is correlated to the rate at which signal propagation converges to its limit in a deep NN. Even at initialization, the distribution of the neurons becomes more heavy-tailed with depth [97], thus making outliers more likely. Noci et al. [24] gives a precise description of the kurtosis for ReLU networks, showing that it grows exponentially with depth. Together with the results presented in this work, there is empirical and theoretical evidence that depth has the double effect of increasing both the correlations and making large activations more likely, which we observe to be detrimental to outliers. However, the theoretical treatment of the emergence of outliers during training is still an open question.

## B Mathematical description of OP block

For completeness, in Eqs (12) and (13) we present a mathematical description of the OP block (Fig 3), based on the notation in App A.1. Recall the OP block replaces the Pre-Norm block by removing its normalisation layers in both Attention and MLP sub-blocks, and making three additional changes: 1) downweighting residual branches with some $\beta_{\text{MLP}}, \beta_{\text{Attn}} = O(1/\sqrt{\text{depth}}) < 1$ to recover Signal Prop benefits of Pre-Norms [51, 23, 27, 30], 2) adding an Entropy Regulation mechanism to prevent Entropy Collapse e.g. QK-Norm or tanh-softcapping, and 3) (optionally) scaling the inputs before the MLP nonlinearity by a scalar $\alpha_{\text{MLP}}$ to ensure the nonlinearity inputs are of order 1, following Brock et al. [53].

Mathematically, this can be expressed as:

$$\mathbf{X}_{\text{out}} = \hat{\mathbf{X}} + \beta_{\text{MLP}}\text{MLP}(\alpha_{\text{MLP}}\hat{\mathbf{X}}), \quad \text{where } \hat{\mathbf{X}} = \mathbf{X}_{\text{in}} + \beta_{\text{Attn}}\text{MHA}(\mathbf{X}_{\text{in}}). \tag{12}$$

With QK-Norm as entropy regulation mechanism, MHA is multihead attention using QK-Norm in the attention heads:

$$\text{Attn}(\mathbf{X}) = \mathbf{A}(\mathbf{X})\mathbf{X}\mathbf{W}^V, \quad \text{where } \mathbf{A}(\mathbf{X}) = \text{Softmax}\left(\frac{1}{\sqrt{d_k}}\text{Norm}(\mathbf{X}\mathbf{W}^Q)\text{Norm}(\mathbf{X}\mathbf{W}^K)^\top + \mathbf{M}\right),$$
$$\tag{13}$$

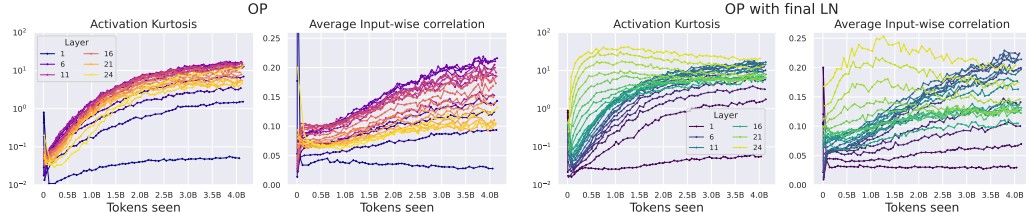

Figure 10: OP layers at 1.2B scale with worse Signal Propagation (i.e. higher input correlations) during training (centre left) have higher feature kurtosis (left). (**Right vs. left two plots**) Introducing a final LN before unembedding causes larger input correlations and feature kurtosis in later layers, even with the OP block. **NB**: y-axes values here are significantly smaller than Fig 5 with Pre-LN.

## C  Modifications That Affect Signal Prop During Training Affect OFE

In this section we continue our discussion of the connection between Signal Propagation and OFE in Sec 4, towards the practical implication that methods designed to improve Signal Propagation also improve OFE as a result.

To the best of our knowledge, in the Signal Propagation literature, most works have focused on characterising and improving Signal Propagation *at initialisation* due to analytic convenience. At initialisation, only the architecture plays a role and not the optimiser. In particular, a practical focus of such works is to design architectural modifications that allow non-degenerate deep limits for models whose input cosine similarities can be well approximated by their large-width limits at initialisation [98, 93, 23, 25, 28, 30]. Those considering training dynamics often reside in the kernel regime [94] and are thus not compatible with feature learning [31, 33] which is necessary for OFE and Signal Prop dynamics during training. Our results connecting Signal Prop and OFE highlight the importance to the community of understanding Signal Prop dynamics during training in feature learning regimes, beyond initialisation. We note Tian et al. [89] predict attention entropy dynamics through joint MLP/Attention. In any case, we empirically study the impact of initialisation-inspired Signal Prop architectural modifications in terms of OFE during training.

**Downweighted residuals**  Of initialisation-inspired Signal Prop modifications, the most prevalent is downweighting residual branches $h(x) = x + \beta f(x)$ with some $\beta \ll 1$ [51, 23, 27].[12] In Fig 5, we see that downweighting residuals (with a trainable scalar $\beta$ initialised to 0.1) improves Signal Propagation in a 24-block 1.2B Pre-LN model, not only at initialisation but also *during training*, thereby reducing OFE (peak kurtosis is an order of magnitude lower). Having said that, Pre-LN with downscaled residuals still leads to higher kurtosis across layers than our OP block in Fig 10. Downscaling Pre-LN residuals leads to a small loss in performance of 0.2 perplexity. We show the corresponding results at 130M scale in Figs 19 to 21. Our results are consistent with previous work by Wortsman et al. [10] who observe that downweighted residuals help for low precision training in CLIP models, motivated as a way to prevent OFs arising through increasing activations scales $\|\mathbf{X}\|_F$ during training. Given that standard models have Norm layers that are scale invariant (as are our OFE and Signal Prop metrics), we complement this argument by highlighting that the feature learning process of OFE is not only associated with increasing activation scales but also worsening Signal Propagation during training. Figs 16 and 48 show that $\|\mathbf{X}\|_F$ does not always correlate with OFs.

**Normalisation layers**  On the other hand, for Norms, the difference between OP and standard blocks with Norms in Figs 5, 10 and 18 respectively is already clear evidence that standard Norm placements can lead to worse Signal Propagation (and OFE) during training. To the best of our knowledge, this observation has not been made previously. To test this further, we reintroduce the final LN right after the final OP block (just before the unembedding layer) into an OP model, with no Pre-Norms, in Fig 10. We see that the final LN causes some layers to see increases in both kurtosis and input correlations, and these layers correspond *precisely* to the final few blocks immediately preceding the LN. On the other hand, earlier layers further away from the final LN are largely unchanged in terms of both Signal Propagation and OFE during training. The model with a final LN performed slightly worse (0.1 perplexity difference).

---

[12]Typically, the theory indicates that $\beta = O(\frac{1}{\sqrt{\text{depth}}})$ enables a well-behaved infinite-depth limit.

Several works have discussed the effect of Norms on Signal Propagation theory at initialisation. The Deep Kernel Shaping [25] framework is compatible with LN (and also RMSNorm) layers, but makes other modifications (in weight initialisation and activation functions) that mean LN has no effect at initialisation in the wide limit. Other works show centred Norms in fact improve Signal Propagation at initialisation in MLPs by correcting imbalances in input activation norms to improve *Isometry* [99, 100] but consider non-standard architectures that are not residual and have Norm immediately following nonlinear activations, whereas standard Norms take the residual stream as input. Our work shows that initialisation and training can have very different Signal Prop behaviours.

**Other Signal Prop modifications** In Figs 22 and 24, we consider the effect of other initialisation-inspired Signal Propagation modifications in terms of OFE. In particular, we consider "transforming" activations to be more linear [25, 86, 26], and "shaping" attention to be more identity-like [28–30]. Although not predicted by initialisation theory, we find that these modifications mostly also reduce OFE and improve Signal Prop during training as well as initialisation. The latter finding is related to the work of Bondarenko et al. [15] who show that "no-op" heads that place large attention weights on *shared* uninformative tokens encourage OFs: large attention weights on shared tokens also worsen signal propagation,[13] compared to identity-dominated attention, which can be seen as a "no-op" that instead encourages a token to attend to itself.

> **App C key takeaways: Signal Propagation and OFE.**
>
> - Signal Propagation is fundamentally connected to OFE: worse Signal Prop generally implies higher kurtosis and vice versa, throughout training (Eq (5), Prop G.1, and Figs 5, 10 and 18).
> - The OP block's mild OFs can be traced to increasing input correlations while training (Fig 10).
> - Choices that improve Signal Prop during training (e.g. scaled residuals) also reduce OFs (Fig 5).
> - Removing standard Norms can improve Signal Prop, & OFE, during training (Figs 5 and 10).

# D  Additional Experimental Details

**CodeParrot** As discussed, all of our experiments at 130M scale (6 layers, width 768 transformers) are on next token prediction with the CodeParrot dataset, with 50K vocabulary size. We use a similar setup to He and Hofmann [30], and have updated their codebase to include the OP block.[14] We train with AdamW optimiser [43] and weight decay 0.1, $(\beta_1, \beta_2) = (0.9, 0.999)$, and $\epsilon = 1e - 8$ unless otherwise stated, and clip the maximum gradient norm to 1. We do not tie embeddings, and remove the final layer norm before unembedding layer. When we plot metrics (kurtosis, signal propagation, MMR etc) we plot the residual stream entering the attention sub-block (plots for the residual stream before the MLP sub-block are qualitatively the same). The only exception is the last layer, which is the input to the unembeddings. When we downweight residuals we set $\beta = 0.3$ in both attention and MLP sub-blocks unless otherwise stated. We do not train residual scalings $\beta$. Unless otherwise stated, we train with sequence length 128 and batch size 32 for 80K steps, with linear warmup to maximum learning rate $1e - 3$, for 5% of the steps, before linear decay. We keep the standard parameter initialisations to $\mathcal{N}(0, \text{std} = 0.02)$ but upweight the input embeddings by a factor of 50 in order to make the average squared input 1 at initialisation, similar to considerations made by the Gemma model [101]. We use ReLU activations and do not scale inputs with an $\alpha$, c.f. Fig 3, because ReLU is 1-homogeneous.

**Languini** For Languini [39] our 100M, 320M, and 1.2B model sizes follow the "small" (depth 12, width 768), "medium" (depth 24, width 1024), and "XL" (depth 24, width 2048) model sizes provided by the authors, respectively. Our setup follows the authors in terms of codebase and tokeniser. We train with sequence length 512 and batch size 128, again with a maximum learning rate of $1e - 3$ unless otherwise stated. This learning rate was the largest stable and best performing choice on a logarithmic grid. We train with AdamW, using weight decay of 0.1 and clip the maximum gradient norm to 0.5. We use linear warmup and linear decay after 1000 steps. We additionally use RoPE [102], with GeLU nonlinearities in the MLPs. We use the same method as Brock et al. [53] to

---

[13]Consider the extreme case when all attention weights are placed onto a single token (say the first one): all attention outputs will be equal to the first token's value representation so all token-wise cosine similarities are 1.

[14]`https://github.com/bobby-he/simplified_transformers`

calculate $\alpha$ to scale inputs to the GeLU. When we downweight residuals, we initialise $\beta = 0.1$ and allow them to be trainable. When we plot layer-wise metrics like kurtosis, we plot the outputs of the Pre-Normalisation layer (if there is one), otherwise, we treat the Normalisation layer as the identity and plot the residual stream going into the attention sub-block. We use tied embeddings. We also keep the standard parameter initialisations to $\mathcal{N}(0, \text{std} = 0.02)$ but upweight the input embeddings by a factor of 50 in order to make the average squared input 1 at initialisation.

**CIFAR-10** For our MLP experiments on CIFAR-10, we train using batch size 2048 for 200 epochs. As described in Sec 5, the model has 6 Pre-Norm layers with width 1024, giving 15M parameters. We zero initialise the last layer, and additionally downweight the output layer by $\sqrt{\text{width}}$ akin to $\mu$P [33], to encourage feature learning. We train with MSE loss and use LR 3 for SGD and 3e-3 for Adam. We use standard betas and epsilon for Adam and we do not use weight decay. We warm up the LR for 200 steps before cosine decay. We additionally found that it was important to whiten the inputs in order to observe OFE in the residual stream. We note that transformer embeddings are independently initialised, which can be thought of as implicitly whitening the embeddings for different tokens. Whitened inputs correspond to signal propagation with zero input correlations. This again suggests that signal propagation (and properties of the data) are important for OFs, but we leave further understanding of this to future work. We use PCA to whiten inputs.

**Non-diagonal preconditioners** For our non-diagonal preconditioning experiments, we take the implementation provided by SOAP [69].[15] We keep the default hyperparameters $\beta_1 = 0.95, \beta_2 = 0.95$ and preconditioning frequency to 10, as suggested by the authors. As in our other experiments, we use weight decay 0.1. It is likely that further hyperparameter tuning would further improve our results with non-diagonal preconditioners.

For our implementation of "AdaFactor in Shampoo's eigenbasis" (shown by [69] to be akin to Shampoo [67]), we do not rotate the input embedding dimension, which has (vocabulary) size 50k in our CodeParrot setup, but continue to store rank-1 moving averages for the squared gradient in that dimension, which are applied to the unprojected parameters in that dimension, as in AdaFactor. For both AdaFactor and its rotated counterpart, we do not use the parameter-norm dependent learning rate schedule proposed by [65], instead keeping the same update rule as [103, 104, 69] which keeps AdaFactor closest to a rank-1 approximation of Adam.

For Fig 8, we set a larger batch size of 4096 in order to demonstrate more clearly the convergence speed per step benefits of SOAP/Shampoo [73], as seen in Fig 32. We additionally use the Warmup-Stable-Decay (WSD) scheduler [105, 106] with LR=0.001 (we found this also to be the maximal trainable LR for SOAP/Shampoo in our setting), 5% warmup and 20% decay steps. WSD maintains a higher LR for longer and increases kurtosis in OF-prone settings like Adam/AdaFactor as a result, as expected from e.g. Fig 6.

**FineWeb-Edu** Our longer 1.2B and 7B experiments (Fig 9) using FineWed-Edu [41] were conducted on a fork of nanotron.[16] The model architectures are based off LLaMa [47] but we use single hidden layer GeLU MLPs instead of SwiGLU, as SwiGLU MLPs are themselves a source of OFs [107]. Our 7B model has hidden width 4096 and 32 layers, and our 1.2B model has hidden width 2048 and 24 layers.

For our 6B token 7B model runs, we train for 20K steps with global batch size 72 (node batch size 6 on 12 nodes of 4×GH200 GPUs, with tensor parallelism) and context length 4096. This gives $72 \times 4096 \times 20000 \approx 6B$ tokens. For our 90B token 1B model runs, we train for 50K steps with global batch size 440 (device batch size 5 on 22 nodes of 4×GH200 GPUs) and context length 4096 to give $440 \times 4096 \times 50000 = 90B$ tokens. We add LayerScale [55] to implement the downweighted residuals in OP block, with trainable residual gain vectors initialised to 0.1 and 0.05 for our 1.2B and 8B experiments respectively. We use RMSNorm for the QK-Norm in the OP block. Like in our Languini experiments, we upweight the input embeddings by a factor of 50 in order to make the average squared input 1 at initialisation.

**Quantisation** As discussed, our quantisation experimental setup closely follows Bondarenko et al. [15] for fair comparison, training OPT models [6] on BookCorpus and English Wikipedia before post

---

[15]https://github.com/nikhilvyas/SOAP/tree/main
[16]https://github.com/huggingface/nanotron

training quantisation. We also use their excellent codebase.[17] The hyperparameters that we change are outlined in Tab 2. Other hyperparameter are set as defaults e.g. $(\beta_1, \beta_2) = (0.9, 0.95)$, 0.1 weight decay, or ReLU activation function, as in Zhang et al. [6].

In Tab 2, as one reads down the rows for a given architecture, the optimisation hyperparameters are changed one after another on top of each other. The only exception is SOAP, where we keep AdamW's $\epsilon = 10^{-8}$ (recall SOAP is just Adam in the eigenbasis of Shampoo's preconditioner, so $\epsilon$ is still used). The model architecture is OPT [6] and have 125m parameters with width 768 and 12 layers. We initialise the trainable scalar residual gains to 0.2 in the OP block and use LN for the QK-Norm. Like [15], we use symmetric quantisation for the weights and asymmetric quantisation for the activations to int8, but also do not quantisation the final unembedding layer. We refer the reader to Bondarenko et al. [15] for more details on the quantisation procedure.

# E    Additional Experiments

In this section, we include all additional experiments not included in the main paper.

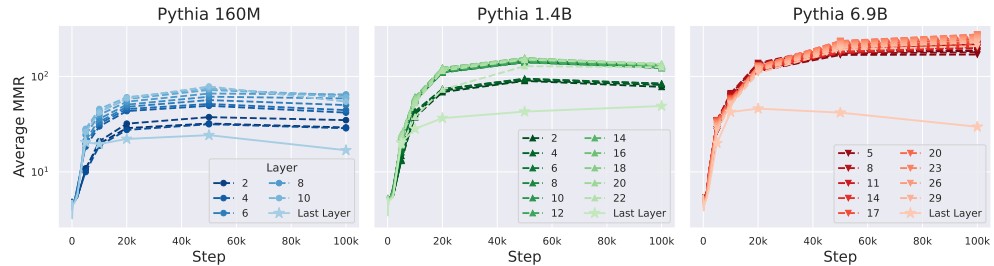

Figure 11: Max Median Ratio metric for Pythia, equivalent to Fig 1. We take the mean to aggregate over inputs

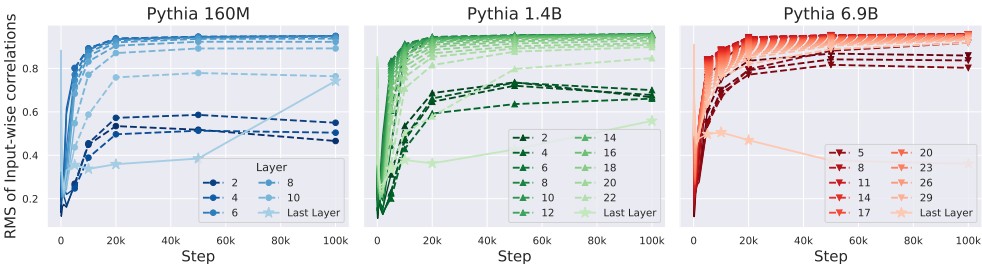

Figure 12: Signal Prop for Pythia, equivalent to Fig 1.

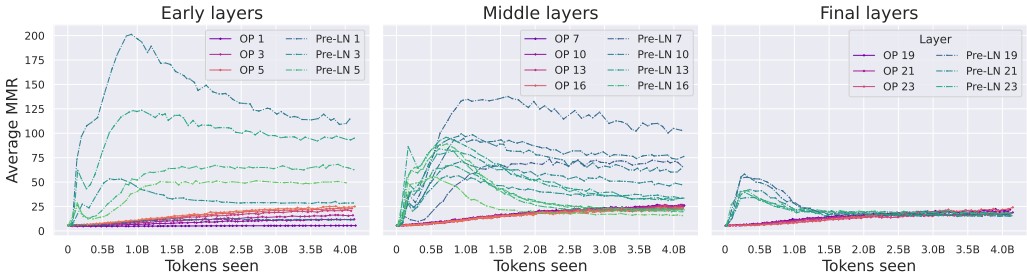

Figure 13: Average MMR metric comparing Pre-LN and OP blocks at 1.2B scale on the Languini Books dataset [39], equivalent to Fig 4.

---

[17]https://github.com/qualcomm-ai-research/outlier-free-transformers

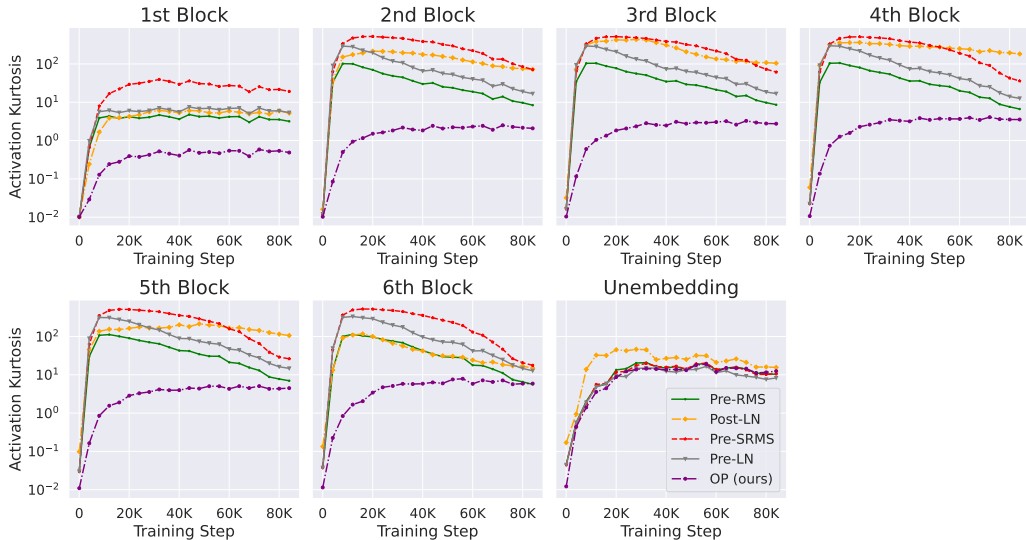

Figure 14: Kurtosis dynamics in different layers using different Norms and Norm locations on CodeParrot at 130M scale. Equivalent of Fig 2 but for the remaining layers. Fig 2 corresponds to the 2nd block.

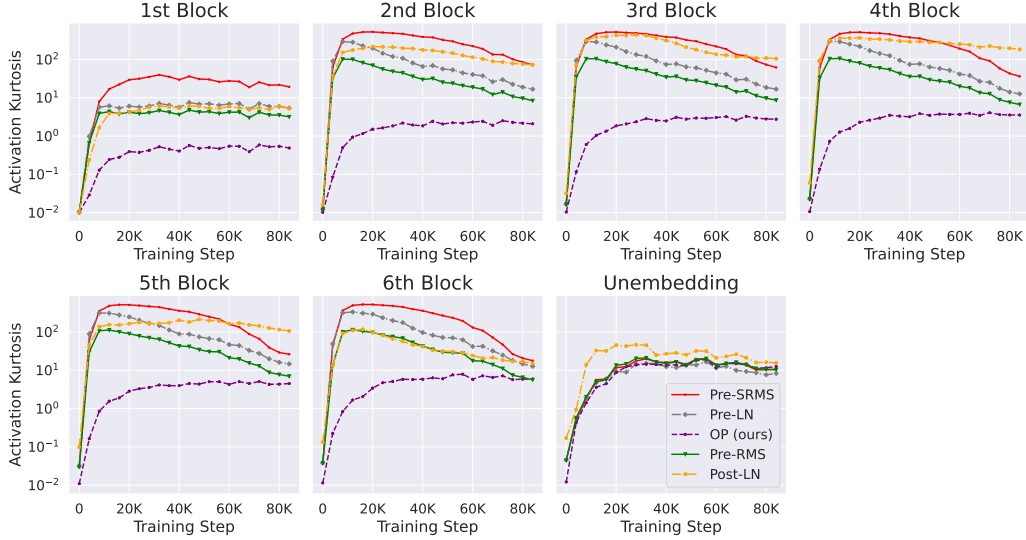

Figure 15: Equivalent of Fig 14 but with **centred activations** (centred along the width dimension). Notice there is no qualitative difference to kurtosis dynamics when centring activations.

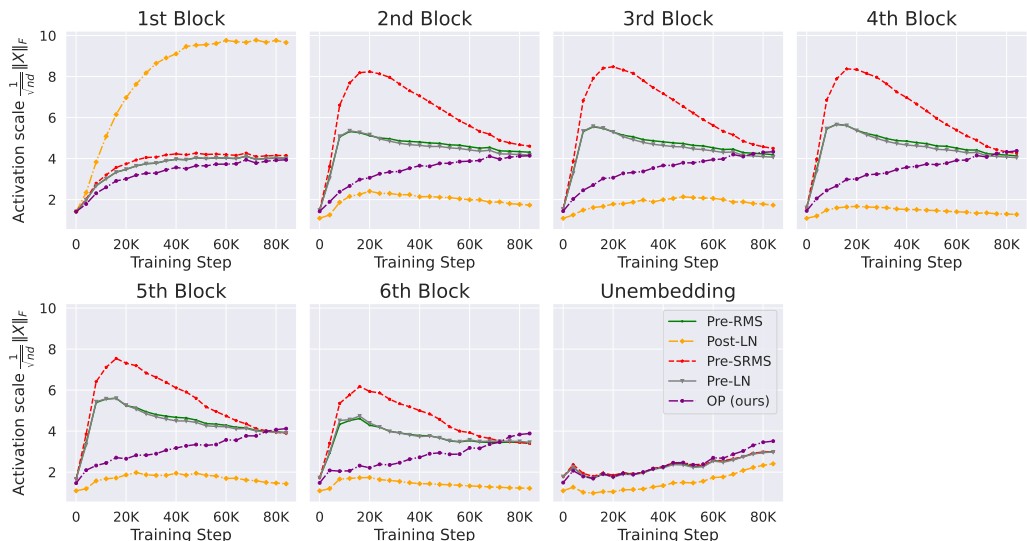

Figure 16: Equivalent of Fig 14 but for activation scale $\|\mathbf{X}\|_F$ trajectories through training. We see that activation scales do not correlate as well with OFs (Fig 14) as signal propagation (Fig 18). For example, Post-LN has smaller activation scales than the OP block in all blocks besides the first one, but much worse kurtosis in Fig 14.

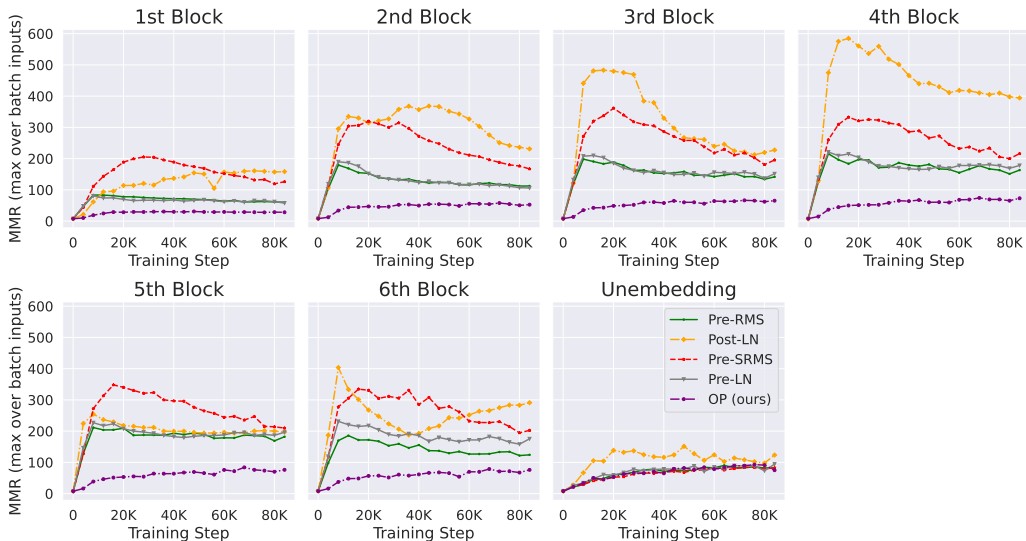

Figure 17: Equivalent of Fig 14 but for the MMR metric (aggregated using maximum over the batch).

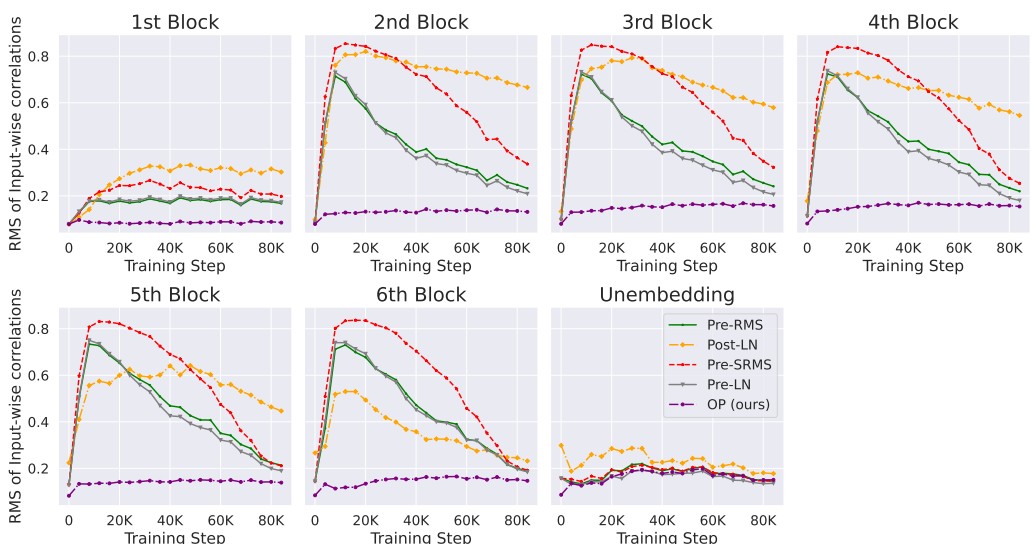

Figure 18: Equivalent of Fig 14 but for Signal Propagation (in terms of RMS of input correlations).

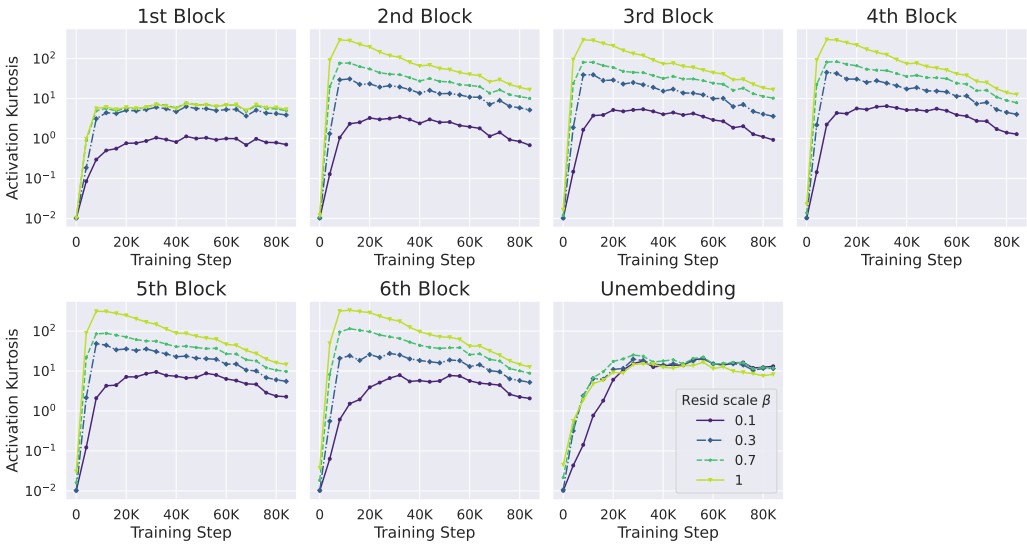

Figure 19: Downweighted residual scalings, $h(x) = x + \beta f(x)$ with $\beta < 1$, reduce OFs at 130M scale on CodeParrot. All models are Pre-LN. We downweight both the MLP and Attention residuals.

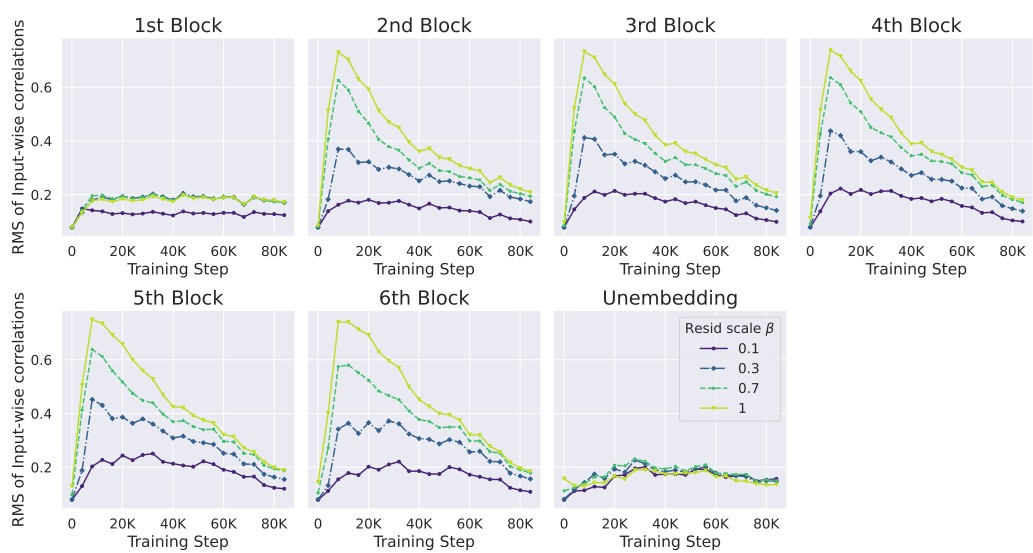

Figure 20: Residual scalings improve Signal Prop at 130M scale. Equivalent to Fig 19.

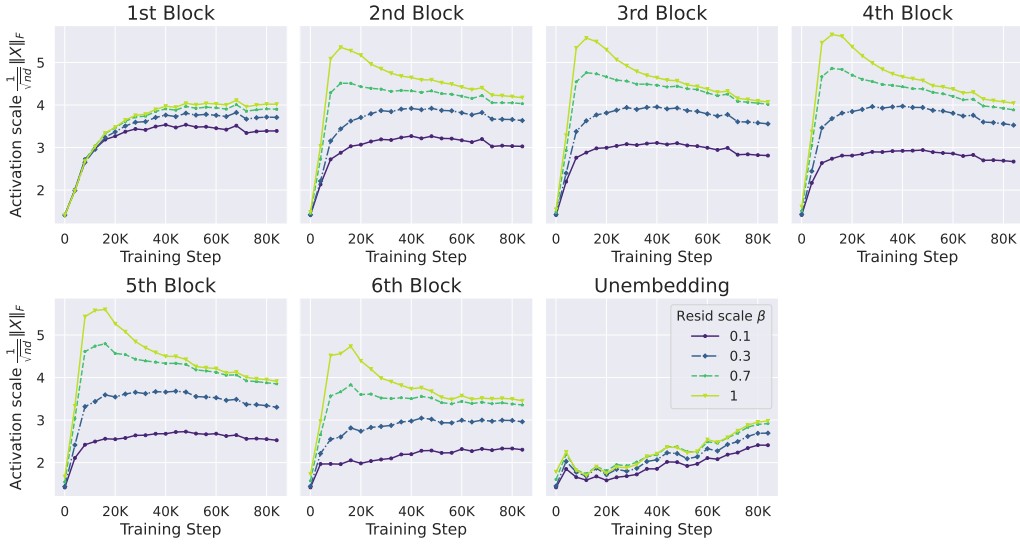

Figure 21: Residual scalings reduce activation scales at 130M scale. Equivalent to Fig 19.

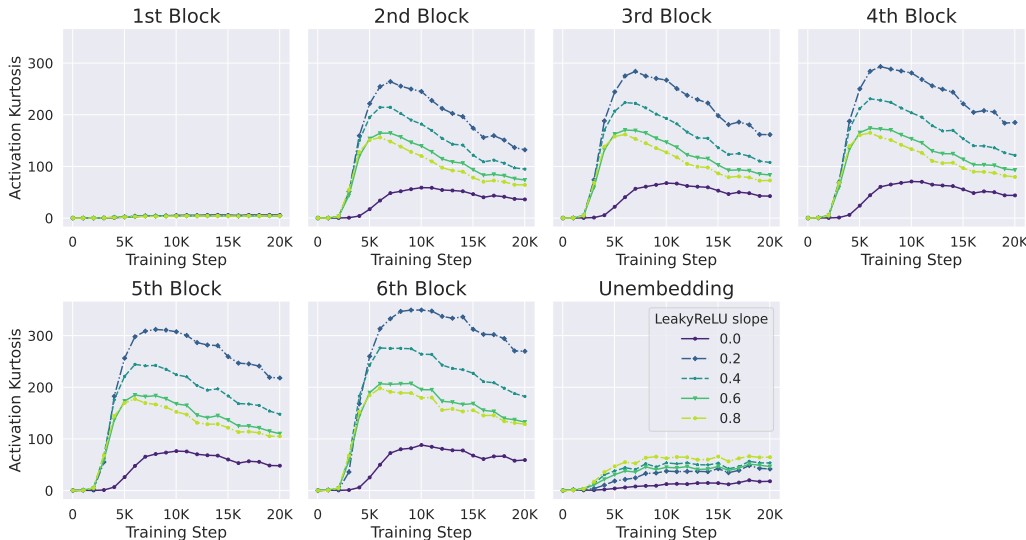

Figure 22: Increasing LeakyReLU slope, $s$, so that the nonlinearity is more linear mostly improves kurtosis during training, as one might expect from Signal Prop initialisation theory [86, 26]. Here our notation is $\text{LeakyReLU}(x) = \max(x, sx)$ for slope $s < 1$. The exception is when the slope is 0, i.e. ReLU, the kurtosis is actually better during training, but this is reflected in the signal propagation during training too (Fig 23). We hypothesise this is because zero neurons get no gradient with ReLU, and this behaves fundamentally differently to a non-zero LeakyReLU slope. The plots show the average over 5 seeds, and we plot the first 20K steps (of 80K). The models are Pre-LN and we downweight the attention residual branch with a factor $\beta = 0.2$ to reduce kurtosis contributions from the attention sub-block, but do not downweight the MLP residual. Note we do not use a log-scaled y-axis to make the differences between LeakyReLU slopes clearer. Experiment is at 130M scale on CodeParrot.

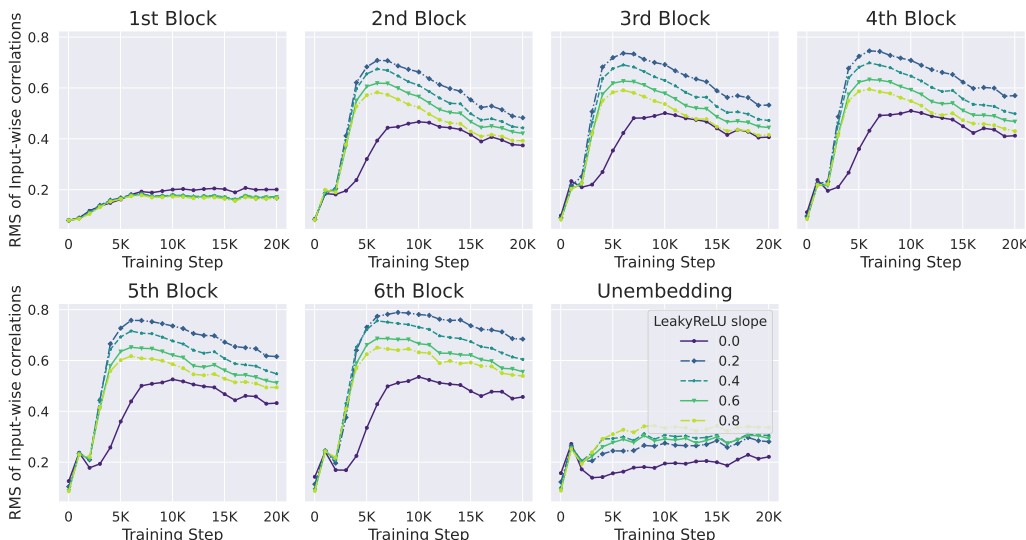

Figure 23: Effect of different LeakyReLU slopes on signal propagation during training, equivalent to Fig 22. Surprisingly, ReLU (i.e. slope 0) has the best signal propagation (lowest input-wise correlations) during training in this setting, even though it has the worst signal prop at initialisation in later layers, compared to all other LeakyReLU variants. This initialisation effect was predicted by Zhang et al. [86], but our findings regarding training were previously unknown and require further research.

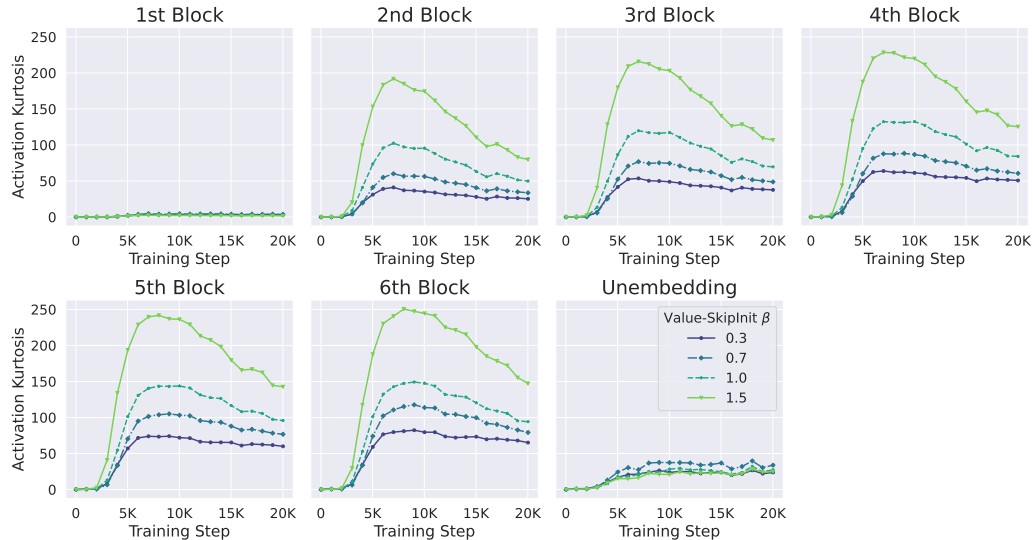

Figure 24: Reducing $\beta$ in Value-SkipInit [28], which replaces Attention matrix $\mathbf{A} \leftarrow \alpha\mathbf{I} + \beta\mathbf{A}$ and makes attention more identity-like also reduces OFs. We do not train $\beta$ in Value-SkipInit and fix $\alpha = 1$. The models are Pre-LN and we downweight the MLP residual branch with a factor $0.2$ to reduce kurtosis contributions from the MLP sub-block, but do not downweight the attention residual. Each curve is an average over 5 seeds and we plot only the first 20K steps (of 80K). Experiment is at 130M scale on CodeParrot.

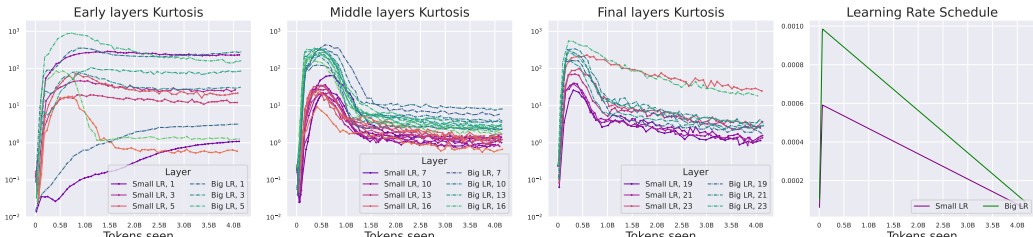

Figure 25: Smaller LR (max value from $0.001 \rightarrow 0.0006$) reduces OFE in a Pre-LN model at 1.2B scale on Languini [39]. Models are slightly different from the Pre-LN model in Fig 4 as we do not upweight the input embeddings as described in App D. Still, we do also observe large increases in kurtosis during training, and that a smaller LR reduces this. In this experiment, reducing the max LR to 0.0006 did not impact convergence speed.

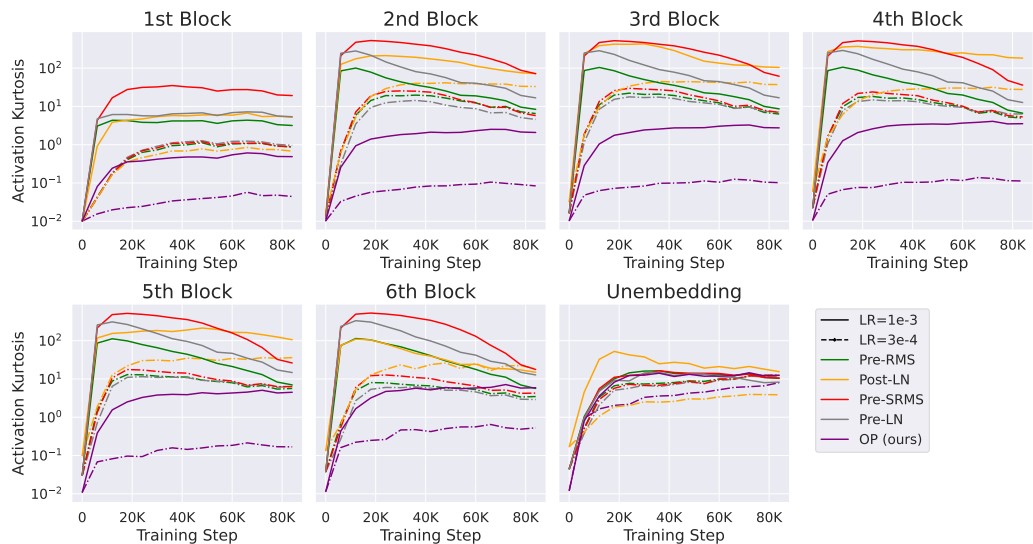

Figure 26: Smaller LRs means reduced OFs, for different Norms and Norm locations. Equivalent of Fig 6, but with all layers. Experiment is on CodeParrot at 130M scale.

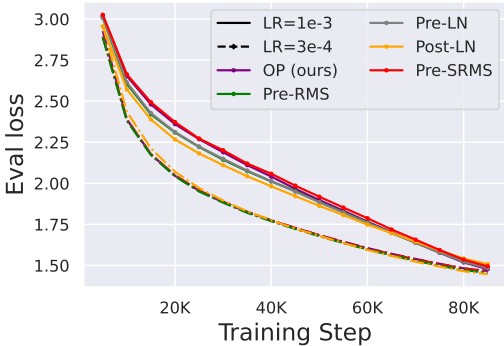

Figure 27: Convergence speed for the runs in Figs 6 and 26 comparing the effect of reduced LRs.

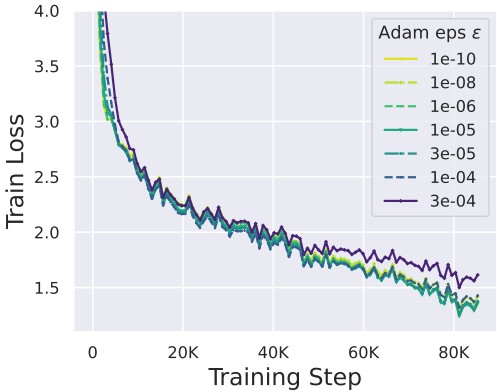

Figure 28: Train loss plot with different Adam epsilon, equivalent to Fig 29. There is not a noticeable difference in convergence speed for $\epsilon < 3e - 4$ in this experiment.

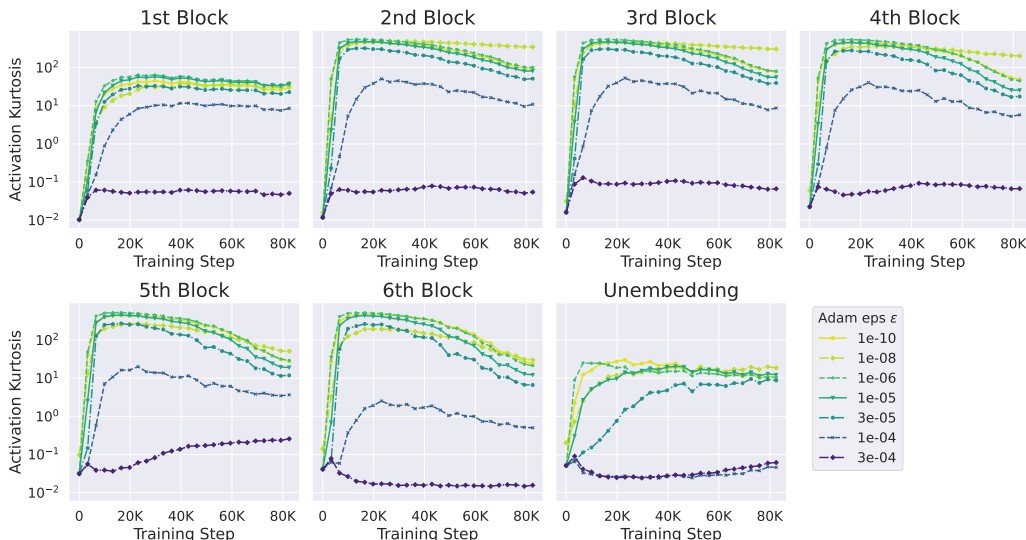

Figure 29: Kurtosis plot with different Adam epsilons on CodeParrot at 130M scale. Each curve is an average over 3 seeds. We see that increasing $\epsilon$ from $1e-6$ to $3e-4$ monotonically decreases OFE. At values of $\epsilon$ smaller than $1e-6$ there is less of a difference in OFE between different $\epsilon$ values.

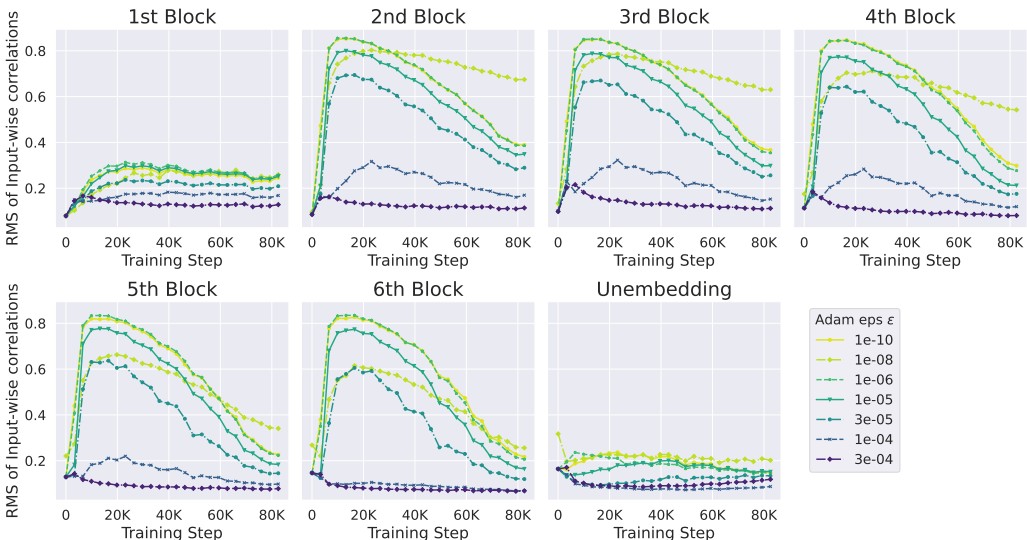

Figure 30: Signal Prop plot with different Adam epsilon. Equivalent of Fig 29.

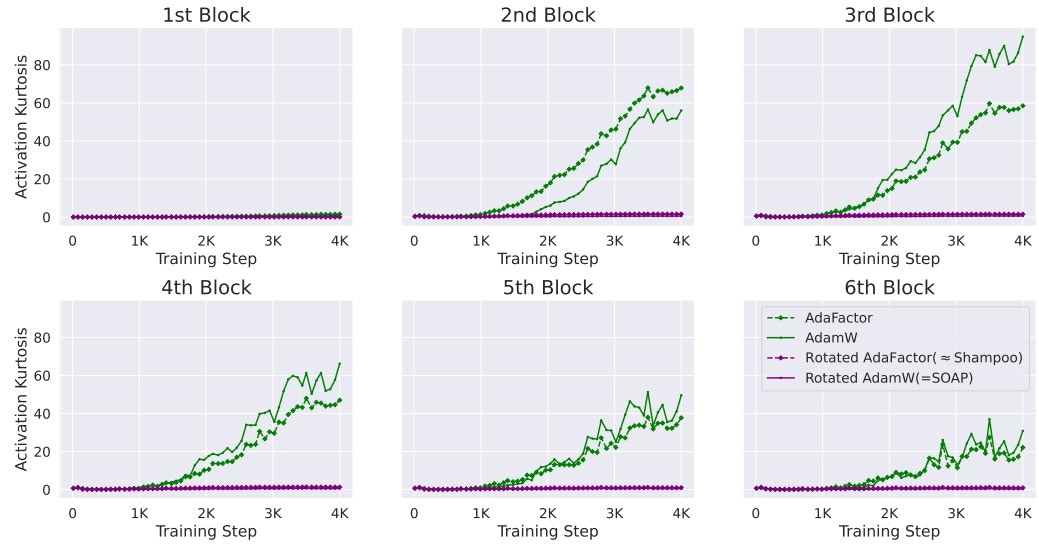

Figure 31: Kurtosis evolution in all 6 layers of a Pre-SRMSNorm transformer trained on CodeParrot with different optimisers. Equivalent to Fig 8.

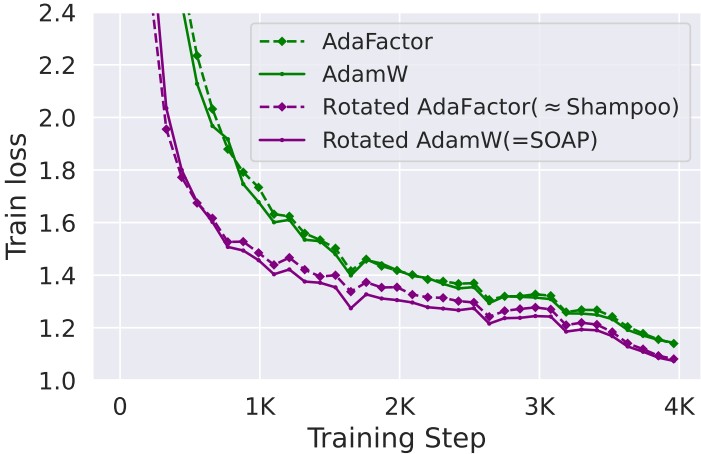

Figure 32: CodeParrot training loss curves for experiments comparing diagonal and non-diagonal preconditioners. Equivalent to the runs found in Fig 8.

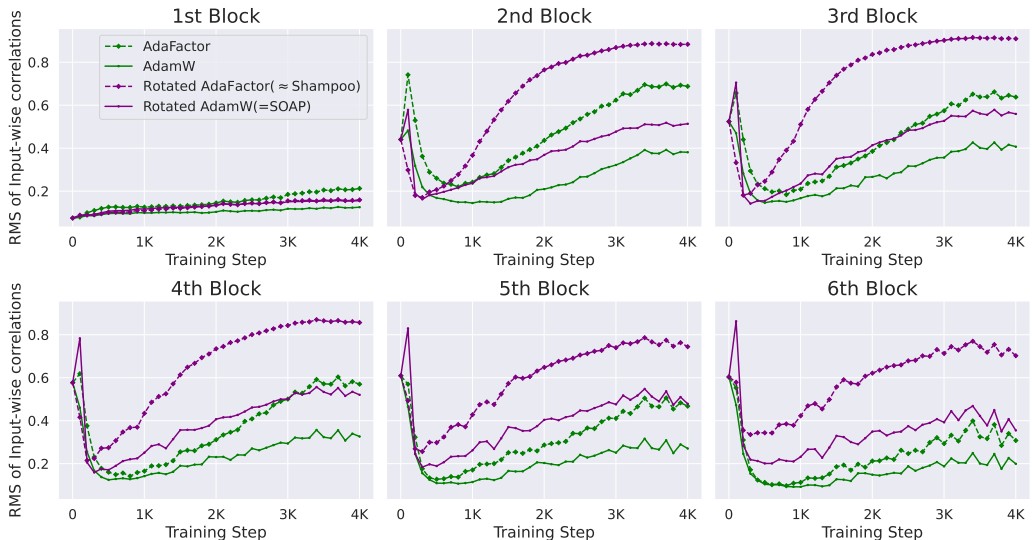

Figure 33: RMS of input-wise correlation (i.e. signal propagation, or equivalently the RHS of Eq (5)) trajectories in all 6 layers of a Pre-SRMSNorm transformer trained on CodeParrot with different optimisers. These optimisers are the diagonal Adam and AdaFactor, and their non-diagonal rotated versions (SOAP and something close to Shampoo, c.f. [69]), like in Sec 5. Notice that the connection between signal propagation and kurtosis we identify in Sec 4 is not apparent with non-diagonal preconditioners. In other words, the non-diagonal rotated versions of Adam/AdaFactor have higher input-wise correlations than diagonal Adam/AdaFactor, despite having better kurtosis properties (seen in Fig 31). This observation is explained by Fig 34, which looks at the corresponding feature-wise correlations. These training runs are equivalent to the runs found in Fig 8.

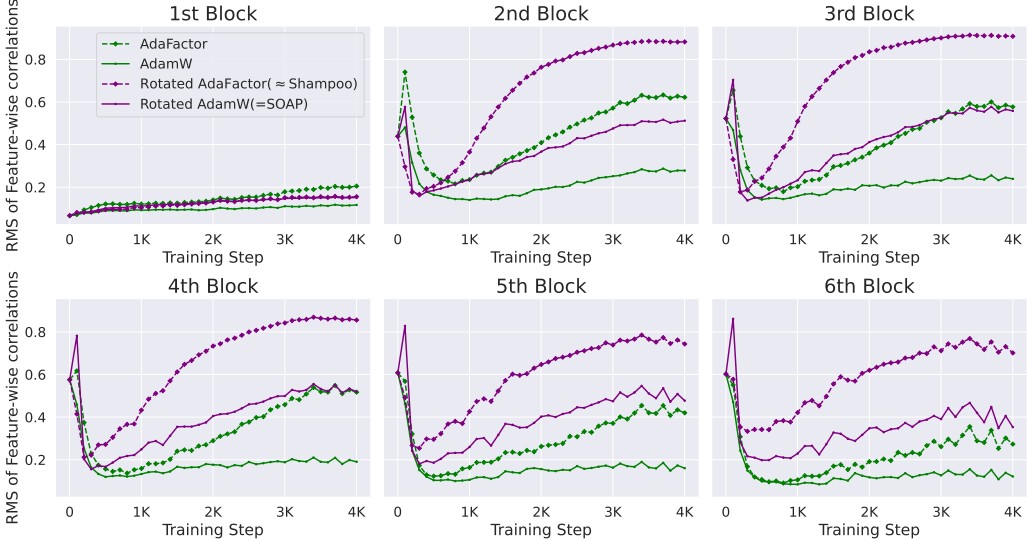

Figure 34: RMS of feature-wise correlation trajectories, equivalent to Fig 33. Or more explicitly, $\sum_{i,j \le d; i \ne j} \left( \Sigma_{\mathrm{F}} \right)^2_{i,j}$ in the notation of Eq (5). Notice that for non-diagonal preconditioners (SOAP/Shampoo), the curves in Fig 34 closely track the signal propagation trajectories in Fig 33 (and both are relatively high), meaning that their difference (the kurtosis in Eq (5)) is small. On the other hand, for diagonal preconditioners (Adam/AdaFactor), the trajectories in Fig 33 are noticeably higher than in Fig 34, and this difference is precisely the increased kurtosis. As discussed in Sec 4, theoretically predicting these feature learning behvaiours (dependent on the choices of architecture and optimiser) during training is beyond the scope of current tools in deep learning theory.

### E.1 Image Classification Experiments

#### E.1.1 Adam vs. SGD

In Sec 5, we saw that the diagonality of the preconditioning in Adam and AdaFactor was important for the emergence of outlier features. To test this further, we consider the effect of replacing Adam with SGD on OFE. SGD does not precondition gradients or, equivalently, SGD preconditions gradient with the identity matrix. In comparison to optimisers like SOAP or Shampoo that diagonally precondition in a rotated parameter space, SGD can also be thought to optimise in a rotated parameter space (for every possible rotation), precisely because it preconditions with the identity matrix.

As transformers are difficult to train (fast) with SGD, we consider OFs in a much simpler architecture and task: an MLP on CIFAR-10 image classification. Like with Shampoo and SOAP (Fig 8), in Fig 35 we see that SGD is not as susceptible to OFs as Adam, even with OF-prone architecture choices, like Normalisation layers. In fact, in this experiment SGD kurtosis actually *decreases* during training with Pre-Norm. Fig 36 shows that SGD matches Adam convergence speed in this setting. The model is a 6-layer Pre-Norm residual MLP of width 1024; we remove Pre-Norms for normless models. This also highlights that OFs are not specific to the Transformer model.

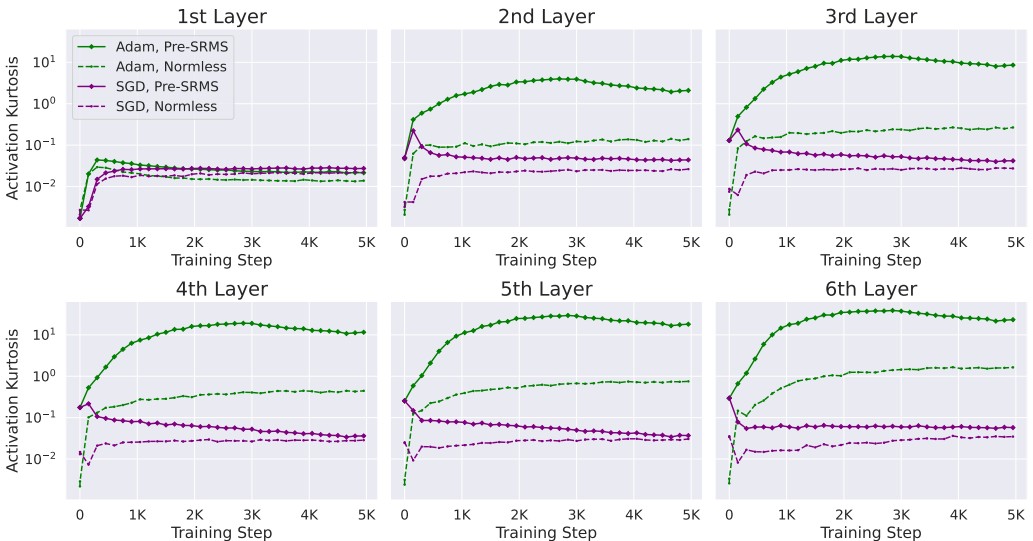

Figure 35: OFs of SGD vs Adam in an MLP on CIFAR-10. Although normalisation layers lead to higher kurtosis for a given optimiser, Adam always has higher OFs than SGD.

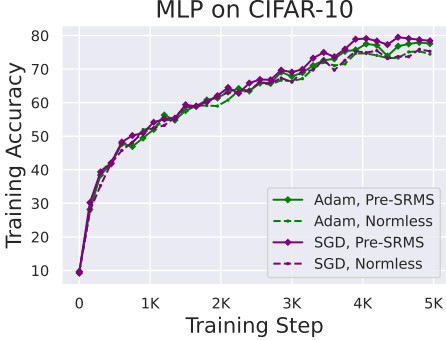

Figure 36: Train accuracy plot with SGD vs Adam of MLP on CIFAR-10, corresponding to Fig 35. Adam $\epsilon$ is the default value of $1e - 8$.

We note that the levels of kurtosis reached by AdamW in CIFAR-10 image classification MLP settings do not reach the same heights (peaking around 40 in individual layers in Fig 35) as in our transformer language modelling settings.[18] Having said that, we also show in this subsection that many of our findings concerning OFs (in terms of kurtosis) carry through to the image classification setting, including: the effect of Adam $\epsilon$ (Figs 37 and 38) and the correlation between signal propagation and kurtosis for Adam and SGD optimisers (Fig 39).

### E.1.2 Vision Transformer Experiments

To assess whether the difference in architecture between our language modelling and image classification experiments explains this observed difference in peak kurtosis, we consider the Vision Transformer (ViT) [42], which is largely the same Pre-LN transformer architecture as in our language modelling experiments. Instead of processing sequences of tokens like its language modelling counterpart, ViTs turn an image into a sequence of patches, and alternate self-attention and MLP sub-blocks (as in App A.1) to process patches.

We consider the ViT-B model, which has 12 blocks and width 768 giving 86M parameters, which is around the same parameter scale as our 130M CodeParrot experiments and also Pythia-160M in Fig 1.[19] We train on ImageNet-1K using the codebase[20] and default hyperparameters of DeIT [108], but only train for 150 epochs (instead of 300) and do not use distillation in the interests of compute time and because in this experiment we are not interested in state of the art results but rather the OF properties of ViTs. We use AdamW optimiser with a max LR of 3.75e-4 (maximum stable LR on a logarithmically spaced grid) with a batch size of 384 (data parallel across 6 RTX2080-Tis), and warm up the LR for 5 epochs before cosine decay.

We compare 3 different ViT transformer blocks: Pre-LN, Pre-LN with LayerScale [55] (LayerScale is a variant of SkipInit [51] that downweights the residual branches with a vector of trainable gains), and our OP block (using LayerScale to downweight residual branches). We use the default initialisation for residual branch downweighting factors in LayerScale, of $1e-4$. QK-Norm is implemented with LN for the OP block.

In Fig 40, we see that Pre-LN ViT-Bs do still suffer from higher peak kurtosis values (around 20 averaged across layers) compared to the OP block (around 4.5 averaged across layers), but the Pre-LN kurtosis decreases sharply as training progresses, unlike in language modelling settings, and actually ends up being lower than the OP block at the end of training. In addition, the kurtosis values on this ViT image classification task are again far lower than transformer language modelling counterparts at similar parameter scales (e.g. Fig 1 or Fig 14). Adding LayerScale [55] reduces Pre-LN kurtosis (to even below that of the OP block in terms of peak value), as expected from our findings in Sec 4. In Fig 41, we see that all three blocks have similar test accuracy performance, with the OP block having a slight advantage of 0.2%.

From these preliminary experiments with ViTs we conclude that, despite similarities, there are qualitative differences between OFs (measured via kurtosis) in image classification and language modelling tasks that are not explained through the choice of architecture (or indeed the choice of AdamW optimiser). We leave a more in-depth study to future work. It remains to be seen if OFs will be a barrier to quantising ViTs at scale, as it has been the case for LLMs, though our findings showing reduced kurtosis (both after and peak during training) suggest this may not be the case. We note Darcet et al. [79] identify artifacts in the attention maps of ViTs, which may potentially be related to OFs.

---

[18]This difference is not necessarily explained by scale, as our MLP residual stream width of 1024 is actually wider than our 130M transformer width of 768, which is the scale that produced the much higher kurtosis values in, for example, Figs 2 and 14.

[19]This is actually deeper than our default 6 layer transformer in our 130M CodeParrot experiments, and shares the same width. The parameter difference with language is that a significant fraction of parameters in language modelling are in the embedding layer.

[20]https://github.com/facebookresearch/deit

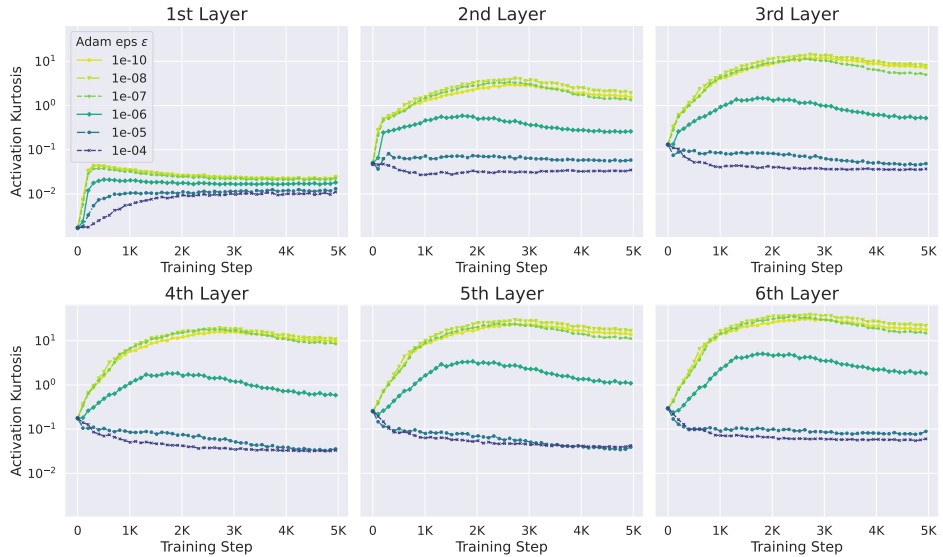

Figure 37: Kurtosis plot with different Adam $\epsilon$ with an MLP on CIFAR-10. The model uses Pre-Norm structure with SRMSNorm normalisation. Like in Fig 29, we see that larger $\epsilon$ generally leads to smaller OFs.

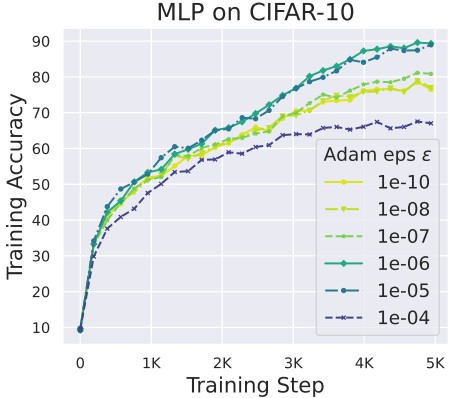

Figure 38: Train accuracy plot with different Adam $\epsilon$ of MLP on CIFAR-10, equivalent to Fig 37. In this experiment, milder values of $\epsilon \in \{1e-5, 1e-6\}$ converge fastest.

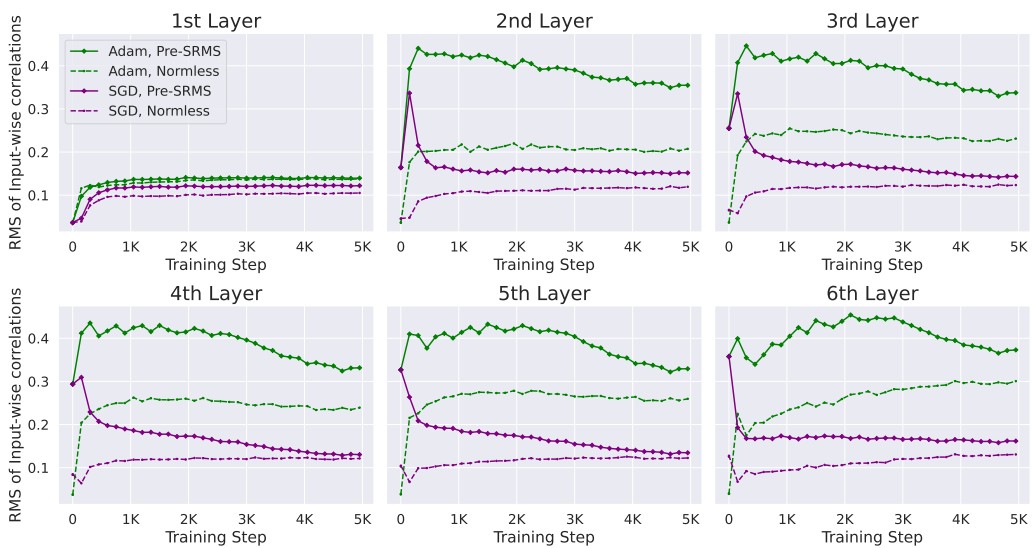

Figure 39: Effect of SGD vs Adam on Signal Prop, for models plotted in Fig 35.

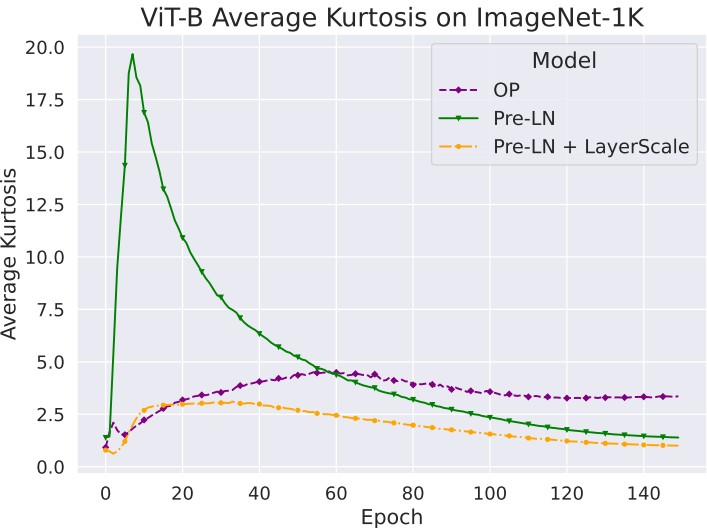

Figure 40: Average Kurtosis trajectories of ViTs trained on ImageNet-1K. Y-axis is average kurtosis across the 12 residual stream layers.

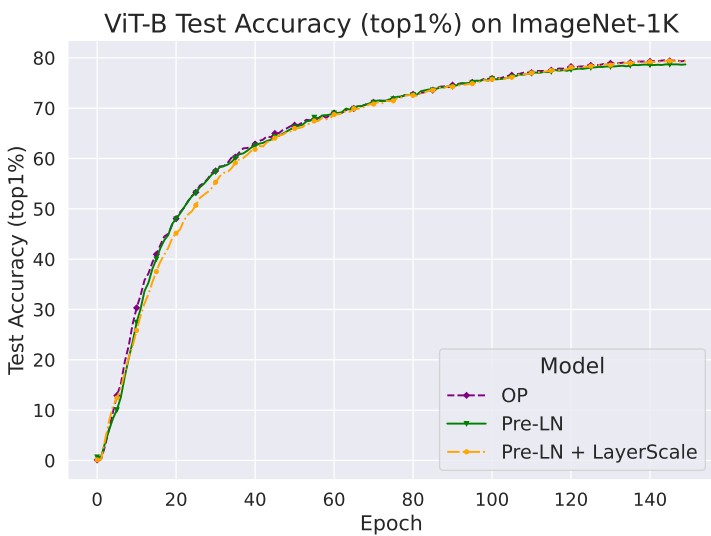

Figure 41: Test accuracies of ViTs trained on ImageNet-1K. The final test accuracies are 79.5% for OP, 79.3% for Pre-LN + LayerScale, and 78.7% for Pre-LN.

## E.2 Ablating the components of the OP block

In Tabs 3 and 4 and Fig 42 we ablate the components of our OP block. Tab 3 assesses the impact of not having an EntReg mechanism on training stability and convergence speed on the Languini dataset [39] at 320M scale. Fig 42 confirms the loss of EntReg causes entropy collapse on CodeParrot at 130M scale, which is shown to lead to unstable training in Fig 43. In these experiments, we also try the tanh thresholding as an alternative EntReg mechanism to QK-Norm. Tab 4 goes from Pre-LN to OP one step at a time, assessing the impact of different norms and downweighted residuals, in terms of OFE.

Table 3: Ablating the convergence and training benefits of the OP block. The asterisk * denotes that training failed without Flash Attention [109], which centres pre-softmax logits based on their max value and is therefore more stable. This highlights the training instability of not having some entropy regulating (EntReg) mechanism, where smaller LRs are required for stability. At a smaller (but stable) LR, the naive unnormalised model without EntReg converges much slower (17.4 vs 16.2 ppl) in this example. Even with larger LR, the EntReg mechanism in the OP block improves convergence (16.6 vs 16.2 ppl for QK-RMSNorm) compared to the naive unnormalised model. Tanh thresholding (from Grok-1) also works as an example of an alternative EntReg mechanism to QK-Norm. Because Pre-Norms appear before Query/Key weights, they already provide an implicit EntReg mechanism. As a result, adding EntReg to Pre-Norm models results in only minor changes to convergence speed in this experiment (though ViT-22B shows in other settings Pre-Norm alone is not enough [59]). Models are 320M parameters, trained also for 3.3B tokens on Languini [39] as in Tab 1.

| Model | MLP/Attn Pre-Norm | EntReg | Scaled Residual | LR | Eval PPL |
|---|---|---|---|---|---|
| Pre-LN | LN | None | Implicit | 1e-3 | 16.2 |
| Pre-RMSNorm | RMS | None | Implicit | 1e-3 | 16.3 |
| Pre-LN+QK-Norm | LN | QK-RMS | Implicit | 1e-3 | 16.0 |
| Pre-LN+Tanh | LN | Tanh | Implicit | 1e-3 | 16.2 |
| Naive unnormalised | None | None | Yes | 3e-4 | 17.4 |
| Naive unnormalised | None | None | Yes | 1e-3 | 16.6* |
| OP (QK-Norm) | None | QK-RMS | Yes | 1e-3 | 16.2 |
| OP (Tanh) | None | Tanh | Yes | 1e-3 | 16.4 |

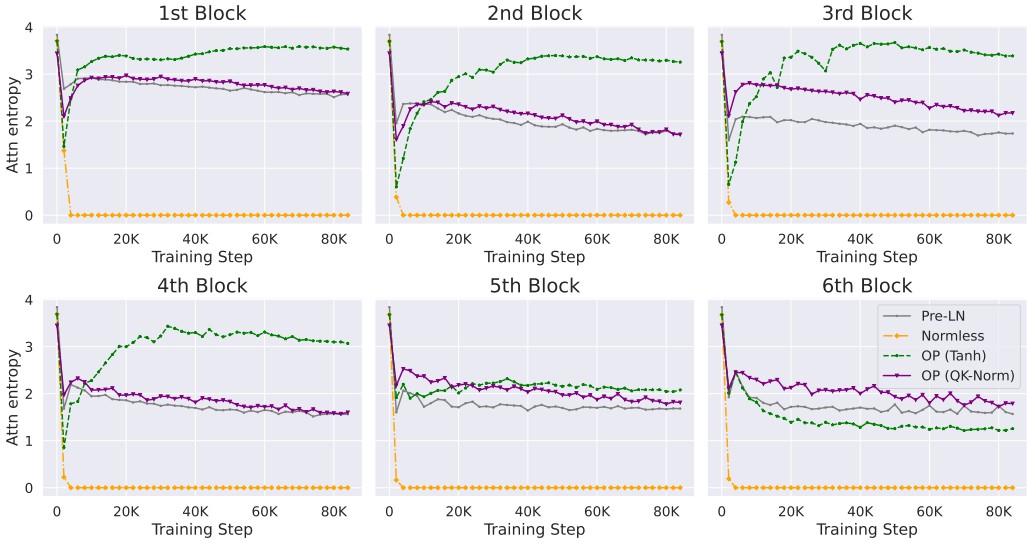

Figure 42: No EntReg leads to entropy collapse without Pre-Norms, which means training fails (as seen in Fig 43).

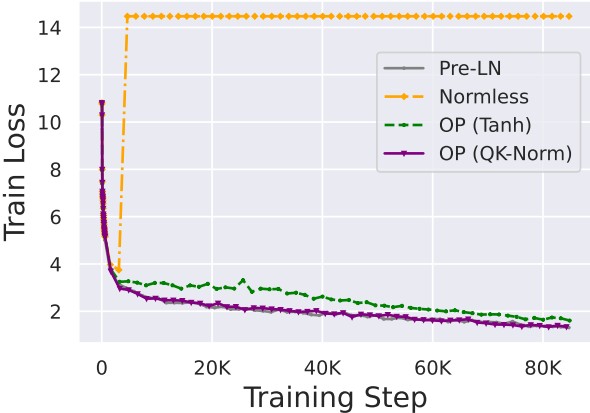

Figure 43: Entropy collapse leads to failed training. Experiment is at 130M scale on CodeParrot. OP with tanh does not fail but does converge slower in this setting; compared to Tab 3, we use learnt positional encodings in the input embedding layer, not RoPE, which may account for this difference. We tuned a few values of the $max\_attn\_val$ hyperparameter with tanh thresholding: $f(x) = max\_attn\_val \cdot \tanh(x/max\_attn\_val)$, which is set by default to 30 in Grok-1, but they did not close the convergence speed loss.

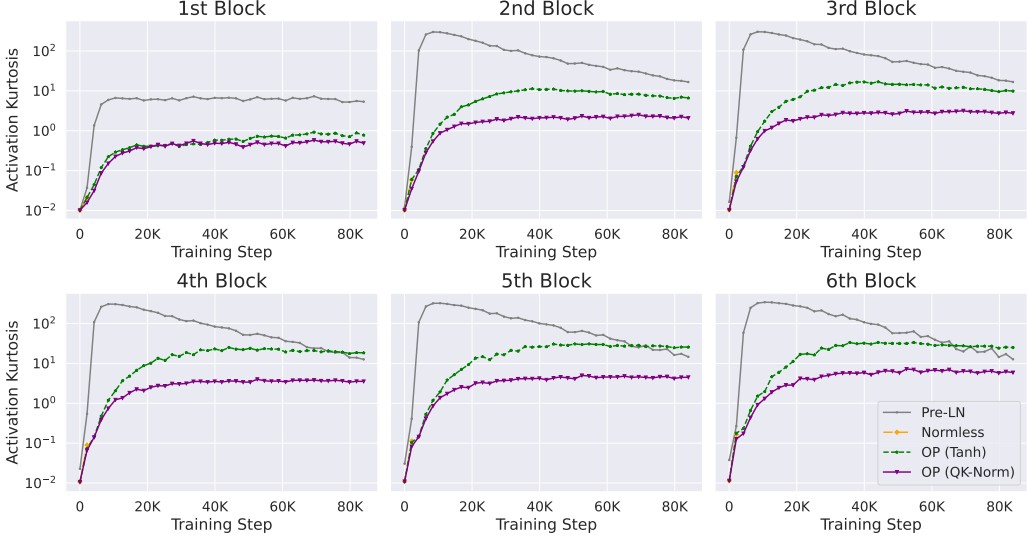

Figure 44: OP with Tanh still has reduced peak OFs compared to Pre-LN. This plot corresponds to the models shown in Fig 42.

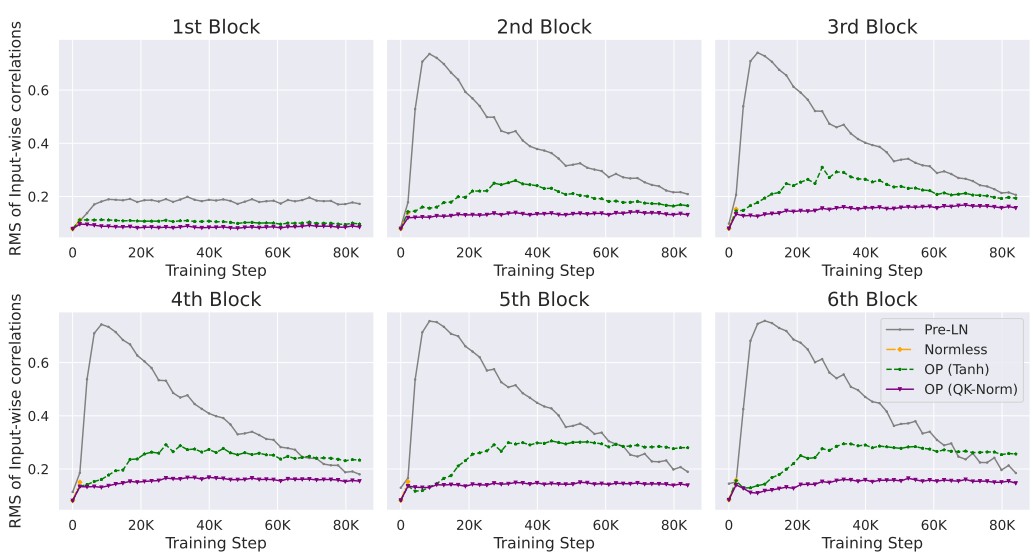

Figure 45: Signal Prop plot with OP Tanh. This plot corresponds to the models shown in Fig 43.

Table 4: Going from Pre-Norm to OP step by step. We remove or add Norms one by one, with different Norm locations depicted in Fig 47. All models trained well (at similar speeds), as they all have some form of entropy regulation (either explicit or implicit) and downweighted residuals. We present the peak Kurtosis (Eq (1)), Signal Propagation (RMS of input-wise correlations), and activation RMS ($\|\mathbf{X}\|_F$) over the training run, with mean and standard deviation over three seeds. We present results where activations $\mathbf{X}$ are the input to the second transformer block. We see that that preventing attention entropy collapse through QK-Norm helps reduce OFs (which we see coincides with improved signal propagation). On the other hand, peak activation RMS does not correlate well as a metric with peak kurtosis, across the different models. In addition, the 2 best models in terms of OFs (our OP and also the third last row, which has no Pre-V or Pre-MLP Norms) are 1-homogeneous (at least at initialisation), which implies that the fact that Pre-V or Pre-MLP Norms make the residual stream scale independent is detrimental for OFE. This is corroborated by Fig 46, which plots the trajectories for the three models (1. Post-QK+Pre-V, 2. QK Norms only and 3. OP) that achieved peak kurtosis lower than 10. Fig 46 shows that the non-homogeneity (due to a Pre-V Norm) leads to a large initial increase in kurtosis and signal propagation in this setting, like we consistently see with Pre-Norm blocks e.g. Fig 5. Models are 130M scale on CodeParrot.

| Model | Norm | | | | Scaled Resid | Homog.? | Act RMS | Signal Prop | Kurtosis |
|---|---|---|---|---|---|---|---|---|---|
| | Post-QK | Pre-QK | Pre-V | Pre-MLP | | | | | |
| Pre-RMS | None | RMS | RMS | RMS | Implicit | No | $5.45_{\pm 0.13}$ | $0.72_{\pm 0.03}$ | $131.8_{\pm 21.2}$ |
| Scaled Resids | None | RMS | RMS | RMS | Yes | No | $3.97_{\pm 0.09}$ | $0.47_{\pm 0.04}$ | $46.4_{\pm 14.0}$ |
| All Norms | RMS | RMS | RMS | RMS | Yes | No | $3.92_{\pm 0.07}$ | $0.24_{\pm 0.05}$ | $12.7_{\pm 10.2}$ |
| Attn Norms only | RMS | RMS | RMS | None | Yes | No | $4.38_{\pm 0.07}$ | $0.29_{\pm 0.04}$ | $11.8_{\pm 8.03}$ |
| Post-QK+Pre-V | RMS | None | RMS | None | Yes | No | $4.40_{\pm 0.06}$ | $0.27_{\pm 0.01}$ | $6.4_{\pm 1.32}$ |
| QK Norms only | RMS | RMS | None | None | Yes | Yes | $4.32_{\pm 0.06}$ | $0.15_{\pm 0.01}$ | $2.5_{\pm 0.93}$ |
| Pre-QK only | None | RMS | None | None | Yes | Yes | $4.38_{\pm 0.01}$ | $0.37_{\pm 0.05}$ | $64.0_{\pm 49.5}$ |
| OP (ours) | RMS | None | None | None | Yes | Yes | $4.46_{\pm 0.09}$ | $0.17_{\pm 0.01}$ | $4.3_{\pm 1.49}$ |

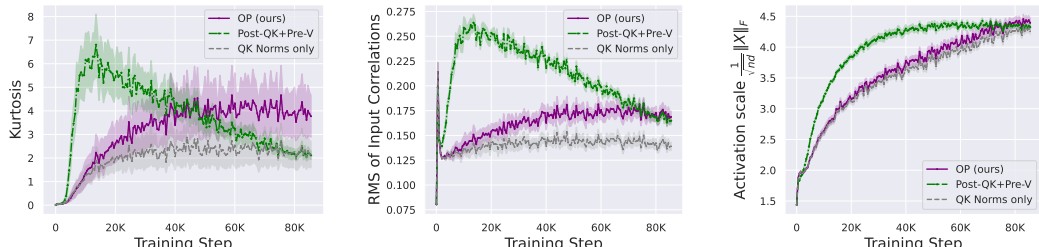

Figure 46: Training trajectories of kurtosis, signal propagation and activation scales for the three best configurations in Tab 4. The setting with Pre-V Norm (which is not 1-homogeneous) sees a large initial increase in all metrics, with kurtosis and input correlations peaking within 10K steps before reducing during training.

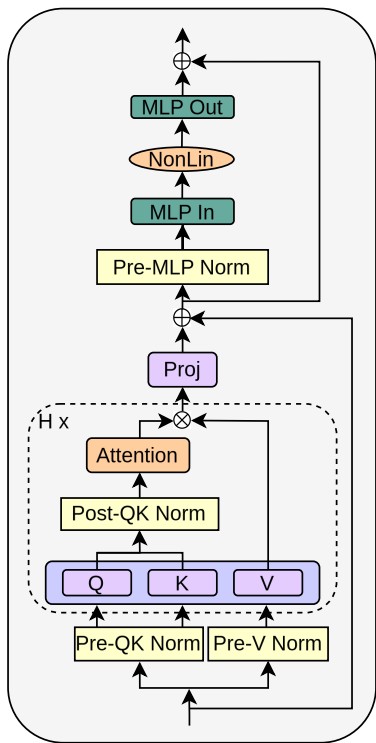

Figure 47: A transformer block with many different Norm layers depicted, to help parse the ablations we consider in Tab 4. Note we break down the standard attention Pre-Norm into Pre-QK Norm and Pre-V Norm because removal of Pre-V Norm makes the attention sub-block homogeneous (i.e. $f(x)$ is homogeneous if $f(kx) = kf(x)$ for some scalar $k > 0$), hence acts differently to Pre-QK Norm, which acts as an implicit regulator for attention entropy.

## F Orders of Activation Updates for Kurtosis

To better appreciate the effect of different optimiser hyperparameters on OFs, we now consider how the updates that arise during training to a representation matrix $\mathbf{X} \in \mathbb{R}^{n \times d}$ can lead to increasing kurtosis (and OFs). In general, a training step (e.g. with a gradient/Adam update on trainable parameters earlier in the forward pass than $\mathbf{X}$) will lead to an update $\mathbf{X} \leftarrow \mathbf{X} + \Delta^{\mathbf{X}}$.

Recall that $\mathrm{Kurt}(\mathbf{X})$ is an defined through comparing the fourth $m_4(\mathbf{X})$ and second $m_2(\mathbf{X})$ moments of neuron RMS $\sqrt{\frac{1}{n}\sum_{\alpha=1}^{n}\mathbf{X}_{\alpha,j}^2}$ for different $j$. As such, it is natural to ask how updating $\mathbf{X} \leftarrow \mathbf{X} + \Delta^{\mathbf{X}}$ updates these moment statistics. We first study the second moment update $u_2$:

$$u_2 \stackrel{\text{def}}{=} m_2(\mathbf{X} + \Delta^{\mathbf{X}}) - m_2(\mathbf{X}) = \frac{1}{d}\sum_{j=1}^{d}\left(\frac{1}{n}\sum_{\alpha=1}^{n}(\mathbf{X} + \Delta^{\mathbf{X}})_{\alpha,j}^2\right) - \frac{1}{d}\sum_{j=1}^{d}\left(\frac{1}{n}\sum_{\alpha=1}^{n}\mathbf{X}_{\alpha,j}^2\right) \tag{14}$$

$$= \frac{1}{nd}\left(u_{2,1} + u_{2,2}\right), \quad \text{with} \tag{15}$$

$$u_{2,1} \stackrel{\text{def}}{=} \sum_{j=1}^{d}\sum_{\alpha=1}^{n}2\Delta^{\mathbf{X}}_{\alpha,j}\mathbf{X}_{\alpha,j}, \quad u_{2,2} \stackrel{\text{def}}{=} \sum_{j=1}^{d}\sum_{\alpha=1}^{n}(\Delta^{\mathbf{X}}_{\alpha,j})^2. \tag{16}$$

Likewise for the fourth moment update $u_4$:

$$u_4 \stackrel{\text{def}}{=} m_4(\mathbf{X} + \Delta^{\mathbf{X}}) - m_4(\mathbf{X}) = \frac{1}{d}\sum_{j=1}^{d}\left(\frac{1}{n}\sum_{\alpha=1}^{n}(\mathbf{X} + \Delta^{\mathbf{X}})_{\alpha,j}^2\right)^2 - \frac{1}{d}\sum_{j=1}^{d}\left(\frac{1}{n}\sum_{\alpha=1}^{n}\mathbf{X}_{\alpha,j}^2\right)^2 \tag{17}$$

$$= \frac{1}{n^2 d}\left(u_{4,1} + u_{4,2} + u_{4,3} + u_{4,4}\right), \quad \text{with} \tag{18}$$

$$u_{4,1} \stackrel{\text{def}}{=} \sum_{j=1}^{d}\sum_{\alpha,\beta=1}^{n}4\Delta^{\mathbf{X}}_{\alpha,j}\mathbf{X}_{\alpha,j}\mathbf{X}_{\beta,j}^2, \quad u_{4,2} \stackrel{\text{def}}{=} \sum_{j=1}^{d}\sum_{\alpha,\beta=1}^{n}2(\Delta^{\mathbf{X}}_{\alpha,j})^2\mathbf{X}_{\beta,j}^2 + 4\Delta^{\mathbf{X}}_{\alpha,j}\mathbf{X}_{\alpha,j}\Delta^{\mathbf{X}}_{\beta,j}\mathbf{X}_{\beta,j}, \tag{19}$$

$$u_{4,3} \stackrel{\text{def}}{=} \sum_{j=1}^{d}\sum_{\alpha,\beta=1}^{n}4\mathbf{X}_{\alpha,j}\Delta^{\mathbf{X}}_{\alpha,j}(\Delta^{\mathbf{X}}_{\beta,j})^2, \quad u_{4,4} \stackrel{\text{def}}{=} \sum_{j=1}^{d}\sum_{\alpha,\beta=1}^{n}(\Delta^{\mathbf{X}}_{\alpha,j})^2(\Delta^{\mathbf{X}}_{\beta,j})^2. \tag{20}$$

Above, we have broken down the $p^{\text{th}}$ moment update $u_p$ into $(u_{p,l})_l$, where $u_{p,l}$ denotes the contribution to $u_p$ that is order $l$ in $\Delta^{\mathbf{X}}$. The reason for this is that, typically, a learning rate parameter $\eta$ is used such $\Delta^{\mathbf{X}}$ is linear in $\eta$, and so $u_{p,l}$ is order $l$ in $\eta$.[21] Usually, $\eta$ is chosen to be small such that $\Delta^{\mathbf{X}}$ is small element-wise relative to $\mathbf{X}$. Note that the quadratic update terms $u_{p,2}$ are always positive,[22] whereas the linear terms $u_{p,1}$ are not necessarily positive, so we might expect quadratic terms to drive any increase in the $p^{\text{th}}$ moment $m_p$.

In Fig 48, we plot the cumulative sum of these $(u_{p,l})$ terms, for our OP block, a default Pre-LN block, and also three modifications that reduce OFs in Pre-LN (increasing Adam epsilon from $1e-8$ to $1e-4$, reducing maximum LR from $1e-3$ to $3e-4$, and using a non-diagonal preconditioner e.g. SOAP [69]) trained on CodeParrot at 130M scale. We see indeed that the cumulative $u_{4,2}$ quadratic term dominates the update to $u_4$ and drives the increase in $m_4$ in the default Pre-LN model. Rducing LR, increasing Adam $\epsilon$, and using SOAP, reduce this term, which also reduces the growth in fourth moment and kurtosis. For the choice of LR this is intuitive: in the small LR $\eta \to 0$ limit the linear first order term $u_{4,1}$ will dominate and the effect of quadratic $u_{4,2}$ can be ignored. The impact of sub-leading order terms like $u_{4,2}$ in OFE is related to the discretisation drift between discrete-time gradient descent and continuous-time gradient flow [110]. Fig 49 plots the non-cumulative version of Fig 48.

On the other hand, in Fig 48 the OP block has a relatively large cumulative increase from $u_{4,2}$ that is matched by a decrease in $u_{4,1}$ and a large increase in $u_2$, which means the kurtosis (which is the

---

[21]For example, if we have $\mathbf{X} = \mathbf{H}\mathbf{W}$ for a previous layer $\mathbf{H}$ that is fixed (e.g. embedding layer in a transformer). Then we usually update weights $\mathbf{W} + \Delta^{\mathbf{W}}$ linearly in $\eta$, and so $\Delta^{\mathbf{X}} = \mathbf{H}\Delta^{\mathbf{W}}$ is also linear in $\eta$. For other layers we need to consider the change in $\mathbf{H}$ too, but this will also be linear in $\eta$ to leading order.

[22]This is straightforward to see for $u_{2,2}$. For $u_{4,2}$ the second summand can be factorised as $\sum_j\left(\sum_\alpha\mathbf{X}_{\alpha,j}\Delta_{\alpha,j}\right)^2$ which is positive.

ratio $m_4/m_2^2$) does not increase as much as Pre-LN. Fig 50 shows that $u_{4,2}$ dominates the cubic $u_{4,3}$ and quartic $u_{4,4}$ update terms to the fourth moment, so we can focus on studying $u_{4,2}$. We plot the moment updates for the input to the second attention block (out of six).

The models presented in Figs 48 to 50 were trained using Adam or SOAP without momentum, akin to RMSProp [111]: we set $\beta_1 = 0$ and $\beta_2 = 0.95$ in Adam.[23] The reason for this was to separate out the contribution of individual training steps on the kurtosis updates. If instead we re-introduce momentum with $\beta_1 = 0.9$, then the different update steps become mixed and the leading order $u_{4,1}$ dominates the updates to the kurtosis for the Pre-LN model, as seen in Fig 51. The choice of $\beta_2 = 0.95$ highlights that OFs are not specific to the standard choice of $\beta_2 = 0.999$ in AdamW.

---

[23]Recall SOAP [69] is just Adam in a rotated basis (obtained via Shampoo), and still uses $\beta_1, \beta_2$ hyperparameters

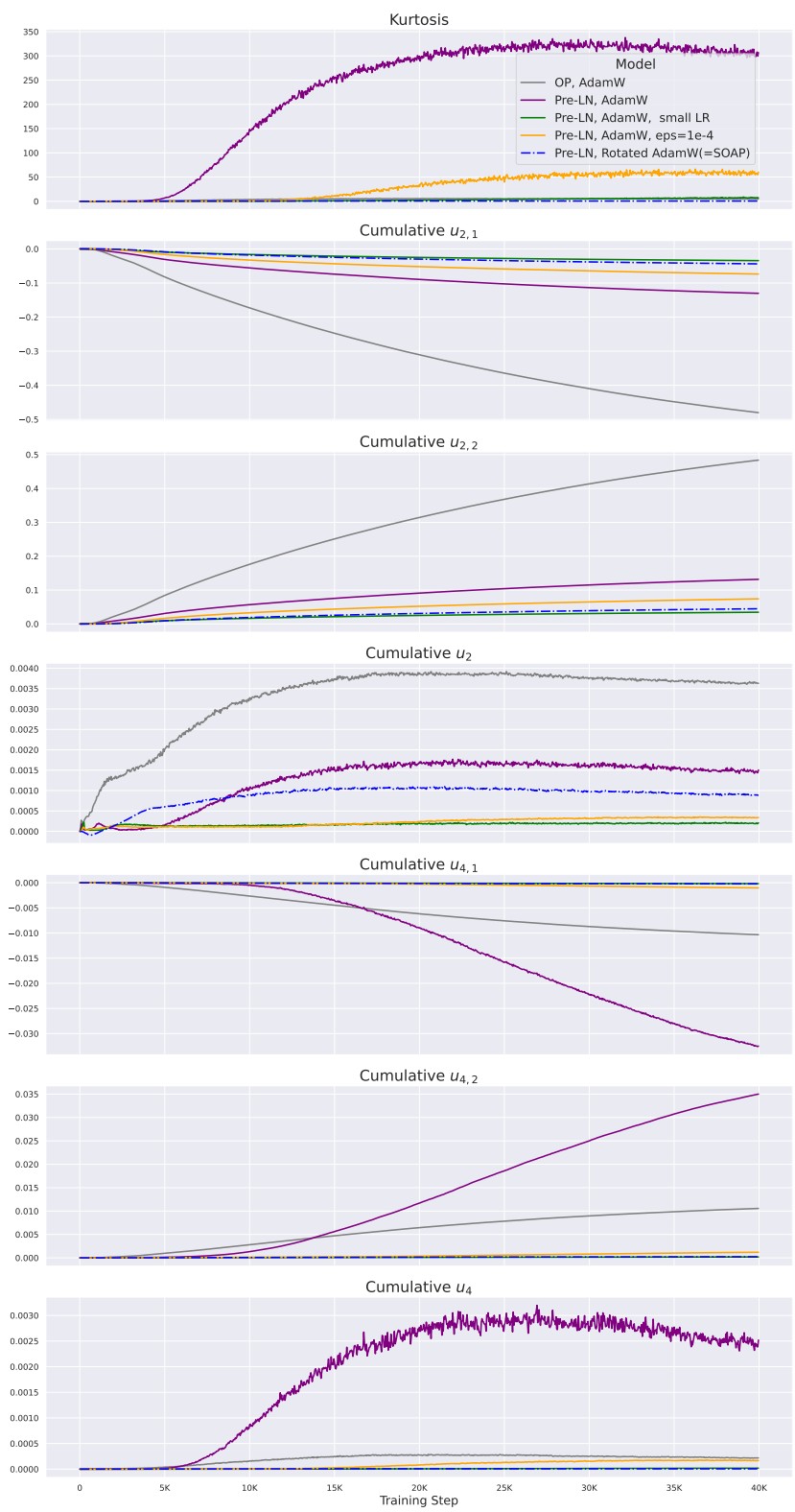

Figure 48: Cumulative metrics to track kurtosis updates. Models were trained without momentum. We see that the quadratic $u_{4,2}$ term dominates updates to the fourth moment.

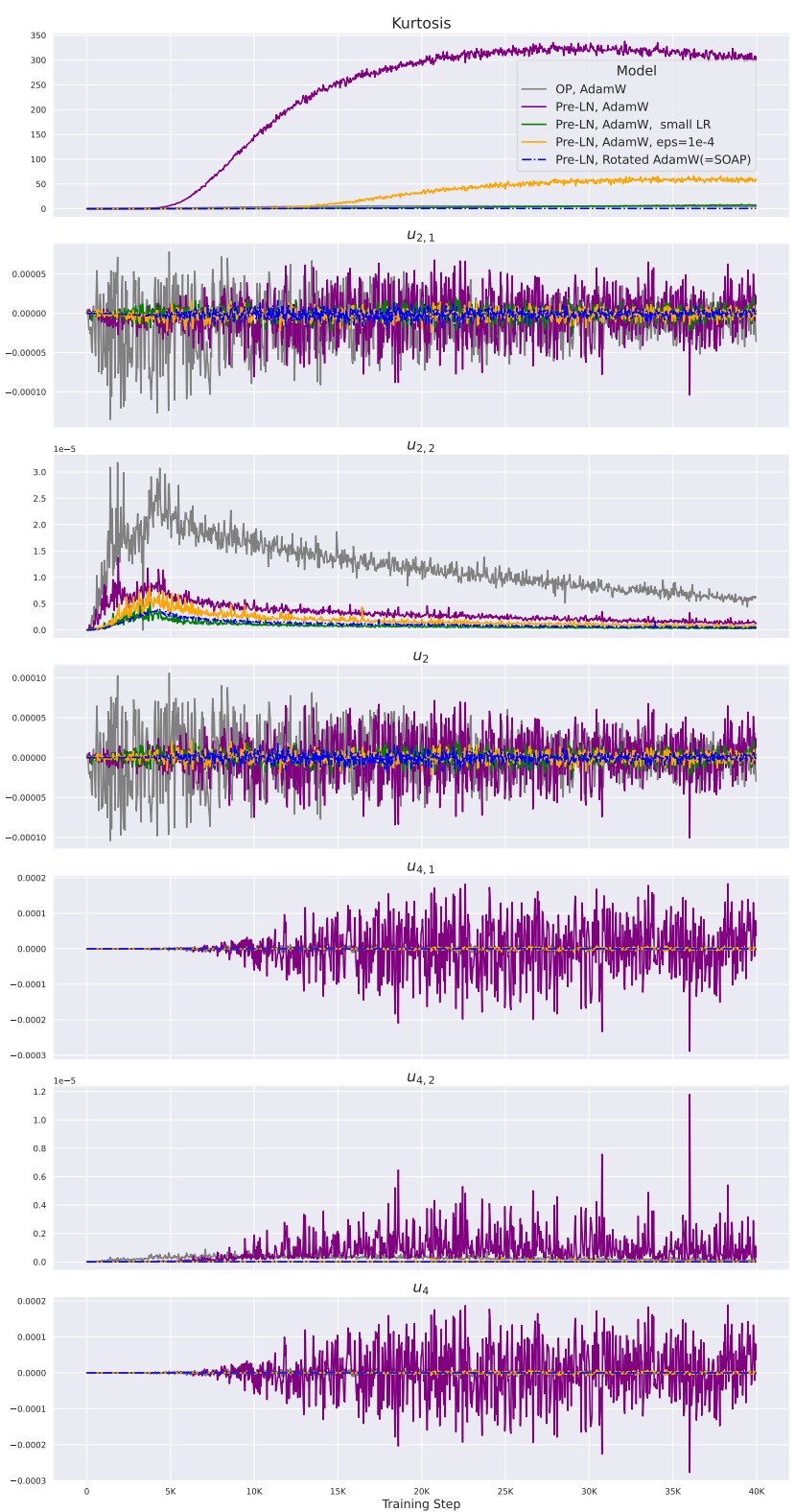

Figure 49: Non-cumulative metrics to track kurtosis updates. Models were trained without momentum.

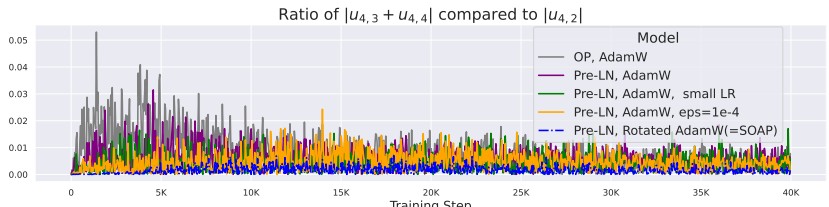

Figure 50: Sub-leading order terms are dominated by $u_{4,2}$.

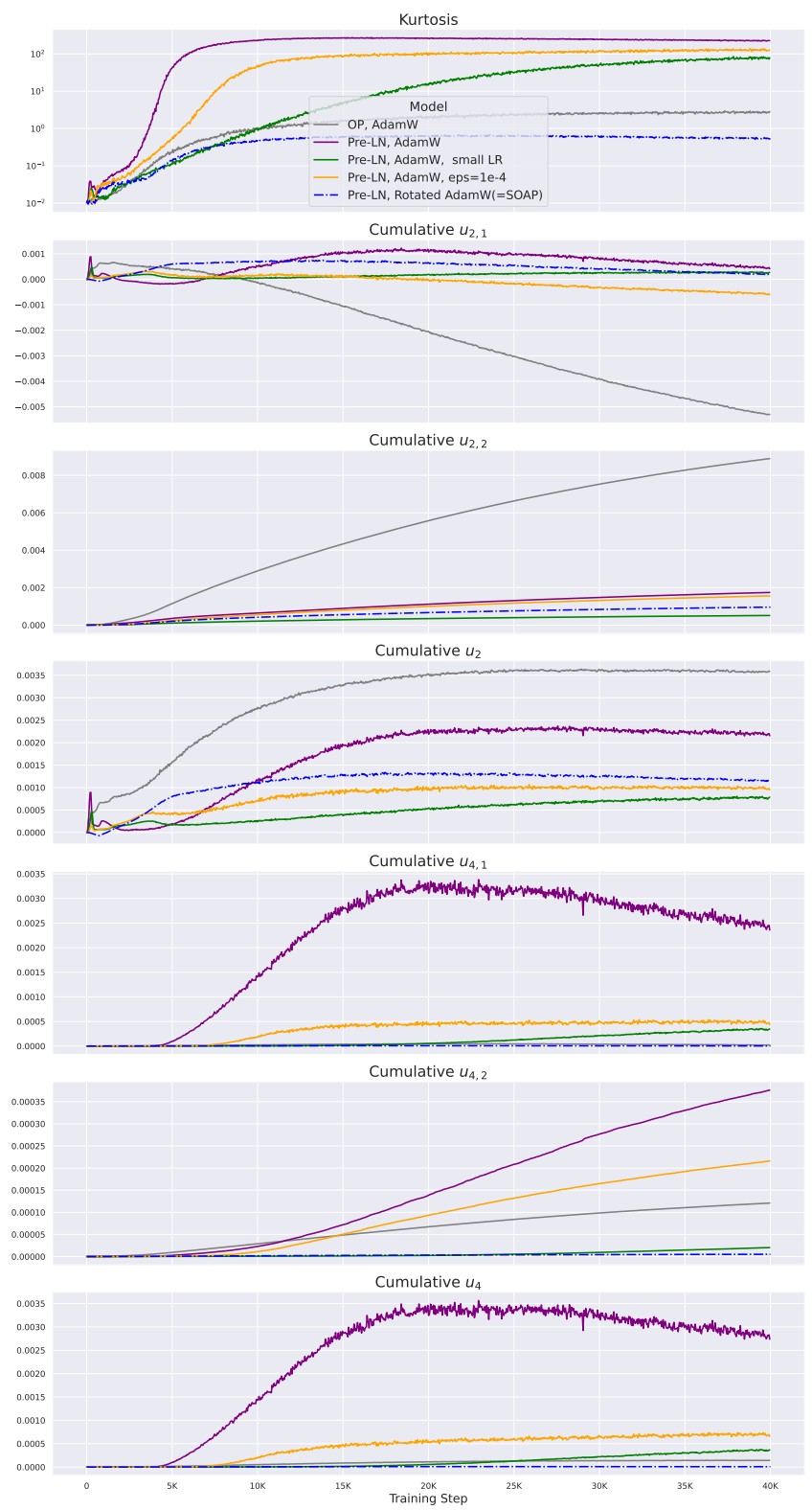

Figure 51: Cumulative metrics to track kurtosis updates. Models trained **with** momentum. The leading order $u_{4,1}$ term now dominates the updates to the fourth moment.

# G   Worse Signal Prop Means Higher Activation Kurtosis in Gaussian Features

**Proposition G.1** (Bad Signal Propagation implies higher kurtosis for Gaussian features). *Suppose we have $\mathbf{X} \in \mathbb{R}^{n \times d}$ zero-mean Gaussian distributed with all inputs uniformly correlated with some $\rho > 0$, and independent features (across columns). That is: $\mathbb{E}[\mathbf{X}] = \mathbf{0}$ and $\mathbb{E}[\mathbf{X}_{\alpha,j}\mathbf{X}_{\beta,k}] = \rho \cdot \mathbf{1}\{j = k\} + (1-\rho) \cdot \mathbf{1}\{j = k\} \cdot \mathbf{1}\{\alpha = \beta\}$.[24]*

*Then, if we consider the feature-wise Gram matrix $\Sigma_F = \frac{1}{n}\mathbf{X}^\top \mathbf{X}$, we have that the expected squared diagonal entry of $\Sigma_F$ is $\mathbb{E}[(\Sigma_F)_{1,1}^2] = 1 + 2\rho^2 + o_n(1)$ increases as $\rho$ increases, whereas the expected diagonal entry is $\mathbb{E}[(\Sigma_F)_{1,1}] = 1$ is independent of $\rho$.*

*Proof.* As Gaussians are determined by their first two moments, let us suppose that $\mathbf{X}_{\alpha,j} = \sqrt{1-\rho}u_{\alpha,j} + \sqrt{\rho}v_j$, where $(u_{\alpha,j})_{\alpha,j}$ and $(v_j)_j$ are independent standard Gaussians. Then, for two neuron indices $k, l \leq d$ we have:

$$\left(\mathbf{X}^T\mathbf{X}\right)_{k,l} = (1-\rho)\sum_{\alpha \leq n} u_{\alpha,k}u_{\alpha,l} \tag{21}$$

$$+ \rho n v_k v_l \tag{22}$$

$$+ \sqrt{\rho(1-\rho)}\sum_{\alpha \leq n} u_{\alpha,k}v_k + u_{\alpha,l}v_l. \tag{23}$$

We are interested in the diagonal elements of $\Sigma_F = \frac{1}{n}\mathbf{X}^\top\mathbf{X}$, when $k = l$ above. In this case, we have $(u_{\alpha,k}^2)_\alpha$ and $v_k^2$ are all independent chi-squared $\chi^2$ distributed with 1 degree of freedom. For $Z \sim \chi_1^2$, we have $\mathbb{E}[Z] = 1$ and $\mathbb{E}[Z^2] = 3$.

For the first moment, we take the expectation above and note that the summands of Eq (23) are products of independent zero-mean Gaussians (so zero mean). This gives $\mathbb{E}[\mathbf{X}^T\mathbf{X}_{k,k}] = n$ and hence $\mathbb{E}[(\Sigma_F)_{1,1}] = 1$, as required.

For the second moment, we note that all cross products in $\left(\mathbf{X}^T\mathbf{X}\right)_{k,k}^2$ will disappear in expectation when we square besides the one involving Eqs (21) and (22), as both terms will be $\chi_1^2$ distributed (hence not zero-mean). On the other hand, all cross products involving Eq (23) will be an odd order in at least one zero-mean independent Gaussian (hence zero-mean).

The square of Eq (21) is $(1-\rho)^2 n(n+2)$ in expectation, which can be seen by the fact that $\sum_{\alpha \leq n} u_{\alpha,k}^2$ is actually a $\chi_n^2$ distribution, with mean $n$ and variance $2n$. Hence for $Z \sim \chi_n^2$, we have $\mathbb{E}[Z^2] = \mathbb{E}[Z]^2 + \text{Var}(Z) = n^2 + 2n$.

The square of Eq (22) is $3\rho^2 n^2$ in expectation, again by properties of $\chi_1^2$ random variables.

The square of Eq (23) is $O(n)$ (in fact $4\rho(1-\rho)n$) in expectation and will be dominated by the $O(n^2)$ terms. To see this, we note that Eq (23) is a sum of $n$ zero mean i.i.d. random variables, so one can use the additive property of variances for independent random variables.

Finally, the cross term between Eqs (21) and (22) is $2\rho(1-\rho)n^2$ in mean. One factor of $n$ comes from the sum of inputs $\alpha \leq n$ and the other comes from Eq (22) already. The product of two independent $\chi_1^2$ random variables is 1 in expectation.

Putting this all together, we have

$$\mathbb{E}[\mathbf{X}^T\mathbf{X}_{k,k}^2] = (1-\rho)^2 n(n+2) + 3\rho^2 n^2 + 4\rho(1-\rho)n + 2\rho(1-\rho)n^2 \tag{24}$$

$$= \left((1-\rho)^2 + 3\rho^2 + 2\rho - 2\rho^2\right)n^2 + O(n) \tag{25}$$

$$= (1 + 2\rho^2)n^2 + O(n) \tag{26}$$

As $\Sigma_F = \frac{1}{n}\mathbf{X}^T\mathbf{X}$, we divide Eq (24) by $n^2$, and obtain our desired result. $\qquad\square$

---

[24]Note this covariance gives a "uniform" correlation structure $\mathbb{E}[\frac{1}{d}\mathbf{X}\mathbf{X}^\top] = (1-\rho)\mathbf{I}_n + \rho\mathbf{1}_n\mathbf{1}_n^\top$, which has been studied before in Noci et al. [27], He et al. [28] as a way to study signal propagation in sequences. Rank collapse [60] is when $\rho = 1$.

Above, we note that $\mathbb{E}[(\Sigma_{\mathrm{F}})_{1,1}^2]$ is equivalent to the fourth moment $m_4$ in our feature-wise kurtosis definition Eq (1), while $\mathbb{E}[(\Sigma_{\mathrm{F}})_{1,1}]$ corresponds to the second moment $m_2$. Hence, Prop G.1 demonstrates that worse signal propagation (in terms of higher $\rho$) leads to higher kurtosis.

We note that the result is restricted to a Gaussian setting with independent features. This is an accurate description of large-width NN initialisation [19–21], but does not capture training dynamics as we discuss in the main paper. Indeed, the maximum kurtosis $(1 + 2\rho^2)$ is 3 when $\rho = 1$, whereas in our experiments we obtain much higher values during training (and the maximum is the width $d$, which is considerably larger than 3 in practice). This represents a gap in our theoretical understanding and practice, which we leave for future study.

