# OpenReview forum: "Understanding and Minimising Outlier Features in Transformer Training"
_NeurIPS.cc/2024/Conference — NeurIPS 2024 poster_

### Official Review · Reviewer_cUUo · 2024-07-11

**Soundness:** 4
**Presentation:** 4
**Contribution:** 4
**Rating:** 10
**Confidence:** 4

**Summary:**

This paper is doing a lot and is a rare case of the abstract/title really underselling what the paper contains. Basically, the paper investigates outlier feature emergence (OFE) in LLMs and some potential fixes for it. The paper argues that such a study is important for both practical reasons (preventing OFE aids quantization) and theory reasons (understanding training dynamics). It clearly succeeds on both fronts, developing a transformer block that empirically reduces OFE and producing theoretical insights that touch on Signal Propagation, entropy collapse, optimization, and more.

**Strengths:**

Originality: This work clarifies that some prior hypotheses about causes of OFE are inconsistent with experiments. It makes strides towards understanding OFE by combining ideas from Signal Prop and entropy collapse (e.g.), in ways that are insightful, novel, and excellently motivated. The related mathematical analyses are also very well done.

Quality: This is a thorough and careful study, making claims that are well supported by the experiments.

Clarity: This paper is extremely well written and contextualized relative to prior work.

Significance: This work will be appreciated by multiple communities for its wealth of insights. For example, the augmentation of norm-free Signal Prop-focused networks with QK norm to address entropy collapse (making norm-free transformer training competitive) -- which was inspired by the idea that vanilla norm layers help prevent entropy collapse -- is likely to have effects on transformer training procedures going forward.

**Weaknesses:**

A lot of experimental details are not in the main text, which should be addressable with an extra page.

Experiments with other LLMs and at larger scales are unnecessary but would further increase the significance of this work. The authors addressed this in their limitations section.

**Questions:**

Line 118 has a typo, I think “untrainable” was intended.

Line 193: isn't it normalized by the square of the second moment?

Line 314 has a typo in a figure reference.

Figure 9: make the solid and dashed lines represent the same thing for both SGD and Adam.

Consider moving more discussion of appendix F to the main text.

**Limitations:**

The limitations are well covered.

---

> ### Author Rebuttal · Authors · 2024-08-07
>
> We thank the reviewer for their time and effort in carefully reviewing our work. We are pleased that the review demonstrates a clear understanding of our contributions, and in particular how our work builds on existing work to relate OFEs to areas such as signal Propagation and entropy collapse. We are also grateful that the reviewer has spent time digging deep into our paper’s appendix, beyond the main 9 pages.
>
> Overall the review is extremely positive, and we are happy that our work was well received by Reviewer cUUo. Indeed, we share the belief that “this work will be appreciated by multiple communities for its wealth of insights” and that our insights will “have effects on transformer training procedures going forward.”
>
> *Weaknesses* We will make sure to use an extra page to include more detail of our experimental setup, and thank the reviewer for spotting the typos, which we will amend. Regarding scale, we agree with the reviewer that scaling beyond 1.2B is “unnecessary but would further increase the significance of this work”. However, to address scale (which is also mentioned as a weakness by R7V85) in Figures 1* and 2* of the additional pdf page we present loss and kurtosis curves of LLaMa-style transformers trained at 7B parameters for 6B tokens. We did not have enough compute to perform hyperparameter tuning, but we still see that the OP block closely matches the performance of Pre-Norm, and is vastly better in terms of kurtosis. This mirrors our findings at smaller scale in the submission. We provide more context and details of the experimental setup in our response to R7V85.
>
> Given the thoughtful nature of the review, we would like to use the remainder of our rebuttal to engage in further discussion with the reviewer on some topics/questions/experiments that may be of interest:
>
> **Quantisation experiments** In Table 1* of the additional pdf page, we go back to our original motivation in studying OFs and assess how our various architecture and optimisation suggestions affect quantisation. We take the OPT-125m setting of Bondarenko et al 2023 [14] and vary architecture and optimiser. For architecture, we compare OP to Pre-LN and also the Gated Attention baseline of [14], and for optimiser choices we take those identified in Section 5 (increased LR, and increased Adam epsilon), as well as removing dropout regularisation present in [14]. After training on BookCorpus+Wikipedia in mixed FP16/32 we quantise the models to W8A8 (int8 weights and activations) and assess the quantisation error (in perplexity lost). We also present the average kurtosis across layers in Table 1*.
>
> In Table 1* we see that our kurtosis metric for OFs is indeed highly correlated with quantisation error across different architectures and optimiser choices. For example, Pre-LN has consistently high kurtosis, and consistently catastrophic quantisation error. Moreover, the OP block has consistently low kurtosis across different optimisation hyperparameters, and also low quantisation error. Finally, the quantised Gated Attention baseline is better than Pre-LN but struggles with aggressive hyperparameters that improve FP16/32 performance but at the expense of OFs (like large LRs as identified in Section 5, and removed dropout reg). However, both kurtosis and quantisation are improved in Gated Attention  when increasing Adam epsilon, as identified in section 5. We believe this experiment ties together our findings and motivation throughout the paper. We present a more detailed discussion of our quantisation results in the global response.
>
> **Underselling in title/abstract** We thank the reviewer for their kind and positive words regarding underselling. Throughout the paper, we were very careful to avoid overclaiming in our wording. This is reflected also in our choice of title and abstract. The reviewer is likely correct that we may have undersold in the process.
>
> **Stated limitations** We are glad that the reviewer appreciates our stated limitations, and sees the strengths of our submission based on their own merit. We chose to include a thorough limitations section in the spirit of honest research and progressing the field in terms of future directions. While we stand by this decision, we also accept that including our limitations may have come to our own detriment in the review process.
>
> **Disparity across reviews** We note that there is an unusually high variance in the reviews for our work. In particular, many of the criticisms of Reviewer Lb1w are in direct opposition of Reviewer cUUo. We find such criticisms to be unfounded, and we have used several quotes from RcUUo’s review in our rebuttal to Reviewer Lb1w.
>
> We hope reviewer cUUo will continue to support our work throughout the discussion period, and thank them once again for their careful review. We also welcome any additional feedback.

---

> ### Comment · Reviewer_cUUo · 2024-08-14
> **Award quality**
>
> AC, I think this paper should be strongly considered for an oral presentation, and it clearly deserves at least a spotlight award.
>
> ---
>
> All, I have read the other reviewers' comments and would like to maintain my high score. Based on the rebuttal, I increased my confidence from 3 to 4. I would also like to clarify that, to the best of my recollection, this is the first time in a couple of years that I have given a score higher than 8 to an ICML, ICLR, or NeurIPS submission.
>
> I had a similarly high opinion of this submission before I saw the rebuttal. The rebuttal then showed that the original submission's findings hold at larger (7B) model sizes, and that the original submission's insights can (a) explain failures of quantization in a new set of experiments, and (b) provide hyperparameter/architecture tuning guidance that mitigates these failures.
>
> ---
>
> > The reviewer is likely correct that we may have undersold in the process.
>
> Maybe you didn't "undersell", but you actually delivered on what you described in the title and abstract much better than expected.
>
> > In Table 1* of the additional pdf page, we go back to our original motivation in studying OFs... We believe this experiment ties together our findings and motivation throughout the paper.
>
> I agree completely. It will make an excellent addition to the final version of the paper, tying everything together while corroborating key findings.

---

> > ### Author Response · Authors · 2024-08-14
> > **Thank you**
> >
> > We sincerely thank the reviewer for championing our paper. We deeply appreciate it.
> >
> > Best,
> > Authors

---

### Official Review · Reviewer_7V85 · 2024-07-16

**Soundness:** 3
**Presentation:** 4
**Contribution:** 3
**Rating:** 7
**Confidence:** 3

**Summary:**

The paper tackles outlier features (OF), i.e. neurons with activation magnitudes that are significantly larger than average which can cause issues with quantization and low-precision training. The paper introduces several metrics for quantifying the existence of OFs and uses them to explore which design choices lead to the emergence of OFs. The authors then introduce an Outlier Protected Block (OP), similar to a standard transformer block, that works without explicit normalization and does not cause a loss in performance and reduces OFs. They then show the connection between signal propagation and OF and demonstrate that OP  improves signal prop and therefore reduces OFs, as predicted by their theory. Lastly, the paper discusses how the optimization, in particular learning rate and adaptivity influence  OFs.

**Strengths:**

- The topic seems relevant and I feel like the authors made a sufficiently large contribution to the research of OFs.
- I liked the connection between signal prop and OFs in Section 4. The section was formal enough while still being easy to understand. I also think it's good that the authors included a discussion about potential issues with the theory (the off-diagonals on the left-hand side of eq 4) and added experiments to show that in practice, bad signal prop seems to lead to OFEs.
- While maybe not quite sufficient in width, the experimental design itself seems good and makes the results look plausible.
- The paper seems well written and is relatively easy to understand considering its topic and while some figures are a bit too small, I overall like their design.

**Weaknesses:**

- The building blocks for the OP block should be explained in the appendix since the paper seems relevant for practitioners who might not be familiar with the literature. For example: formal definition of pre and post-norm, entropy regularization signal prop theory.
- My only significant concern is the width of the experiments. I do understand that training LLMs with 10s of billions of parameters is expensive, but it would clearly strengthen the message of the paper if OP could keep up in performance with standard normalized transformers at a much larger layer count, since it could be that this is only true for relatively "small" transformers. If training larger language models is impossible, maybe the authors could add experiments for ViTs. Transformers have obviously become very relevant in the vision domain and it would be very interesting to see if OFs are a problem in vision and, if so, if OP can help reduce them while offering similar performance.
- Section 5 felt more like an after-thought. While I think it is still interesting, I would personally much rather see a larger experimental discussion (as mentioned in the previous point) and have some of this content be moved to the appendix if necessary.

**Questions:**

- The reference to Fig 28 in 314 seems off

**Limitations:**

The authors discuss the lack of large-scale experiments and give experimental evidence for the validity of the conclusions obtained from eq (4).

---

> ### Author Rebuttal · Authors · 2024-08-07
>
> We thank the reviewer for their constructive review. We are pleased the reviewer feels we “made a sufficiently large contribution to the research of OFs”, appreciates our experimental design, and finds our paper “well written and relatively easy to understand considering its topic”. We address the raised concerns in order.
>
> **Background knowledge** Thanks for the suggestion. We assume familiarity with the terms “post-norm” and “pre-norm” as they are ubiquitous to transformer design, with relevant citations on lines 113-114. However, following the suggestion we will add more discussion to formalise these terms. For entropy regularisation we included a thorough discussion in appendix A, but will formalise the notation. For Signal Prop we believe there is sufficient relevant background and formalism in appendix A, and would be curious to know if the reviewer agrees.
>
> **Scale** Scale is discussed by both Reviewers 7V85 and cUUo. We accept the shared point of Reviewers 7V85 and cUUo that larger scales would “strengthen the message” or “further increase the significance” of our work. However, we disagree with R7V85 that scale is a ‘significant concern’, and agree with RcUUo that scaling further is “unnecessary” for the paper. We are at an academic lab, and our results at 1.2B are prohibitive for most in academia, so we disagree that the scales in our paper are “relatively small”.
>
> Moreover, the OP block itself is not the main contribution of this work. As on lines 164-171 and 346-348, as well as our title, our main focus is to understand and minimise Outlier Features, and we introduce the OP block to test the impact of normalisation layers on OFE. We believe we present an extensive study on this question, and that the scales we test at are sufficient to validate our other findings on OFs e.g. the links between OFs and signal propagation or optimisation choices.
>
> Having said that, we managed, with much difficulty, to access compute to train 7B parameter models during the author response period. Due to short notice, we spent the weekend preparing/running this experiment in a setup based on HF nanotron, with tensor+data parallelism. As such, we did not tune hyperparameters, and our hyperparameters are from the default Pre-Norm mode (see below). Moreover, we use fixed residual gains of beta=0.05 and did not implement trainable betas in the OP, which as discussed in our response to Reviewer Yyp1 would give a small improvement in perplexity.
>
> Despite this, in Figure 1* the OP block closely matches the Pre-Norm performance in loss over steps with a minimal gap at 7B scale, which we believe would be closed with hyperparameter tuning. The plot for kurtosis, averaged across layers, is Figure 2*. We see at 7B scale the OP block also has significantly lower kurtosis than Pre-LN, mirroring smaller scales. We implemented the OP block before the kurtosis logging, and when rerunning for logging our compute allowance unfortunately ran out. As such, Figure 2* stops after around 2B (of 6B tokens). Despite this, we clearly see in this case the OP block’s kurtosis has stabilised (around 8) whereas the Pre-Norm block’s kurtosis is still 450+ and increasing.
>
> 7B hyperparameters: These autoregressive language models have depth 32, and hidden width 4096, like the LLaMa-7B models. Standard LLaMa design choices (AdamW betas, RoPE, RMSNorm, cosine decay etc) are used apart from a standard GeLU MLP (instead of gated MLP), and we use max LR=3e-4 like in LLaMa-7B. We trained on the FineWeb-Edu dataset for 20K steps with batch size 72 and context length 4096, giving ~6B tokens.
>
> We hope this experiment alleviates scaling concerns. As we wrote in lines 348-349, we had no reason to suspect the OP block would struggle at bigger scales because the OP block is designed with scaling principles like signal propagation in mind. We maintain this position. We will endeavour to empirically scale to further parameter and token counts, and also investigate other settings like ViTs beyond the language settings that are typically studied in the OF literature, in future work.
>
> **Section 5** Thanks for this point. Section 5 is very important because OFs only arise during training. Thus, to understand OFs one must consider optimisation choices. This leads to several interesting findings as the Reviewer states, particularly the effect of adaptivity.
>
> The brevity of Section 5 is due to the page limit. We point the Reviewer to Appendix F which discusses why the different optimisation suggestions we propose (smaller LR and bigger adam epsilon) result in reduced OFs, by breaking down the updates to the kurtosis into different moment statistics. Reviewer cUUo even suggested moving the discussion in Appendix F to the main paper.
>
> Besides the results in the submission, we present in Table 1* a new experiment that studies the quantisation effect of our proposed optimisation and arch choices, as well as their effect on kurtosis. We take the setting of Bondarenko et al. [14] on OPT-125m, which trains on BookCorpus+Wikipedia and quantises post-training from mixed FP16/32 to W8A8 (weight-and-activation int8). We see that our optimisation interventions (bigger LR/Adam epsilon) have the same effect on kurtosis as found in Section 5, across different architectures, which further reinforces Section 5.
>
> Table 1* also shows kurtosis is highly correlated to quantisation error e.g. our OP block has lowest kurtosis and also lowest quantisation error, across optimiser choices. A complete discussion of Table 1* is in the global response. We believe this experiment brings to full circle our paper's narrative for studying the roles of different architecture/optimisation choices on OF, in which Section 5 plays a crucial role.
>
> We believe we have addressed the reviewer's concerns thoroughly in our response and would appreciate reconsideration of the score if so. We thank the reviewer again for their constructive feedback and would welcome any additional feedback.

---

> ### Comment · Reviewer_7V85 · 2024-08-08
>
> I would like to thank the authors for the rebuttal.
>
> **Background knowledge** Not too surprising given its title, Appendix A  is written like a related work section rather than a background section. I think the paper would be more accessible if the most important concepts were properly defined in a unified mathematical notation and with the most important equations not being inlined such that they are easy to find. This does not have to include all concepts cited in Appendix A, but I think methods that are used in the paper, such as QK-Norm should be defined in formal mathematical notation. After reading Section 3 again, I also noticed that the OP Block is only described as a diagram, and via the 3 changes to the Pre-Norm block. I think the paper/appendix should include a proper mathematical definition of the forward pass of the OP block since I find it difficult to think about this only in terms of changes to a pre-norm block that is not even defined anywhere in the current version of the paper. The diagram helps but I believe mathematical notation is the best way to express this and could help people that try to reimplement the OP block on their own.
>
> **Scale** I disagree with the authors that scale is not a concern when it comes to LARGE language models. That being said, I understand the issues with running large-scale experiments in academia and I understand that the analysis of OFE is the main contribution of this paper. I also want to thank the authors for running these additional experiments and I think that demonstrating that OP norm works well for a 7B model and significantly reduces kurtosis is a strong contribution since this is the standard size for small-scale LLMs that are being used in practice. That being said, I would encourage the authors to train at least one ViT for the camera-ready version since a ViT-Bor a ViT-L are much cheaper to train than LLMs and to my knowledge, the phenomenon of OFEs has been mostly studied in the language literature.
>
> **Section 5** I agree with the other reviewer that extending Section 5 in the main paper with some of the results from the Appendix and the rebuttal would be a good idea. I also liked the quantisation experiments here in particular.
>
> In conclusion, I would recommend acceptance given the author's detailed rebuttal.

---

> > ### Author Response · Authors · 2024-08-08
> > **Thank you**
> >
> > We are pleased that the reviewer appreciated our rebuttal and has updated their score. We thank them once more for the (additional) constructive feedback, which we will incorporate to improve our paper.

---

### Official Review · Reviewer_Yyp1 · 2024-07-23

**Soundness:** 3
**Presentation:** 2
**Contribution:** 3
**Rating:** 5
**Confidence:** 3

**Summary:**

The paper focuses on Outlier Features (OF) in neural networks, particularly transformers, where certain neurons exhibit significantly higher activation levels than others. OFs hinder model quantisation and their emergence during training is poorly understood. The study introduces quantitative metrics to measure OFs, such as kurtosis of neuron activation norms, and investigates how network architecture and optimization choices affect their occurrence. Practical insights are provided to mitigate OFs, emphasizing the importance of managing signal propagation during training. The Outlier Protected transformer block is proposed as a solution, removing standard Pre-Norm layers to reduce OFs without compromising convergence speed or stability. Overall, the paper advances our understanding of OF dynamics and offers strategies to address this challenging aspect of neural network training.

**Strengths:**

1. The paper introduces metrics such as kurtosis of neuron activation norms to quantify Outlier Features (OFs) in neural networks.
2. It provides insights into how architectural and optimization choices influence the emergence of OFs during transformer training.
3. The study proposes the Outlier Protected transformer block, which removes standard Pre-Norm layers to mitigate OFs without impacting training stability or convergence speed.
4. Findings are supported by empirical validation across various datasets, demonstrating the effectiveness of the proposed metrics and strategies.

**Weaknesses:**

1.	For the OP module proposed in this paper, all the parameters $\beta$  are added before the residual connection, but not after the residual connection. Why is this done? Will this result in the inability to eliminate outliers in the input X of the first block?
2.	Regarding the selection of parameter $\beta$, this paper proposes in the appendix experimental details that it is selected in advance as a fixed parameter in the "CodeParrot" experiment, and as a trainable parameter in the "Languini" experiment. So how should we choose for other scenarios? In the second experiment, what are the update details of parameter  $\beta$? Can $\beta=\mathcal{O}(1 / \sqrt{\text { depth }})$  be guaranteed all the time?
3.	In the comparative experiment of this paper on how the OP module can eliminate OFE, such as Figure 4, the OFs metric using the OP module has a trend of increasing with “Token seen”. How to explain this phenomenon? What is the effect for larger “Token seen”?

**Questions:**

see Cons

**Limitations:**

Yes

---

> ### Author Rebuttal · Authors · 2024-08-07
>
> We thank the reviewer for their time and effort in reviewing our work. We are pleased that the reviewer writes that our “findings are supported by empirical validation across various datasets, demonstrating the effectiveness of the proposed metrics and strategies”. The reviewer’s concerns largely centre around the beta residual scaling gain parameter, in the context of our OP block introduced in Section 3. We address the raised concerns in order:
>
> **Beta before residual connection not after** The reason for beta before the residual is that as per Signal Propagation theory we care about reducing the relative weight of the residual branch relative to the skip branch in order to reproduce the initialisation benefits of pre-normalisation. We outline relevant citations for this fact in lines 127-129, and perhaps the single most relevant citation for this is De and Smith 2020 (which is citation [45] in the submission). Once the skip and residual branches have been summed, it is not possible to change their relative weightings.
>
> In the inputs to the first block, e.g. output of the embedding layers in Transformers, we do not tend to observe the most extreme OFE, as can be seen across different settings in Figures 13, 18, 21, 23, 25, 27, 30, 32 (note the lack of log scale in Figures 21,23). Intuitively, this makes sense because preceeding the input to the first block there is only a single linear matrix of parameters (the embedding matrix), compared to other layers where there are the trainable weights in the non-linear attention and MLP sub-blocks.
>
> **Trainable beta or not** We are not the first to propose the residual scaling idea to enable scaling to large depths, so we based our experiments off existing practice. Many have proposed trainable beta initialised to a small value, like SkipInit (De and Smith 2020, [34]), NF-Nets (Brock et al 2021, [47]), Wortsman et al 2023 [10]. From Signal Propagation theory, Hayou et al 2021 [22] and Noci et al 2022 [25] show that $\beta=O(1/\sqrt{\text{depth}})$ is needed for a well-behaved infinite depth limit at initialisation, so the question of trainable beta is outside the current theoretical scope of the literature.
>
> Empirically, we find that trainable betas lead to slight improvements in perplexity/loss, but stress that they are not essential for the OP block to be performant and scalable, as seen in our 7B scale experiment in Figure 1* of the additional pdf page. We provide a plot of the evolution of betas in our 1.2B Languini experiments in Figure 3* of the additional pdf page, where we see the trainable betas evolve during training but stay roughly in the range of their initialisation 0.1 for both OP and Pre-LN blocks. The slight improvement with trainable betas intuitively makes sense, because beta can be seen as scaling the learning rate for parameters on the residual branch (e.g. https://arxiv.org/abs/2205.15242 or He and Hofmann [28]), and so trainable beta amounts to block-wise adaptive LRs. We think analysing the training dynamics of beta (or proposing modifications to ensure it remains $O(1/\sqrt{\text{depth}})$) is an interesting direction for future work.
>
> **Increasing kurtosis with tokens seen** We address the point on kurtosis increasing (to comparatively low values) with OP block in Section 4 (particularly on lines 236-245). There we show that kurtosis can be seen to closely track with signal propagation dynamics, which we attribute to the model creating structure in its hidden layer representation space to fit the task at hand. Identifying this relationship between signal propagation and OFs is one of our most significant contributions, as noted by Reviewers 7V85 and cUUo, in addition to the OP block.
>
> For increased “tokens seen”, in our new quantisation experiment, Table 1*, we take the OPT-125m setting of Bondarenko et al. [14], which at 12B is more than double the number of tokens seen in Figure 4. Even with this increased training length, we again see that the OP-block has significantly lower kurtosis, across various training hyperparameters, than not only the Pre-LN block but also the Gated Attention method of [14], which was designed to reduce OFs. Moreover, we note that this reduced kurtosis directly translates to reduced quantisation errors with Post-training quantisation from mixed FP16/FP32 to int8 weights-and-activations, which motivated our study of OFs in the first place.
>
> We believe we have addressed the reviewer's concerns in our response and would appreciate reconsideration of the score if so. We thank the reviewer again for their review and would welcome any additional feedback.

---

> > ### Comment · Reviewer_Yyp1 · 2024-08-09
> >
> > Thanks for the efforts of addressing my concerns. This confirms my score.

---

### Official Review · Reviewer_Lb1w · 2024-07-29

**Soundness:** 2
**Presentation:** 1
**Contribution:** 1
**Rating:** 3
**Confidence:** 4

**Summary:**

This paper addresses the issue of Outlier Features (OFs), which are neurons with activation magnitudes significantly exceeding the average during neural network training, particularly in transformers. These OFs are undesirable to model quantization, leading to high quantization errors. The authors propose quantitative metrics, like kurtosis, to measure OFs and study how architectural and optimization choices affect them. They also introduce the Outlier Protected transformer block, which removes standard Pre-Norm layers to reduce OFs without affecting convergence speed or training stability. Their findings highlight the importance of controlling signal propagation and suggest practical interventions to minimize OFs during training.

**Strengths:**

1. The definition of Outlier Features (OFs) is clear and quantitative with kurtosis score. It allows for a more objective and precise analysis of OFs across different neural network architectures.
2. The paper presents non-trivial amount of empirical studies that analyze the impact of various architectural and optimization choices on OFs.

**Weaknesses:**

1. Writing Quality: The paper is poorly written, with comments and claims listed in a piecemeal fashion rather than being unified into a structured story. Glitches: on line 59, the phrase "we matrix multiply" is unclear. The writing overall lacks fluidity and coherence across sections.

2. Insufficient Evidence for Outlier Protected Block (OP): In Section 3, the authors claim that the Outlier Protected Block (OP) reduces Outlier Features. However, the evidence provided is insufficient to support this contribution. The figures show a rough trend, but more thorough investigation is needed:
a) Is this observation consistent across all layers? (Figure 4 shows different trends in "final layers" compared to "early layers.")
b) Does the claim hold under different training schemes?
c) Are there existing baselines that achieve similar effects?
d) What are the gains of using the OP block? How much does it improve network quantization efficiency?

3. Relation Between OF and Signal Propagation: In Section 4, the authors assert that the OF phenomenon is related to Signal Propagation. However, both metrics are mathematically defined by matrix X in different ways, so it is unsurprising that they are related. This diminishes the novelty of the finding.

4. Lack of Depth in Training Design Analysis: In Section 5, while the authors provide several training designs and analyze their effects on OFE, the analysis lacks depth. The paper should offer deeper insights into why these effects occur rather than presenting rough findings and claims.

Overall, the paper feels more like a technical report than a conference paper ready for publication. It lacks a solid contribution and fails to adequately answer the research question of "why OFs emerge during training." The authors acknowledge this in Section 6: "Perhaps the most important is understanding how signal propagation evolves during training, and which factors affect this." This is the core question that the paper should address comprehensively.

**Questions:**

See weakness.

**Limitations:**

Yes, in Section 6.

---

> ### Author Rebuttal · Authors · 2024-08-07
>
> We thank Reviewer Lb1W for their time. However, the reviewer has made several assertive yet unsubstantiated criticisms of our work, which oppose all the other reviewers. We are concerned by these criticisms as they lack constructive feedback and could be perceived as overly critical from our view. All other reviewers gave scores leaning towards accept, with RcUUo giving the maximum score 10, in contrast to RLb1W’s score 3. We hope our rebuttal will address these points effectively and convince RLb1W of our paper’s merit, hopefully leading to an improved score.
>
> **Writing Quality**: RLb1W claims the paper is “poorly written” and “overall lacks fluidity and coherence across sections”. This is in stark contrast to R7V85 and RcUUo who praise the paper for being “well written”, and “extremely well written”. RLb1W provides no evidence for this purported poor writing, besides line 59 where the verb form of the noun “matrix multiplication” is used. As such, this criticism is unsubstantiated. With regards to fluidity/coherence, we gave significant thought to the paper’s structure/flow, as seen via the key takeaway boxes between sections. The effect of this, quoting R7V85, is: “the paper is relatively easy to understand considering its topic”.
>
> **Insufficient Evidence for OP Block**: We contrast RLb1W’s concern with RYyp1 and RcUUo, who write: “findings are supported by empirical validation across various datasets, demonstrating the effectiveness of the proposed metrics and strategies”, and “this is a thorough and careful study, making claims that are well supported by the experiments”.
>
> More concretely, Fig 4 plots all different layers (the legend samples layers to save space) which addresses weakness 2a and we discuss later layers and their connection to signal propagation in lines 236 to 245 of the submission. Besides Fig 4, we show the effectiveness of the OP block for OF reduction in five other Figs (2, 13, 14,  32, and 38) which constitute extensive ablations across datasets, entropy regulation mechanism, centring in kurtosis metric, and the effect of norm in MLP. This answers weakness 2b regarding training schemes, and so we view weaknesses 2a+b to be unfounded. We respond to weakness 2c-d (and provide additional evidence for 2b) in the next point.
>
> Also, in Figures 1* and 2* of the additional pdf page we show the effectiveness of the OP block at 7B scale.
>
> **Quantisation Experiments** In response to weakness 2c-2d, we present new quantisation results in Table 1* of the additional pdf page, which compares OP to Pre-LN and the best baseline (Gated Attention) from Bondarenko et al [14]. To our knowledge [14] is the only existing paper that has studied architecture and OFs. We test these models across training schemes, and find the OP block has significantly lower kurtosis across different optimiser choices. This lower kurtosis directly translates to improved quantisation, as seen in the significantly smaller drops in performance when going from mixed FP16/32 to W8A8 (weight and activation in int8) with OP, compared to Pre-LN and Gated Attention. We provide a complete discussion of the results of Table 1* in the global response.
>
> **Lack of novelty of relation between OF and Signal Propagation**: To us, this is again unfounded: according to RcUUo, we "make strides towards understanding OFE by combining ideas from Signal Prop and entropy collapse (e.g.), in ways that are insightful, novel, and excellently motivated.” We take it as a strength not weakness that RLb1W sees the link between OF and Signal Prop unsurprising, as it implies the significant effort taken to make the paper accessible was successful in Section 4 (going back to the point on writing quality). This criticism also contradicts R7V85: who “liked the connection between signal prop and OFs in Section 4. The section was formal enough while still being easy to understand." We add that a result that is unsurprising in hindsight is not necessarily unsurprising.
>
> **Lack of depth in training design analysis**: We disagree: due to the page limit, we could not fit all our optimiser choices analysis into Section 5, but this does not mean our analysis lacks depth. As also discussed in line 339, we refer the reviewer to Appendix F, where we dive deeper into the reasons why our proposed optimisation changes lead to reduced OFs, by breaking down the kurtosis updates into different moment statistics. Reviewer cUUo read Appendix F and suggested we include it in the main paper.
>
> Moreover, our claims regarding the importance of large adaptive learning rates for OFs are not “rough findings”, but instead substantiated across multiple architectures and datasets. Namely, Figures 7, 24 and 25 show the effect of large learning rates, and our claims regarding adaptivity with Adam epsilon are supported by Figures 8, 27, 30. As such, we again fail to see the basis of this criticism.
>
> In addition to the existing depth of analysis of optimiser choices in the submission, we refer to our new quantisation experiment in Table 1*, which provides further evidence of the effect of optimiser choices we identify in Section 5 in terms of OFE, on a new dataset and across multiple architectures.
>
> **More like a technical report than a conference paper ready for publication**: We are completely surprised by this assertion. The quote from lines 353-354 is misquoted without context, and thus misleading. The missing context clearly refers to theoretical understanding; we give many empirical insights into signal propagation training dynamics in Section 4. We leave RLb1W with quotes from R7V85 and RcUUo, who write: “I feel like the authors made a sufficiently large contribution to the research of OFs” and “this work will be appreciated by multiple communities for its wealth of insights”.
>
> We believe we have addressed the reviewer's concerns thoroughly in our response and would appreciate reconsideration of the score if so. We would also welcome any additional feedback.

---

> > ### Comment · Reviewer_Lb1w · 2024-08-11
> >
> > Thank you for your response. However, it appears that the revision did not adequately address the core concerns, instead referencing unhelpful comments from other reviewers. The citations from other evaluations do not substantiate your responses unless they are accompanied by supporting evidence. Consequently, my concerns remain unresolved, and I have retained my original score.
> >
> > 1. **Unsatisfactory Writing Quality**: The manuscript suffers from numerous stylistic and structural weaknesses, making it difficult to follow. Starting in Section 3, from Line 87-103, the paragraph introduces background on normalization but fails to signal this topic in the opening sentence. Line 104-177, abruptly begins with "In Figure 2" without a transition from the previous paragraph, leaving the reader unclear about the figure's contect...... Limited by time and energy, the reviewer can not point out all the places that writing can be improved since there are too many. Most of the paragraphs in the document lack a coherent top-down structure.
> >
> > 2. **Insufficient Support for Architectural Contribution**: The "Outlier Protected transformer block" is presented as a highlighted innovation, yet its description is confined to a single paragraph (Lines 146-163). For an effective presentation of a new architecture, consider the detailed exposition found in Section 4 of Reference [14] (frequently mentioned by the author). If there are substantial results supporting the proposed architecture, they should be prominently featured, akin to how Tables 2 and 3 are presented in [14]. The lack of a unified section detailing the experimental approach, akin to Section 5 in Reference [14], makes it challenging for the readers to effectively understand the context and validity of your findings.
> >
> > 3. **Insignificant Observation**: My concern regarding the relationship between Outlier Features (OF) and Signal Propagation has not been addressed. You provide several opposing subjective comments but fail to engage with the core issue. Specifically, you claim a significant observation that Signal Propagation relates to OF, yet both measures you discuss (Kurtosis and the Gram matrix) are mathematical functions of the same variable, X. The inherent mathematical relationship between these functions should not be surprising, and you have not clarified why this finding is noteworthy.
> >
> > 4. **Insufficient Fundamental Insight**: The title and abstract of your paper promises a deep understanding of why OFs emerge during training. However, the empirical analysis over optimization choices in Section 5, while useful, does not address this core question from a fundamental perspective. An effective discussion should integrate more profound insights into the nature and implications of OFs within neural network dynamics. To enhance the paper, one possible analysis technique can be associated with the neural tangent kernel or training dynamics. For guidance on conducting this type of rigorous research, I recommend referring to [A,B,C], which illustrate how scholars **deeply** uncover the underlying causes of phenomena during training.
> >
> > Overall, the manuscript requires significant improvements in writing, experimental design, and technical depth to meet the publication standards.
> >
> > [A] Kumar, A., Raghunathan, A., Jones, R. M., Ma, T., & Liang, P. Fine-Tuning can Distort Pretrained Features and Underperform Out-of-Distribution. In International Conference on Learning Representations 2022.
> >
> > [B] Kou, Y., Chen, Z., Chen, Y., & Gu, Q. Benign overfitting in two-layer ReLU convolutional neural networks. In International Conference on Machine Learning (pp. 17615-17659). PMLR 2023.
> >
> > [C] Mahankali, A. V., Hashimoto, T., & Ma, T. One Step of Gradient Descent is Provably the Optimal In-Context Learner with One Layer of Linear Self-Attention. In The Twelfth International Conference on Learning Representations 2024.

---

> > > ### Author Response · Authors · 2024-08-12
> > > **Additional author response**
> > >
> > > We thank Reviewer Lb1w for engaging during the rebuttal period. We respond to Reviewer Lb1w with the following points:
> > >
> > > **On the depth of the experimental analysis**
> > > In their original review, Reviewer Lb1w stressed the importance of quantisation experiments to have more thorough analysis of the effectiveness of the OP block (which we agree). We have since run the experiments, reported in Table 1* in the attached one-page pdf (see details and discussion in the global repsonse). In summary, the proposed modifications do enhance low precision training, which nicely correlates with the observation of less OFs. We would appreciate it if the Reviewer acknowledged this and potentially reassessed their considerations on the depth of our analysis.
> > >
> > > **Insufficient Support for Architectural Contribution**
> > > Given what we said above, we politely disagree that there is not enough experimental evidence for the architectural modification. In particular, with regards to the four axes mentioned in the initial review, we believe that (a), (b) and (c) are resolved in the original rebuttal and references to the appendix. Finally, see above for the quantization experiment, requested in (d).
> > >
> > > We also add that Reviewer Lb1w has raised new concerns, relating to the description of the OP block and lack of unified experimental section, in their response that were not raised in their original review. Reviewer 7V85 raised similar concerns in their response to our rebuttal, which we have already incorporated into our revised work.
> > >
> > > Finally, we have demonstrated the efficacy of the OP block at 7B scale in Figures 1* and 2* of the additional rebuttal pdf, and would appreciate it if the Reviewer acknowledges this.
> > >
> > > **Insignificance of Signal Propagation link**
> > > The reviewer has raised the “insignificance/noteworthiness” of the OF and Signal Prop link as a new concern in their most recent response. The original review (weakness 3) criticised the “surprisingness/novelty” of the finding, which we addressed in our rebuttal.
> > >
> > > The significance/noteworthiness of the relation between OF and Signal Prop is that it: (i) establishes a connection between two previously unconnected sub-branches of the literature which allows ideas from both to help drive progress in the other e.g. the identification that normalisation layers make Signal Prop worse during training (Figures 5 and 6), and (ii) helps us motivate why choices from the Signal Prop literature that improve Signal Prop (like downweighted residuals [22, 25] or shaped activations [63, 24, 23]) also help to reduce OFs, which we provide empirical evidence for in Figures 5 and 21.
> > >
> > > **Insufficient Fundamental Insight**
> > > As we openly state in our limitations section, there are a lot of interesting future directions for mathematically rigorous studies that would build on the results of our paper, but are outside the scope of our work. We make the distinction between “mathematically rigorous” and “fundamental” as a lot of empirical results in deep learning are fundamental in our eyes, e.g. the Edge of Stability [30].
> > >
> > > The reviewer’s suggestion of the “NTK” or “training dynamics” as theoretical analysis techniques are insufficient for OFE because the NTK limit is incompatible with feature learning, which is necessary for OFs, as we state in lines 254-256 of the submission. We are unsure what is meant by “training dynamics” as an analysis technique here.
> > >
> > > We do not wish to come across as overly harsh on the Reviewer Lb1w, but we find the phrasing “[..] which illustrate how scholars deeply uncover the underlying causes of phenomena …” to be somewhat inappropriate, as it seems to hierarchically separate such “scholars” and the authors of this paper.
> > >
> > > **Unsatisfactory Writing Quality**
> > > We have already significantly improved the manuscript, particularly with regards to a more extensive description of the OP block. The reviewer raises 2 additional concrete examples of what they perceive as poor writing: the openings of lines 87-103 and lines 104-117. We thank the reviewer for these examples but politely disagree and view these as subjective stylistic preferences instead of substantial criticisms that merit the review’s score. We appreciate it takes time and energy to review a paper, but strongly believe there is a burden of proof here and the claimed criticisms to writing quality need to be backed up with more evidence to be warranted.
> > >
> > >
> > > **Comments from Other Reviews**
> > > Finally, we think it is somewhat inappropriate to call the comments of the other reviewers “unhelpful”: the fellow reviewers have read the same manuscript and are entitled to their own viewpoints. It is natural for there to be disagreement between reviewers, but the level of disagreement with our paper this far into the author-reviewer discussion period is unprecedented. With our rebuttal and additional response, as well as the comments of the other reviewers, we once again politely ask Reviewer Lb1w to reconsider their appraisal of our work.

---

### Author Rebuttal · Authors · 2024-08-07

We thank the reviewers for their time in reviewing our work. We are encouraged that three out of four reviewers gave scores leaning towards accept, with an average score of 6 across all four reviews. In particular, reviewer cUUo gave the highest possible score of 10, commenting: “this work will be appreciated by multiple communities for its wealth of insights”.


There is, however, a disappointingly large variance in the scores, with Reviewer Lb1w giving a score of 3 and writing that the paper feels “more like a technical report than a conference paper ready for publication”. We believe this, and several of Reviewer Lb1w’s other criticisms, are unsubstantiated with no evidence, which we have discussed at length in our individual response to Reviewer Lb1w.  Moreover, many of Reviewer Lb1w’s criticisms are in direct opposition to praise received from all three other reviews. We urge the reviewers to reach a more consistent appraisal of our work during the discussion period. Through our author response, we hope to convince Reviewers Lb1w, 7V85 and Yyp1, to raise their scores.

In the individual responses, we believe we have addressed all concerns of each individual reviewer, so in the remainder of the global response we would like to draw attention to additional results in the rebuttal that may be of shared interest to the reviewers.

*Notation for additional results* Whenever we write an asterisk after a Figure or Table, e.g. Table 1*, this means we refer to a Figure/Table from the additional pdf page. Unasterisked Figures/Tables refer to the original submission.


**Quantisation Experiment and Section 5**
In Table 1* we present an experiment testing the combined effects of our different architecture and optimisation choices in affecting quantisation errors. This goes back to our original motivation in studying OFs, and how we can minimise their emergence with different design choices.

In Table 1*, we take the 125m scale OPT setting of Bondarenko et al. [14], which trains models in standard mixed FP16/FP32 precision on BookCorpus+Wikipedia for around 12B tokens seen. Post training, we quantise to int8 weight-and-activations (W8A8). We use the same  quantisation recipe and reproduce their results. We present results with standard deviations from 3 seeds in the post training quantisation step. We note that the GPTQ paper https://arxiv.org/abs/2210.17323 shows that post training quantisation is harder at smaller scales like this 125m setting than at larger scales.

Table 1* compares both the standard precision performance and also quantisation performance across architecture and optimiser choices. We compare 3 different architectures: 1) standard Pre-LN, 2) the Gated Attention baseline of [14] (which was their best baseline), and 3) our OP block, as well 4 different optimisation setups that are added one after another on top of each other: 1) the default hyperparameters of [14], 2) removing the dropout regularisation, 3) increasing the maximum LR from 4e-4 to 1e-3, and 4) increasing Adam Epsilon from 1e-8 to 1e-5.

Optimiser choices 2) and 3) were designed to improve standard precision performance, albeit potentially at the detriment of quantisation performance due to OFs (as suggested by our findings with increase LRs in Section 5). Optimiser choice 4) was chosen to improve and reduce OFs following our contributions in Section 5, and thereby hopefully improve quantisation errors.

As seen in Table 1*, our findings throughout the rest of our paper are validated. Firstly, the kurtosis is seen to be highly correlated with quantisation error, for example the Pre-LN model has consistently high kurtosis, and also consistently catastrophic performance at W8A8 (>45 perplexity increase in W8A8 across all optimiser settings). Secondly, our OP block has consistently low kurtosis (around or below 10), and this directly translates to low quantisation error (the difference in perplexity between standard mixed precision at W8A8 is around or below 0.7 across all optimiser settings). This low kurtosis/quantisation error of the OP block holds true even for aggressive optimiser hyperparameters like large LRs that increase kurtosis, but also improve standard mixed precision performance. Finally, the baseline of [14] struggles when dropout is removed and a large learning rate is used, with kurtosis of ~30 and large quantisation error around 2-3 perplexity but our suggestion of increasing Adam epsilon in Section 5 reduces the kurtosis back down to 15 which translates to a quantisation error of 0.77 perplexity.

Indeed, the best quantised W8A8 model in this case is actually the Gated Attention baseline of [14] with our optimisation choices from Section 5, which is very slightly better than our OP block (15.54 vs 15.62 perplexity). In relation to comments from reviewers 7V85 and Lb1w, this highlights that Section 5 is a fundamental and key section of the paper, and that our contributions to understanding the impact of optimisation choices on OFs are also important, in addition to our architectural contributions.

**Scaling experiment**
We agree with Reviewer cUUo that scaling further beyond 1.2B parameters is “unnecessary but would further increase the significance of this work”. To address scale, over the weekend of the response period, we were able to access resources to run limited experiments at 7B, which we present in the additional pdf page. In Figure 1* we see that the OP block closely matches standard Pre-Norm in terms of loss curves, and in Figure 2* we see that the Pre-Norm block suffers from significantly worse OFs, in the sense that its kurtosis is much larger. These results mirror our findings at smaller scales in the submission. Further details and context can be found in our response to Reviewer 7V85.

---

### Decision · Program_Chairs · 2024-09-25

**Decision:**

Accept (poster)

**Comment:**

This paper presents a study on reducing outlier features in neural network training. The paper has sparked considerable discussion regarding its novelty, experimental rigor, and practical implications for neural network quantization and training dynamics. The major concern highlighted is the paper's ability to convincingly demonstrate the efficacy of the proposed techniques across varying network architectures and larger datasets. In their rebuttal, the authors have addressed these issues comprehensively, providing additional data and clarifications that enhance the original submission. While one reviewer remains critical, focusing particularly on the paper's writing style and depth of technical contribution, the other reviewers acknowledge the paper’s potential impact and the rigor of the experimental setup.

Given the mixed nature of the reviews but considering the strong rebuttal and the additional experiments provided, the decision leans towards acceptance, contingent upon the successful incorporation of suggested improvements in its final version.